# The Influence of Environmental Variability on the Biogeography of Coccolithophores and Diatoms in the Great Calcite Belt

Helen E. K. Smith[1,2], Alex J. Poulton[1,3], Rebecca Garley[4], Jason Hopkins[5], Laura C. Lubelczyk[5], Dave T. Drapeau[5], Sara Rauschenberg[5], Ben S. Twining[5], Nicholas R. Bates[2,4], William M. Balch[5]

[1]National Oceanography Centre, European Way, Southampton, SO14 3ZH, U.K.
[2]School of Ocean and Earth Science, National Oceanography Centre Southampton, University of Southampton Waterfront Campus, European Way, Southampton, SO14 3ZH, U.K.
[3]Present address: The Lyell Centre, Heriot-Watt University, Edinburgh, EH14 4AS, U.K.
[4]Bermuda Institute of Ocean Sciences, 17 Biological Station, Ferry Reach, St. George's GE 01, Bermuda.
[5]Bigelow Laboratory for Ocean Sciences, 60 Bigelow Drive, P.O. Box 380, East Boothbay, Maine 04544, USA.

*Correspondence to*: Helen E.K. Smith (helen.eksmith@gmail.com)

**Abstract.** The Great Calcite Belt (GCB) of the Southern Ocean is a region of elevated summertime upper ocean calcite
concentration derived from coccolithophores, despite the region being known for its diatom predominance. Overlap of two major phytoplankton groups, coccolithophores and diatoms, in the dynamic frontal systems characteristic of this region, provides an ideal setting to study environmental influences on the distribution of different species within these taxonomic groups. Samples for phytoplankton enumeration were collected from the upper mixed layer (30 m) during two cruises, the first to the South Atlantic sector (Jan-Feb 2011; $60^o$ W-$15^o$ E and 36-$60^o$ S) and the second in the South Indian sector (Feb-
Mar 2012; 40-$120^o$ E and 36-$60^o$ S). The species composition of coccolithophores and diatoms was examined using scanning electron microscopy at 27 stations across the Sub-Tropical, Polar, and Sub-Antarctic Fronts. The influence of environmental parameters, such as sea-surface temperature (SST), salinity, carbonate chemistry (pH, partial pressure of $CO_2$ ($pCO_2$), alkalinity, dissolved inorganic carbon), macro-nutrients (nitrate+nitrite, phosphate, silicic acid, ammonia), and mixed layer average irradiance, on species composition across the GCB, was assessed statistically. Nanophytoplankton (cells 2-20 μm)
were the numerically abundant size group of biomineralizing phytoplankton across the GCB, with the coccolithophore *Emiliania huxleyi* and diatoms *Fragilariopsis nana*, *F. pseudonana* and *Pseudo-nitzschia* sp. the most numerically dominant and widely distributed. A combination of SST, macro-nutrient concentrations and $pCO_2$ were the best statistical descriptors of biogeographic variability of biomineralizing species composition between stations. *Emiliania huxleyi* occurred in silicic acid-depleted waters between the Sub-Antarctic Front and the Polar Front; a favorable environment for this species after
spring diatom blooms remove silicic acid. Multivariate statistics identified a combination of carbonate chemistry and macro-nutrients, co-varying with temperature, as the dominant drivers of biomineralizing nanoplankton in the GCB sector of the Southern Ocean.

# 1 Introduction

The Great Calcite Belt (GCB), defined as an elevated particulate inorganic carbon (PIC) feature occurring alongside seasonally elevated chlorophyll $a$ in austral spring and summer in the Southern Ocean (Fig. 1; Balch et al., 2005), plays an important role in climate fluctuations (Sarmiento et al., 1998, 2004), accounting for over 60% of the Southern Ocean area (30-60$^o$S; Balch et al., 2011). The region between 30-50$^o$S has the highest uptake of anthropogenic carbon dioxide ($CO_2$) alongside the North Atlantic and North Pacific Oceans (Sabine et al., 2004). Our knowledge of the impact of interacting environmental influences on phytoplankton distribution in the Southern Ocean is limited. For example, we do not yet fully understand how light and iron availability, or temperature and pH, interact to control phytoplankton biogeography (Boyd et al., 2010, 2012; Charalampopoulou et al., 2016). Hence, if model parameterizations are to improve (Boyd and Newton, 1999) to provide accurate predictions of biogeochemical change, a multivariate understanding of the full suite of environmental drivers is required.

The Southern Ocean has often been considered as a micro-plankton (20-200 μm) dominated system with phytoplankton blooms dominated by large diatoms and *Phaeocystis* sp. (e.g., Bathmann et al., 1997; Poulton et al., 2007; Boyd, 2002). However, since the identification of the GCB as a consistent feature (Balch et al., 2005; 2016) and recognition of pico- (< 2 μm) and nanoplankton (2-20 μm) importance in High Nutrient Low Chlorophyll (HNLC) waters (Barber and Hiscock, 2006), the dynamics of small (bio-)mineralizing plankton and their export need to be acknowledged. The two dominant biomineralizing phytoplankton groups in the GCB are coccolithophores and diatoms. Coccolithophores are generally found north of the PF (e.g., Mohan et al., 2008), though *Emiliania huxleyi* has been observed as far south as 58$^o$S in the Scotia Sea (Holligan et al., 2010), at 61$^o$S across Drake Passage (Charalampopoulou et al., 2016) and 65$^o$S south of Australia (Cubillos et al., 2007).

Diatoms are present throughout the GCB, with the Polar Front marking a strong divide between different size fractions (Froneman et al., 1995). North of the PF, small diatom species such as *Pseudo-nitzschia* sp. and *Thalassiosira* sp. tend to dominate numerically, whereas large diatoms with higher silicic acid requirements (e.g. *Fragilariopsis kerguelensis*) are generally more abundant south of the PF (Froneman et al., 1995). High abundances of nanoplankton (coccolithophores, small diatoms, chrysophytes) have also been observed on the Patagonian shelf (Poulton et al., 2013) and in the Scotia Sea (Hinz et al., 2012). Currently, few studies incorporate small biomineralizing phytoplankton to species level (e.g., Froneman et al., 1995; Bathmann et al., 1997; Poulton et al., 2007; Hinz et al., 2012). Rather, the focus has often been on the larger and non-calcifying species in the Southern Ocean due to sample preservation issues (i.e., acidified Lugol's solution dissolves calcite and light microscopy restricts accurate identification to cells > 10 μm; Hinz et al., 2012). In the context of climate change and future ecosystem function, the distribution of biomineralizing phytoplankton is important to define when considering phytoplankton interactions with carbonate chemistry (e.g., Langer et al., 2006; Tortell et al., 2008) and ocean biogeochemistry (e.g., Baines et al., 2010; Assmy et al., 2013; Poulton et al., 2013).

The GCB spans the major Southern Ocean circumpolar fronts (Fig. 1a): the Sub-Antarctic front (SAF); the Polar Front (PF); the Southern Antarctic Circumpolar Current Front, SACCF); and occasionally, the Southern Boundary of the Antarctic Circumpolar Current (ACC, see Tsuchiya et al., 1994; Orsi et al., 1995; Belkin and Gordon, 1996). The Subtropical Front (STF; at approximately 10° C) acts as the northern boundary of the GCB and is associated with a sharp increase in PIC

southwards (Balch et al., 2011). These fronts divide distinct environmental and biogeochemical zones making the GCB an ideal study area to examine controls on phytoplankton communities in the open ocean (Boyd, 2002; Boyd et al., 2010). High PIC concentration observed in the GCB (1 µmol PIC $L^{-1}$) compared to the global average (0.2 µmol PIC $L^{-1}$) and significant quantities of detached *E. huxleyi* coccoliths (in concentrations > 20,000 coccoliths $mL^{-1}$; Balch et al., 2011) both characterize the GCB. The GCB is clearly observed in satellite imagery (e.g., Balch et al., 2005; Fig. 1b;) spanning from the

Patagonian Shelf (Signorini et al., 2006; Painter et al., 2010), across the Atlantic, Indian and Pacific Oceans and completes the Antarctic circumnavigation via the Drake Passage.

GCB waters are characterized as High Nitrate Low Silicate Low Chlorophyll (HNLSiLC; e.g., Dugdale et al., 1995; Leblanc et al., 2005; Moore et al., 2007; Le Moigne et al., 2013), where dissolved iron (dFe) is considered an important control on microplankton (>20 µm) growth (e.g., Martin et al., 1990; Gall et al., 2001; Venables and Moore, 2010). Sea-surface

temperature (SST) gradients are a driving factor behind phytoplankton biogeography and community composition (Raven and Geider, 1988; Boyd et al., 2010). The influence of environmental gradients on biomineralizing phytoplankton in the Scotia Sea and Drake Passage has also been assessed (Hinz et al., 2012; Charalampopoulou et al., 2016). However, the controls on the distribution of biomineralizing nanoplankton are yet to be established for the wider Southern Ocean and GCB. Previous studies have predominantly focused on a single environmental factor (e.g., Eynaud et al., 1999) or

combinations of temperature, light, macronutrients and dFe (e.g., Poulton et al., 2007; Mohan et al., 2008; Balch et al., 2016) to explain phytoplankton distribution. The inclusion of carbonate chemistry as an influence on phytoplankton biogeography is a relatively recent development (e.g., Charalampopoulou et al., 2011, 2016; Hinz et al., 2012; Poulton et al., 2014; Marañón et al., 2016). Furthermore, natural variability in ocean carbonate chemistry and the resulting impact on in situ phytoplankton populations remains a significant issue when considering the impact of future climate change.

Increasing concentration of dissolved $CO_2$ in the oceans is resulting in 'ocean acidification' via a decrease in ocean pH (Caldeira and Wickett, 2003). In the high latitudes, where colder waters enhance the solubility of $CO_2$ and reduce the saturation state of calcite, there may be potential detrimental effects on calcifying phytoplankton (Doney et al., 2009). However, this may be species- (Langer et al., 2006) or even strain-specific (Langer et al., 2011), showing an optimum-response when the opposing influences of pH and bicarbonate are considered in a substrate-inhibitor concept (Bach et al.,

2015). The response of non-calcifiers (e.g., diatoms) to ocean acidification is a greater unknown but no less important given their ~40 to 50% contribution to global primary production (e.g., Tréguer et al., 1995; Sarthou et al., 2005). Tortell et al., (2008) observed a switch from small to large diatom species with increasing $CO_2$, indicating a potential change in future

community structure. Large phytoplankton species (>50 µm) may also have physiological traits to withstand changes in ocean chemistry over smaller (<50 µm) celled species (Flynn et al., 2012), as well as potentially being less susceptible to grazing pressure (Assmy et al., 2013). Alternatively, there may be a shift towards small phytoplankton groups due to the expansion of low-nutrient subtropical regions (Bopp et al., 2001; Bopp, 2005). The response of Southern Ocean phytoplankton biogeography to future climate conditions, including ocean acidification, is complex (e.g., Charalampopolou et al., 2016; Petrou et al., 2016; Deppeler and Davidson, 2017) and therefore understanding existing relationships between *in situ* phytoplankton communities and ocean chemistry is an important stepping-stone for predicting future changes.

Here, we assess the distribution of coccolithophore and diatom species in relation to the environmental conditions encountered across the GCB. Diatom and coccolithophore cell abundances were obtained from analysis of scanning electron microscopy (SEM) images, and their distribution statistically assessed in relation to SST, salinity, mixed layer average irradiance, macronutrients and carbonate chemistry. Herein, we examine the spatial differences within the biomineralizing phytoplankton in the GCB, the main environmental drivers behind their biogeographic variability and the potential effects of future carbonate chemistry perturbations.

## 2 Methods

### 2.1 Sampling area

Two cruises were undertaken in the GCB during 2011 and 2012 (http://www.bco-dmo.org/project/473206). The Atlantic sector of the Southern Ocean (GCB1) was sampled from 11[th] January to 16[th] February 2011 onboard the R/V *Melville*, between Punta Arenas, Chile and Cape Town, South Africa (Balch et al., 2016; Fig. 1). The Indian sector of the Southern Ocean (GCB2) was sampled from 18[th] February to 20[th] March 2012 onboard the R/V *Revelle* between Durban, South Africa and Fremantle, Australia (Fig. 1). Water samples were taken at 27 stations across a latitudinal gradient ranging from $38^{o}$ S to $60^{o}$ S and a longitudinal gradient ranging from $60^{o}$ W to $120^{o}$ E during the GCB cruises, which enabled sampling of the major oceanographic features of this region.

### 2.1 Physiochemical environmental conditions

Water samples were collected from the upper 30 m of the water column using a Niskin bottle rosette and CTD profiler for sea surface temperature, salinity, chlorophyll *a* (Chl *a*), nitrate plus nitrite (NOx), ammonia ($NH_4$), phosphate ($PO_4$), silicic acid ($Si(OH_4)$), and carbonate chemistry. Nutrient analyses of NOx, $PO_4$, $Si(OH_4)$ and $NH_4$ were run on a Seal Analytical continuous-flow AutoAnalyzer 3, while salinity was determined using a single Guildline Autosal 8400B stock salinometer (S/N 69-180). Chlorophyll *a* was sampled in triplicate following Joint Global Ocean Flux Studies (JGOFS; Knap. et al, 1996) protocols. Mixed layer depths were calculated from processed CTD data applying a criteria of a 0.02 kg m$^{-3}$ density change from the 5 m value (Arrigo et al., 1998). Daily Photosynthetically Active Radiation (PAR, mol PAR m$^{-2}$ d$^{-1}$) was

estimated from eight-day composite Aqua MODIS data from the closest time and latitude-longitude point (averages were taken where necessary). Mixed layer average irradiance ($\bar{E}_{MLD}$) was calculated from daily PAR following Poulton et al., (2011).

Water samples were collected for total dissolved inorganic carbon ($C_T$) and total alkalinity ($A_T$) following standardized
methods and analyzed using a Versatile Instrument for the Determination of Titration Alkalinity (VINDTA) with precision and accuracy of ±1 µmol kg$^{-1}$ (Bates et al., 1996; Bates et al., 2012). The remaining carbonate chemistry parameters were calculated from the $C_T$ and $A_T$ values using CO2SYS (Lewis and Wallace, 1998) and CO2calc (Robbins et al., 2010), with the carbonic acid dissociation constants of Mehrbach et al., (1973) refitted by Dickson and Millero (1987). This includes computation of the saturation state ($\Omega$) for calcite (i.e., $\Omega_{calcite}$).

**2.2 Phytoplankton enumeration**

Samples for biomineralizing phytoplankton community structure were taken from the upper 30 m of the water column. One litre seawater samples were collected and pre-filtered through a 200 μm mesh to remove any large zooplankton. Seawater samples were gently filtered through a 25 mm, 0.8 μm Whatman® polycarbonate filter placed over a 200 μm backing mesh to ensure an even distribution of cells across the filter. Filters were rinsed with ~5 mL potassium tetraborate (0.02 M) buffer
solution (pH = 8.5) to prevent salt crystal growth and PIC dissolution, air dried and stored in petri slides in the dark with a desiccant until further analysis.

To identify coccolithophores to the species level, each sample was imaged using the SEM methodology of Charalampopoulou et al., (2011). A central portion of each filter was cut-out, gold-coated and 225 photographs were taken at a magnification of 5000x (equivalent to ~1 mm$^2$; GCB1) or 3000x (~2.5 mm$^2$; GCB2) using a Leo 1450VP SEM (Carl Zeiss,
Germany). Detached coccoliths and whole coccolithophore cells (coccospheres) were identified following Young et al., (2003). Diatoms and other recognizable protists were identified following Hasle and Syvertsen (1997) and Scott and Marchant (2005). Where a confident species level identification was not possible, cells were assigned to the level of genera (e.g., *Chaetoceros* sp. or *Pappamonas* sp.). Each species identified was enumerated using the freeware ImageJ (v1.44o) for all 225 images or until 300 cells (or coccoliths) were counted. A minimum of 10 random images was picked for enumeration
when species were in high abundance (>1000 cells mL$^{-1}$). The abundance of each species was calculated following Eq. (1):

$$Cells\ mL^{-1} = (C \times F/A)/V \tag{1}$$

where C is the total number of cells (or coccoliths) counted, A is the area investigated (mm$^2$), F is the total filter area (mm$^2$) and V is the volume filtered (mL).

## 2.3 Statistical analysis

Multivariate statistics (PRIMER-E v.6.1.6; Clarke and Gorley, 2006) were used to examine spatial changes in coccolithophore and diatom abundance, species distribution and the influence of environmental variability on biogeography (e.g., Charalampopoulou et al., 2011, 2016).  Environmental data was initially assessed for skewness, most likely due to
strong chemical gradients across fronts. Heavily left-skewed variables (NOx, silicic acid and $NH_4$) were log(V+0.1) transformed to reduce skewness and stabilize variance. Other environmental data, including SST, salinity, $\bar{E}_{MLD}$, NOx, silicic acid, $NH_4$, pH, $p\mathrm{CO}_2$ and $\Omega_{calcite}$ was then normalized to a mean of zero and a standard deviation of one, and Euclidean distance was then used to determine spatial changes in these parameters. A principal component analysis (PCA) was used to simplify environmental variability, by combining the more closely correlated variables and the relative influence of the
environmental variables within the data (Clarke, 1993; Clarke and Warwick, 2001; Clarke and Gorley, 2006).

Coccolithophore and diatom species diversity was assessed as the total number of species (S), and Pielou's evenness index (J') which assesses how evenly the count data was distributed between the different species present (before further statistical analysis). Species with cell counts of less than 1 cell $mL^{-1}$, and/or consistently representing less than 1% of the total cell abundance, were excluded from multivariate statistical analysis to reduce the influence of rare species. Analysis of
coccolithophore and diatom community structure was carried out on standardized and square root transformed cell abundance (to reduce the influence of numerically abundant species) using a Bray-Curtis similarity matrix. Bray-Curtis similarity describes the percentage similarity (or dissimilarity) between different communities according to their relative species composition. To identify which stations had a statistically similar biomineralizing phytoplankton community across the GCB a SIMPROF routine (1000 permutations, 5% significance level) was applied to the Bray-Curtis similarity matrix.
SIMPROF identifies, based on pairwise tests of the calculated Bray-Curtis percentage similarity, whether the similarities between samples are smaller and/or larger than those expected by chance, grouping those which are statistically distinct (Clarke et al., 2008). The phytoplankton species driving the differences between the groups were identified through a SIMPER routine and presented using non-metric multidimensional scaling (nMDS; Clarke, 1993; Clarke and Warwick, 2001; Clarke and Gorley, 2006). SIMPER allows statistical identification of which species are primarily responsible for
differences between groups of samples and breaks down the Bray-Curtis similarity into individual species contributions.

A BEST routine was applied to environmental and plankton data to determine the combination of environmental variables that 'best' described the variability in coccolithophores and diatoms across the GCB. The BEST routine searches statistically for relationships between the biotic and abiotic patterns and to identify which environmental variable(s) explained most of the variation in species distribution. Spearman's rank correlations were used to further investigate the relationship between
key environmental variables identified in the BEST routine and selected coccolithophore and diatom species.

## 3 Results

### 3.1 General Oceanography

The GCB cruises crossed various biogeochemical gradients associated with the Antarctic Circumpolar Current (ACC) fronts and currents, with most parameters following a recognizable latitudinal (or zonal) pattern. The position of oceanic fronts referred to in the text relates to those defined in Fig. 1 (see also Balch et al., 2016). Sea-surface temperature decreased southwards from $21^o$ C north of the STF to $1.1^o$ C close to $60^o$ S (Table 1). Calcite saturation state ($\Omega_{calcite}$) decreased from 5.2 north of the subtropical front to 2.6 close to $60^o$ S (Table 1). Macronutrient concentrations generally increased southwards with a distinct divide across the SAF. NOx ranged from below detection limits (<0.1 µM) to as high as 28 µM, with higher concentrations generally south of the Sub-Antarctic Front (>12 µM), and lower concentrations (<7 µM) north of the Sub-Antarctic Front (Table 1). $PO_4$ followed a very similar pattern with concentrations generally greater than 1 µM south of the Sub-Antarctic Front and <0.6 µM to the north. Silicic acid concentrations were divided by the PF, being generally less than 2 µM to the north and up to 78.5 µM to the south (Table 1). $\bar{E}_{MLD}$ was highest on the Patagonian Shelf (~40 mol PAR $m^{-2}$ $d^{-1}$) and generally less than 10 mol PAR $m^{-2}$ $d^{-1}$ south of the Sub-Antarctic Front (Table 1). There was no distinct latitudinal trend in pH or $p$CO$_2$. Surface water pH was generally greater than 8.06, ranging from 8.03 on the Kerguelen plateau to 8.13 in the Sub-Tropical Front south-west of Australia (Table 1). Surface water $p$CO$_2$ ranged from 299 µatm to 444 µatm with both extremes in the vicinity of the Atlantic STF (Table 1). Chl $a$ concentrations were variable across the oceanic gradients, highest on the Patagonian Shelf (2.78 mg $m^{-3}$) and on average less than 1 mg $m^{-3}$ in the South Atlantic compared with less than 0.5 mg $m^{-3}$ in the South Indian Ocean (Table 1).

### 3.2 Coccolithophores and diatoms

The most frequently occurring and abundant size group within the coccolithophore and diatom counts were the nanoplankton (cells 2-20 µm). Large diatom species (cells >20 µm) were found in higher numbers (up to 50 cells $mL^{-1}$) south of the PF. Consideration of community biomass would potentially reduce the dominance of the nanoplankton relative to microplankton in the GCB. However, converting from cell size to biomass is not straightforward for diatoms, as highlighted by Leblanc et al., (2012), and to avoid such issues we consider species abundance only. Total cell abundances were less than 1000 cells $mL^{-1}$ at most stations (Table 2), which are indicative of late summer, non-bloom conditions. In the South Atlantic, the highest abundance of coccolithophores was on the Patagonian Shelf (station GCB1-16; 1,636 cells $mL^{-1}$) and the highest abundance of diatoms was east of the South Sandwich Islands (station GCB1-77; 6,893 cells $mL^{-1}$; Table 2). In the South Indian Ocean, coccolithophore abundance was highest near the Crozet Islands (station GCB2-27; 472 cells $mL^{-1}$) and diatom abundance was highest at the most southerly station (station GCB2-73; 538 cells $mL^{-1}$; Table 2). There were no stations in the South Indian Ocean where coccolithophore and diatom abundances were greater than 1,000 cells $mL^{-1}$ (Fig. 2, Table 2). Additionally, the silicifying chrysophyte *Tetraparma* sp. was particularly abundant east of the South Sandwich Islands

(station GCB1-77), at a cell density of 2000 cells mL$^{-1}$, though they were present in low numbers (< 5 cells mL$^{-1}$) at three more stations in the South Atlantic and absent throughout the rest of the GCB.

Coccolithophores dominated the biomineralizing community at twelve stations in terms of abundance north of the PF (Fig. 2, Table 2). On average coccolithophores contributed approximately 38% to total (coccolithophore and diatom) abundance in the GCB. Coccolithophores were greater than 75% of total abundance at only one station, north of South Georgia (station GCB1-59), and never accounted for 100% of total cell numbers. Twenty-eight species of coccolithophores were identified as intact coccospheres across the GCB. Coccolithophore diversity decreased south towards 60$^o$ S, with the highest coccolithophore diversity (19 species) found in the vicinity of the STF in the eastern part of the South Indian Ocean (station GCB2-106), while coccolithophore abundance was more evenly distributed between the different species in the lower latitudes (i.e., high J'; Table 2). *Emiliania huxleyi* was the most numerically abundant coccolithophore at all but four stations and encountered in the mixed layer at all stations except one (station GCB2-73, the most southerly station in the Indian Ocean). Other coccolithophore species (e.g., *Syracosphaera* sp. and *Umbellosphaera* sp.) were present north of the PF throughout the GCB and were most abundant north of the STF. At stations south of the SAF (50$^o$ S) only one (*E. huxleyi*) or two species (*E. huxleyi* and *Pappamonas* sp.) were observed as intact coccospheres.

Diatoms dominated 15 stations in terms of biomineralizing plankton abundance across all environments sampled (Fig. 2, Table 2), being found in every sample analysed and contributing 62% (on average) to total (coccolithophores + diatoms) abundance. Diatoms made up 100% of the total cell counts at the most southerly station in the South Indian Ocean (station GCB2-73) and 99.7% east of the South Sandwich Islands (station GCB1-77; Fig. 2). Seventy-six species of diatom were identified as intact cells across the entire GCB. The most frequently occurring species in the GCB were small (< 5 μm in length) *Fragilariopsis* spp.. The highest abundance of diatoms in the South Atlantic Ocean (6,893 cells mL$^{-1}$) was dominated by *F. nana* east of the South Sandwich Islands (station GCB1-77). The highest diatom abundance in the South Indian Ocean (538 cells mL$^{-1}$) was dominated by *F. pseudonana* at the most southerly station (station GCB2-73) sampled. Another frequently dominant diatom was *Pseudo-nitzschia* sp., which was most abundant north of the PF (Table 2).

Diatom species richness increased south towards 60$^o$ S with the contribution of the different diatom species to total biomineralizing plankton abundance fairly even (J' > 0.5, Table 2), except at stations (stations GCB1-70, GCB1-77, GCB2-27 and GCB2-63) where *Fragilariopsis* spp. <5 μm were dominant (>70% of the diatom population, J' < 0.5). The highest diatom species richness (32 species) was found in the GCB south of the SAF (station GCB2-36) at a temperature of 8$^o$C, in HNLSiLC conditions (NOx 18.9 μM, silicic acid 1.7 μM, 0.21 mg Chl *a* m$^{-3}$).

**3.3 Statistical Analysis**

Three of the environmental variables were removed from the statistical analysis following a Spearman's rank ($r_s$) correlation analysis (Table S1). NOx and PO$_4$ had a strong significant positive correlation ($r_s = 0.961$, $p < 0.0001$) and so NOx was

deemed representative of the distribution of both nutrients. Sea-surface temperature displayed significant negative correlations with both $C_T$ ($r_s$ = -0.981, p < 0.0001) and $A_T$ ($r_s$ = -0.953, p < 0.0001), and so sea surface temperature was taken as being representative of these two variables of the carbonate chemistry system.

The variation in environmental variables across the GCB was examined using a Principal Component Analysis (PCA), which simplifies environmental variability by combining closely correlated variables into principal components in order to account for the greatest variance in the data with the fewest components. The first principal component (PC1) accounted for 58% of the variation in environmental variables, with an additional 17% of environmental variation described by PC2 (Table 3). PC1 describes the main latitudinal gradients of environmental changes across the GCB (decreasing SST, increasing macronutrients). PC1 is a predominantly linear combination of SST, salinity, NOx, silicic acid, $NH_4$, and $\Omega_{calcite}$, where there is a significant positive correlation of PC1 with SST and salinity and a significant negative correlation with all other variables (Table 3). PC2 represented the environmental variation in the GCB occurring independently of latitude, and was driven predominantly by variation in $pCO_2$, with weaker influences from $\bar{E}_{MLD}$ and pH (Table 3). PC2 had significant positive correlations with $pCO_2$ and $\bar{E}_{MLD}$ and a negative correlation with pH.

The SIMPROF routine identified the stations in the GCB that had statistically similar coccolithophore and diatom community composition through a comparison of Bray-Curtis similarities. Six statistically significant groups (*p*< 0.05) were defined across the GCB (Fig. 3). Three groups of these groupings (A, B, C) were specific to the South Atlantic Ocean (Fig. 3). For example, groups A and B represented individual stations GCB1-46 and GCB1-117 respectively, in the sub-tropical region of the South Atlantic Ocean. The most southerly stations in the South Atlantic Ocean (stations GCB1-70 and GCB1-77) defined group C (Fig. 3). Groups D, E and F included stations across the GCB in both ocean regions. Here, group D was defined by eight stations sampled predominantly north of the SAF, while group F was defined by 11 stations predominantly sampled south of the SAF (Fig. 3). These statistically defined similar community structures indicate that although the GCB covers a wide expanse of ocean, the community structure is consistently latitudinal defined across its longitudinal range.

A SIMPER routine statistically identified the species that define the difference between (and similarity within) the statistically different community structures defined by the SIMPROF routine (Table 4). The abundance and distribution of four phytoplankton species (*E. huxleyi*, *Psuedo-nitzschia* sp., *F. nana* and *F. pseudonana*; Fig. 4), were identified as having the most significant contribution to differences in community structure across the GCB (Table 4). *Emiliania huxleyi* and *F. pseudonana* were the most numerically dominant coccolithophore and diatom species, respectively, across the GCB (Table 2). *Fragilariopsis pseudonana* was the numerically dominant diatom (> 30%) at seven stations in the South Indian Ocean (Table 2). The diatom with the highest abundance, *F. nana* (6,797 cells mL$^{-1}$), was almost exclusively found in the South Atlantic Ocean (Table 2) and the more frequently occurring *Pseudo-nitzschia* sp. was present at all but one station.

The non-metric Multi Dimensional Scaling (nMDS) plot of the Bray-Curtis similarities (Fig. 5) shows the station distribution with respect to the SIMPROF defined groups (Fig. 5a), the four main species (Fig. 5b-e) and also holococcolithophores (Fig. 5f). The more closely clustered the stations, the more similar their biomineralizing species composition. Groups A and B were defined by the absence of *E. huxleyi* (Fig. 5b) and the presence of either holococcolithophores (group A; Fig. 5f) or the diatom *Cylindrotheca* sp. (group B). Group C was defined by the dominance of *F. nana* (Table 4; Fig. 5d) and low contributions from *E. huxleyi* and *Pseudo-nitzschia* sp. (Table 2; Fig. 5b,e), resulting in a significant difference from the other groups. Group D had high total species diversity overall (19-41 species; Table 2) and was defined by similar relative abundances of *E. huxleyi* and *Pseudo-nitzschia* sp., which were not found elsewhere (Fig. 5b,e). Group E, composed of stations north of the SAF (Fig. 3, Fig. 5a), included *E. huxleyi*, *U. tenuis* and holococcolithophores (Table 4, Fig. 5b,f). The low abundance and diversity (3-125 cells mL$^{-1}$, 7-11 species; Table 2) of diatoms within group E separated it from the other groups. The combination of *E. huxleyi*, *F. pseudonana* and *Pseudo-nitzschia* sp. that defined group F (Table 4, Fig. 5b,c,e) represented stations on the Patagonian Shelf and south of the SAF (Fig. 3, Fig. 5a). The almost mono-specific *E. huxleyi* coccolithophore community (Table 2) in group F highlights its strong dissimilarity from the other community structure groups identified (Fig. 5).

The influence of environmental variables on the biogeography of coccolithophores and diatoms in the GCB was assessed using the BEST routine. The strongest Spearman's rank correlation ($r_s = 0.55$, $p < 0.001$) between all possible environmental variables and the biogeographical patterns observed came from a combination of five variables, including: (1) SST; (2-4) macronutrients (NOx, silicic acid, NH$_4$); and (5) $p$CO$_2$. This was followed by a correlation of $r_s = 0.54$ (p < 0.001) that included these parameters as well as $\Omega_{calcite}$. Salinity was included in the third highest correlation, whereas $\bar{E}_{MLD}$ and pH did not rank as significant factors in the BEST analysis.

## 4 Discussion

### 4.1 Biogeography of coccolithophores and diatoms in the Great Calcite Belt

Studies of Southern Ocean phytoplankton productivity have generally focused on the micro-phytoplankton (Barber and Hiscock, 2006) as these species contribute around 40% to total oceanic primary production (Sarthou et al., 2005; Uitz et al., 2010). However, nanoplankton and picoplankton are becoming increasingly recognised as important contributors to total phytoplankton biomass, productivity and export in the Southern Ocean (e.g., Boyd, 2002; Uitz et al., 2010; Hinz et al., 2012), both as the dominant size group in post-bloom (Le Moigne et al., 2013) and non-bloom conditions (Barber and Hiscock, 2006).

In this study, coccolithophores were generally numerically dominant at stations sampled north of the PF, particularly around the Sub-Antarctic Front, whereas diatoms were dominant at stations south of the PF (Fig. 2). There was also a significantly

different species distribution (*a priori* ANOSIM; R = 0.227, *p* < 0.01) north and south of the Sub-Antarctic Front, which has been previously identified as the divider between calcite and opal dominated export in the Southern Ocean (e.g., Honjo et al., 2000; Balch et al., 2016). Diatoms were more abundant (~570 cells mL$^{-1}$) than coccolithophores (~160 cells mL$^{-1}$) on average in the entire GCB. This contrasts to Eynaud et al., (1999) in the South Atlantic Ocean at a similar time of year who

reported a peak in coccolithophore cell abundance in the vicinity of the PF (a feature that was not observed in this study). These differences are likely due to variability of Southern Ocean plankton on short temporal scales (Mohan et al., 2008), including variability in the seasonal progression of the spring bloom (Bathmann et al., 1997).

The coccolithophore *E. huxleyi* and diatoms *F. pseudonana*, *F. nana* and *Pseudo-nitzschia* sp. (Fig. 4) were all identified as

being central to defining the statistical similarities within, and the differences between, the different biomineralizing phytoplankton groups (Table 4, Fig. 5). Three of these species *(E. huxleyi, F. nana* and *F. pseudonana)* are part of the nanoplankton, whilst *Pseudo-nitzschia* sp. is at the lower end of the size range of the microplankton *(Pseudo-nitzschia* sp. *is* > 20 μm in length but < 5 μm in width) and contributes significantly to biomass in Southern Ocean HNLC regions (Boyd, 2002). *Emiliania huxleyi* and *Fragilariopsis* sp. smaller than 10 μm have been identified as two of the most abundant

biomineralizing phytoplankton further south in the Scotia Sea (Hinz et al., 2012). Our results further highlight that nanoplankton have the potential to contribute a significant proportion to GCB community composition alongside larger phytoplankton (including large diatoms) typical of HNLC regions.

Abundance of HNLC diatoms such as *F. kerguelensis* (<10 cells mL$^{-1}$), *T. nitzschioides* (<20 cells mL$^{-1}$) and large *Chaetoceros* sp. (<10 cells mL$^{-1}$) were lower than those observed in other studies (e.g., Poulton et al., 2007; Armand et al.,

2008; Korb et al., 2010, 2012). Furthermore, the absence of the diatom *Eucampia antarctica* (<1 cell mL$^{-1}$) in this study does not reflect the typical assemblage (sometimes > 600 cells mL$^{-1}$) found in previous studies (e.g., Kopczyaska et al., 1998; Eynaud et al., 1999; de Baar et al., 2005; Poulton et al., 2007; Salter et al., 2007; Korb et al., 2010). Low abundances of the large-celled diatoms in the silicic acid replete regions may partly relate to the small filter area analyzed using SEM; in this study the area imaged equates to a relatively small volume of water (2-6 mL depending on magnification) relative to the

larger volumes (10-50 mL) often examined for light microscopy in other studies. Large, rare cells may not be enumerated from such small sample volumes, however the numerically abundant nanoplankton groups were well represented in SEM images. Conversely, samples preserved in acidic Lugol's solution for light microscopy analysis are biased towards larger species since small diatoms (<10 μm) are not clearly visible and coccolithophores are not well preserved (Hinz et al., 2012). In future a combination of both imaging techniques is recommended to fully express the phytoplankton community structure

of the Southern Ocean.

## 4.2 *Emiliania huxleyi* in the Great Calcite Belt

The importance of coccolithophores in the GCB was examined via species composition and abundance of intact cells, focusing on areas identified as having high PIC reflectance from underway sampling and satellite observations (Balch et al., 2014, 2016; Hopkins et al., 2015). Higher species diversity of coccolithophores occurred north of the STF (i.e., 6-19 species; Table 2). Coccolithophores are diverse in the stratified and low-nutrient waters associated with lower latitudes (Winter et al., 1994; Poulton et al., 2017). Only a few species are found in the colder waters south of the STF (Mohan et al., 2008), the most successful being *E. huxleyi*, which was observed at an abundance of 103 cells mL$^{-1}$ at 1$^\textbf{o}$C in this study in the South Atlantic (station GCB1-70). The 2$^\text{o}$C isotherm has been previously assumed to represent the southern boundary of *E. huxleyi* (e.g., Verbeek, 1989; Mohan et al., 2008) and inter-annual variability could be influenced by movement of the southern front of the Antarctic Circumpolar Current (Holligan et al., 2010). The Southern Ocean *E. huxleyi* morphotype (Cook et al., 2011; Poulton et al., 2011) may therefore have a wider temperature tolerance than its northern hemisphere equivalent (Hinz et al., 2012) and has been observed poleward of 60$^\text{o}$ S further east in the Southern Ocean (Cubillos et al., 2007) and across Drake Passage (Charalampopoulou et al., 2016). There were three distinct *E. huxleyi* occurrences (the Patagonian Shelf, north of South Georgia and north of the Crozet Islands) within the GCB where *E. huxleyi* contributed > 50% of the total cell counts of biomineralizing phytoplankton. *Emiliania huxleyi* was most abundant (1,636 cells mL$^{-1}$) on the Patagonian Shelf and was the most frequently occurring coccolithophore across the entire GCB. The main *E. huxleyi* occurrences are discussed further below to examine why this species is so widely distributed in the GCB.

### 4.2.1 Patagonian Shelf

The Patagonian Shelf is a well-known region for *E. huxleyi* blooms, as observed in satellite imagery between November and January (Signorini et al., 2006; Painter et al., 2010; Balch et al., 2011; Garcia et al., 2011; Balch et al., 2014). The *E. huxleyi* cell abundance observed in this study (~1,600 cells mL$^{-1}$) was similar to that found by Poulton et al., (2013; >1,000 cells mL$^{-1}$). Using a value of 0.2 pg Chl *a* per cell (Haxo, 1985), following the approach in Poulton et al., (2013), such *E. huxleyi* abundance levels are equivalent to estimated contributions of only ~12% to the total Chl *a* signal (~2.8 mg m$^{-3}$). This estimate is similar to that estimated in an identical way by Poulton et al., (2013) and highlights the significant contribution of phytoplankton other than coccolithophores (flagellates, diatoms) to phytoplankton biomass and production during coccolithophore blooms. It should be noted that the cell Chl *a* content from Haxo (1985) falls at the lower end of the current range of measurements for *E. huxleyi* cell Chl *a* content (e.g., 0.24-0.38 pg Chl *a* per cell; Daniels et al., 2014) and leads to conservative estimates of Chl *a* contribution from this species. This data, combined with satellite observations, supports the hypothesis of a repeating phytoplankton structure on an inter-annual basis, although the contribution of *E. huxleyi* to primary production may vary. The optimum range for *E. huxleyi* blooms on the Patagonian Shelf has been identified as between 5-15$^\text{o}$C at depleted silicic acid levels relative to nitrate (Balch et al., 2014; 2016). During this study, silicic acid was at almost undetectable levels on the Patagonian Shelf (Table 1), with the source water for this region being Southern Ocean HNLSiLC

waters transported northwards via the Falklands current (Painter et al., 2010; Poulton et al., 2013). The persistent low silicic acid availability and residual nitrate (defined as $[NO_3^-] - [Si(OH)_4]$) on the Patagonian Shelf is therefore an ideal environment for *E. huxleyi* to outgrow large, fast growing diatoms (Balch et al., 2014).

### 4.2.2 South Georgia

South Georgia is renowned for intense diatom blooms of over 600 cells $mL^{-1}$ with Chl *a* over 10 mg $m^{-3}$ and integrated primary production up to 2 g C $m^{-2} d^{-1}$ (Korb et al., 2008). However, *E. huxleyi* was the dominant species (>75% of total cell numbers) within the diatom and coccolithophore population at the station north of South Georgia (Table 2, Fig. 2). The associated calcite feature can also be identified from the satellite composite in Fig. 1 (38$^o$ E, 51$^o$ S). *Emiliania huxleyi* contributed approximately 15%, applying a value of 0.2 pg Chl *a* per cell (Haxo, 1985) following Poulton et al., (2013), to

the total Chl *a* signal (0.71 mg $m^{-3}$) around South Georgia. The high calcite feature at South Georgia was found at a SST of 5.9$^o$C, which is below the considered 'optimum' growth conditions for *E. huxleyi* previously cultured (Paasche, 2001). This population of *E. huxleyi* was most likely an adapted cold water morphotype (Cook et al., 2011; Poulton et al., 2011; Cook et al., 2013). The dominant diatom species here was *Actinocyclus* sp. and highly silicified *Thalassionema nitzschioides* with silicic acid concentrations likely limiting (1.7 µmol Si $L^{-1}$; Paasche 1973a & b), whereas NOx concentrations (17.5 µmol N

$L^{-1}$) and PO$_4$ concentrations (1.22 µmol P $L^{-1}$) can be considered replete. The low silicate concentrations could explain why *Eucampia antarctica* was not observed in this study, though it has been observed north of South Georgia (Korb et al., 2010, 2012). This indicates that preceding diatom growth depleted silicic acid (and other nutrients such as dissolved iron), allowing *E. huxleyi* to become more dominant in the population with a similar residual nitrate environment as found on the Patagonian Shelf (this study, Balch et al., 2014; Balch et al., 2016) and also in the North Atlantic (Leblanc et al., 2009).

### 4.2.3 Crozet Islands

The *E. huxleyi* feature north of the Crozet Islands with an abundance of 472 cells $mL^{-1}$ (highest in the South Indian Ocean) confirms the presence of coccolithophores in this region. Coccolithophore abundances have not previously been reported in this region, although elevated PIC had been observed and attributed to *E. huxleyi* (Read et al., 2007; Salter et al., 2007). Chl *a* was lowest (0.47 mg $m^{-3}$) at Crozet out of all three high PIC features, with *E. huxleyi* contributing ~20% of this signal,

applying a value of 0.2 pg Chl *a* per cell (Haxo, 1985) following Poulton et al., (2013), proportionally higher than on the Patagonian Shelf and near South Georgia. Previous studies around the Crozet Islands and plateau (2004-2005) have found evidence of coccolithophores in sediment trap samples (Salter et al., 2007) and large (>30 mmol C $m^{-2} d^{-1}$) calcite fluxes (Le Moigne et al., 2012), though surface cell counts were unavailable (Read et al., 2007). The satellite-derived calcite signal was observed to increase after the main Chl *a* event in this study (Fig. S1) and in previous years (Salter et al., 2007). An increase

in coccolithophore abundance following a diatom bloom is also observed in similar oceanic regions from satellite-derived products (Hopkins et al., 2015) and is associated with depletion of dissolved iron and/or silicic acid (Holligan et al., 2010) in addition to a stable water column and increased irradiance (Balch et al., 2014).

#### 4.2.4 Summary of biogeochemical characterization of coccolithophore occurrence and abundance

The Southern Ocean has been considered to have a biomineralizing phytoplankton community dominated by diatoms. This study highlights that *E. huxleyi* can form distinct features within the GCB and contribute up to 20% towards total Chl *a* in these features compared to an average of less than 5% of Chl *a* across the rest of the GCB. Hence, *Emiliania huxleyi* is likely to have a more important role in biogeochemical processes in the GCB than previously thought. This is particularly important to consider when assessing the impact on calcium carbonate associated export (e.g., Honjo et al., 2000; Balch et al., 2010; Balch et al., 2016) in the Southern Ocean. If *E. huxleyi* is migrating poleward with time (Winter et al., 2013) then the dynamics of the carbon system in the GCB may change, particularly south of the SAF, where silicic acid derived export has historically been dominant (Honjo et al., 2000; Pondaven et al., 2000). Thus, it is essential to gain an understanding of the environmental factors driving the distribution of *E. huxleyi* (Winter et al., 2013, Charalampopoulou et al., 2016) amongst other phytoplankton in the GCB to better understand biogeochemistry of the Southern Ocean.

### 4.3 Environmental controls on biogeography

The environmental variables that best describe coccolithophore and diatom species distribution in this study were SST, macronutrients (NOx, silicic acid, $NH_4$) and $pCO_2$ (Spearman's rank correlation = 0.55, $p < 0.001$), with the second highest correlation (Spearman's rank correlation = 0.54, $p < 0.001$) including calcite saturation state ($\Omega_{calcite}$). The inclusion of $pCO_2$ and $\Omega_{calcite}$ as important factors indicates a potential influence of carbonate chemistry on coccolithophore and diatom distribution (and *vice versa*) in the GCB. However, $\Omega_{calcite}$ had a very strong positive correlation (r = 0.964, $p < 0.0001$) with SST (Table S1), and therefore separating the influences of the two variables was impossible in this study due to the tight coupling between carbonate chemistry and temperature (as also observed by Charalampopoulou et al., 2016).

#### 4.3.1 Temperature

Temperature is recognized as a strong driving factor behind plankton biogeography and community composition (Raven and Geider, 1988; Boyd et al., 2010). The abundance of two of the dominant species, *E. huxleyi* and *F. pseudonana*, did not significantly correlate (Pearson's product moment correlation = 0.147, $p = 0.493$ and r = -0.247, $p = 0.357$ respectively) with SST, which does not agree with previous work (e.g., Mohan et al., 2008) and implies that *E. huxleyi* distribution is not solely determined by latitudinal variations in temperature. Nanoplankton are subject to high grazing pressure (Schmoker et al., 2013), with the growth and mortality of a species both directly influencing cell abundances (Poulton et al., 2010), which could result in nanoplankton patchiness additional to the influence of temperature and/or other environmental gradients. In contrast, the negative correlation of *F. nana* (Pearson's product moment correlation = -0.976, $p < 0.05$, n = 4) versus the positive correlation of *Pseudo-nitzschia* sp. (Pearson's product moment correlation = 0.544, $p < 0.05$, n = 19) with SST indicates that these two species have distinctly different physiological tolerances. Southern Ocean diatoms are often observed to have negative relationships with temperature (e.g., Eynaud et al., 1999; Boyd, 2002). *Pseudo-nitzschia* sp. was

predominantly found in waters north of the PF in this study, as seen by Kopczynska et al., (1986), and is likely to be out competed by other diatom species (e.g., *Chaetoceros* sp. and *Dactyliosolen* sp.) further south due to different nutrient affinities and requirements (Kopczynska et al., 1986), particularly for dissolved iron and silicic acid.

### 4.3.2 Nutrients

Macronutrient gradients, particularly silicic acid, are considered one of the key driving factors between the differences in community structure in the Southern Ocean (Nelson and Treguer, 1992). NOx (and $PO_4$ by association) was identified in the BEST test as an important factor in the variability of biomineralizing species distribution, but did not significantly correlate with the four statistically dominant species (Fig. 4) contributing over 50% to changes in species composition in the GCB.

Nitrate drawdown by Southern Ocean diatoms is limited by dissolved iron (dFe) availability south of the STF (Sedwick et

al., 2002), which may explain the dominance of the nanoplankton (with lower dFe and macronutrient requirements; Ho et al., 2003) in this study as they are not affected by low dFe concentrations as severely as the microplankton. The low silicic acid concentrations in the region between the SAF and the PF indicate that there was sufficient dFe to allow silicification and diatom growth, but either one or both of the macronutrients were then depleted to limiting concentrations (Assmy et al., 2013). As an essential nutrient for diatoms, silicic acid concentrations less than 2 μM were most common in the GCB, a

level which is considered limiting for most diatom species (Paasche, 1973a & b; Egge and Asknes, 1992). However, even at stations with greater than 5 μM silicic acid, the small diatom species (<10 μm) were still dominant and represented over 40% of the total coccolithophore and diatom assemblage (numerically). A significant positive correlation occurred between silicic acid and the small (<5 μm) diatom *F. nana* (Pearson's product moment correlation = 0.986, $p < 0.05$, n = 4). *Fragilariopsis nana* may have a low cellular silicate requirement similar to *F. pseudonana* (Poulton et al., 2013) relative to larger diatom

species, so the high abundance of *F. nana* in the high silicic acid waters could be indicative of a seasonal progression driven by light and/or temperature rather than silicic acid dependence. *Fragilariopsis* sp. have been observed at high abundances near the Ross Sea ice shelf (Grigorov and Rigual-Hernandez, 2014) and high abundances of large diatoms in silicic acid- (and dFe-) replete waters may occur further south than we sampled. In the South Atlantic and the South Pacific Ocean, silicic acid depletion moves southwards as spring to summer progresses, with maximum diatom biomass observed in late January at

65°S (Sigmon et al., 2002; Le Moigne et al., 2013).

A significant negative correlation between *E. huxleyi* and silicic acid (Pearson's product moment correlation = -0.410, $p <$ 0.05, n = 24) in this study has also been identified in the Scotia Sea (Hinz et al., 2012) and Patagonian Shelf (Balch et al., 2014) in the Southern Ocean, as well as in the North Atlantic (Leblanc et al., 2009). Low silicic acid may be considered a positive selection pressure for coccolithophores (Holligan et al., 2010), especially when other macronutrients (and dFe) are

replete. However, a few non-blooming coccolithophore species are now recognized as having silicic acid requirements, though this requirement is absent in *E. huxleyi* (Durak et al., 2016). Therefore, low silicic acid in surface waters of the GCB

may negatively impact coccolithophore species that have a silicic acid requirement, such as *Calcidiscus leptoporus*, and favour bloom-forming species that have no silicic acid requirement (e.g., *E. huxleyi*). To the south of the PF, silicic acid increased (from < 1 to > 3 µM) with five stations between the SAF and PF (and one south of the PF, station GCB1-59), all numerically dominated by *E. huxleyi*, while other stations to the south of the PF were dominated by diatoms (Fig. 2).

These results from the GCB indicate a progression of biomineralizing phytoplankton southwards during spring as irradiance conditions become optimal and macronutrients are depleted. Low silicic acid is often associated with a high residual nitrate concentrations (defined as $[NO_3^-]$ - $[Si(OH)_4]$), as has been observed on the Patagonian Shelf (Balch et al., 2014). The highest coccolithophore abundances in this study (excluding the Patagonian Shelf) were observed in regions with 'residual nitrate' concentrations greater than 10 µM (Balch et al., 2016). As silicic acid becomes depleted in the more northerly

surface waters in spring, diatoms progressively become more successful further south as irradiance conditions allow, thereby producing a large HNLSiLC area between the Sub-Antarctic Front and Polar Front; an ideal environment for late summer *E. huxleyi* communities to develop (Fig. 6).

    Dissolved iron (dFe) acts as a strong control on phytoplankton growth, community composition and species biogeography (e.g., Boyd, 2002; Boyd et al., 2015). In this study, dFe measurements were only made at a small number of sampling

stations (n = 6; Twining, unpublished data, Balch et al., 2016) limiting their use in the multivariate statistical analysis of community composition. For these stations, dFe showed a statistically significant negative correlation (Pearson's product moment = -0.957, $p < 0.01$) with PC2 from the environmental analysis (Fig. S2). PC2 described the environmental variables least related to latitude (pH, $p$CO$_2$ and $\bar{E}_{MLD}$), indicating that dFe was also decoupled from the strong latitudinal gradient in environmental parameters (i.e. SST, $\Omega_{calcite}$, macronutrients) in austral spring/summer. Interestingly, dFe concentrations did

positively correlate with coccolithophore abundance (Pearson's product moment correlation = 0.858, $p < 0.05$) rather than diatom abundance ($p = 0.132$, ns) (Fig. S2). Overall, these data support the hypothesis that coccolithophores occupy a niche unoccupied by large diatoms when dFe is replete and silicic acid is depleted (Balch et al., 2014; Hopkins et al., 2015). The numerical dominance of small diatoms less than 20 µm in the GCB during austral spring and summer, alongside the coccolithophore *E. huxleyi*, is thus potentially due to the reduced impact of nutrient limitation (dFe, silicic acid) on small

cells with high ratios of surface area to volume (e.g., Hinz et al., 2012; Balch et al., 2014).

### 4.4 Relating the Great Calcite Belt to carbonate chemistry

Relating carbonate chemistry to phytoplankton distribution, growth and physiology is an important step when considering the potential effects of climate change and ocean acidification on marine biogeochemistry. In this study, no significant correlation (Spearman's r = 0.259, $p = 0.164$, n = 27) occurred between pH and Chl *a*. The inclusion of $p$CO$_2$ and $\Omega_{calcite}$ as

influential factors in the statistical results describing GCB species biogeography highlights the importance of understanding phytoplankton responses to carbonate chemistry as a whole rather than as individual carbonate chemistry parameters (Bach

et al., 2015). Of the four major species driving the differences in biomineralizing plankton community composition and biogeography across the GCB, only *F. pseudonana* abundance was positively correlated with $pCO_2$ (Pearson's product moment coefficient = 0.577, $p < 0.05$, n = 16).

The response of diatoms to increasing $pCO_2$ is not straight forward (e.g., Boyd et al., 2015), with some studies implying that large diatoms may be more successful in future climate scenarios (e.g., Tortell et al., 2008; Flynn et al., 2012), although changes in nutrient and light availability (via stronger stratification) may prevent a permanent switch in phytoplankton community structure (Bopp, 2005). The carbonate chemistry system is complex as biological activity also impacts the concentration of each of the components. Organic matter production reduces total dissolved inorganic carbon ($C_T$) and hence $pCO_2$ via photosynthesis, as well as increasing alkalinity ($A_T$) through nutrient uptake, while subsequent respiration and remineralisation of organic matter has the opposite impact. The simultaneous actions of biological and physical processes result in seasonal and localized changes in the carbonate system, which are often difficult to decouple.

In our study, there was no significant correlation between *E. huxleyi* and $\Omega_{calcite}$ (Pearson's product moment = 0.093). However, the waters of the GCB remained oversaturated ($\Omega_{calcite} > 2$) throughout, and furthermore the relationship between coccolithophores, calcification and carbonate chemistry is now recognized as being complex and non-linear (e.g., Beaufort et al., 2011; Smith et al., 2012; Poulton et al., 2014; Rivero-Calle et al., 2015; Bach et al., 2015; Charalampopoulou et al., 2016; Marañón et al., 2016). Hence, significant gaps remain in our understanding of the *in situ* coccolithophore response to increasing $pCO_2$, reduced pH or decreasing $\Omega_{calcite}$. Notably, a significant positive correlation between *Pseudo-nitzschia* sp. and $\Omega_{calcite}$ also existed (Pearson's product moment correlation = 0.5924, $p < 0.01$, n = 19) across the GCB despite there being presently no known detrimental effect on diatoms of low saturation states. However, due to the tight coupling of temperature and $\Omega_{calcite}$ (and *Pseudo-nitzshia* sp. and temperature) the correlation is more likely to be temperature driven.

## 5 Summary

This study of the GCB further highlights the importance of understanding the environmental controls on the distribution of biomineralizing nanoplankton in the Southern Ocean. The results of this study suggest that three nano- (<20 μm) and one micro- (>20 μm) phytoplankton species (three diatoms and one coccolithophore; *F. pseudonana*, *F. nana*, *Pseudo-nitzschia* sp., and *Emiliania huxleyi*) numerically dominated the compositional variation in biomineralizing phytoplankton biogeography across the GCB. The contribution of *E. huxleyi* to phytoplankton biomass (as estimated from cell counts and Chl *a*) was generally less than 5%, although it increased up to 20% in association with high reflectance PIC features found on the Patagonian Shelf, north of South Georgia in the South Atlantic Ocean, and north of the Crozet Islands in the South Indian Ocean. This indicates that in the post-spring bloom conditions of the GCB, *E. huxleyi* is an important contributor to phytoplankton biomass and primary production at localized spatial scales.

Out of a wide suite of environmental variables, latitudinal gradients in temperature, macro-nutrients, $pCO_2$ and $\Omega_{calcite}$ 'best' described statistically the variation of phytoplankton community composition in this study, whereas $\bar{E}_{MLD}$ and pH did not rank as significant factors influencing species composition. However, not all species were directly sensitive to the same environmental gradients as determined to be influencing the overall biogeography. The negative correlation between *E. huxleyi* and silicic acid highlights the potential for a seasonal southward movement of *E. huxleyi* once diatom blooms have depleted silicic acid.

These results highlight that the Southern Ocean is a highly dynamic system and further studies examining environmental controls on community distribution earlier in the productive season would greatly enhance overall understanding of the progression of phytoplankton community biogeography. The phytoplankton dynamics of the GCB are also more complex than first considered, with nanophytoplankton (e.g., *F. pseudonana*) numerically dominant in non-bloom conditions (as opposed to microphytoplankton), which has further implications for modelling carbon export and projecting phytoplankton changes in future oceanic scenarios.

## Competing interests

The authors declare that they have no conflict of interest.

## Data availability

The coccolithophore and diatom abundance data can be accessed via the PANGAEA database:
https://doi.pangaea.de/10.1594/PANGAEA.879790

## Acknowledgements

We would like to thank all the scientists, officers and crew on board the R/V Melville and R/V Revelle, and Matt Durham (Scripps Institution of Oceanography) in particular. Nutrient and salinity data are presented courtesy of the Oceanographic Data Facility, Scripps Institute of Oceanography (ODF/SIO), with thanks to shipboard techs Melissa Miller and John Calderwood. The GCB cruises were supported by the National Science Foundation (OCE-0961660 to WMB and BST, OCE-0728582 to WMB, OCE-0961414 to NRB) and National Aeronautical and Space Administration (NNX11AO72G, NNX11AL93G, NNX14AQ41G, NNX14AQ43A, NNX14AL92G, and NNX14AM77G to WMB). HEKS and AJP were supported by the Natural Environmental Research Council (NERC), including a NERC Fellowship to AJP (NE/F015054/1) and the UK Ocean Acidification Research Program (NE/H017097/1) with a tied studentship to HEKS and an added value award to AJP.

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

**Tables**

Table 1: Details of Great Calcite Belt sampling stations including station and cruise identifier; date of sample collection (DD.MM.YYYY); station position decimal latitude (Lat) and longitude (Long); sea surface temperature (SST); surface
salinity (Sal); mixed layer average irradiance ($\bar{E}_{MLD}$); surface macronutrient concentrations (nitrate and nitrite, NOx; phosphate, $PO_4$; silicate, $Si(OH)_4$; ammonia, $NH_4$) surface carbonate chemistry parameters (normalized total alkalinity, $A_T$; dissolved inorganic carbon, $C_T$; pH; partial pressure of carbon dioxide ($pCO_2$); calcite saturation state ($\Omega_{calcite}$); and surface chlorophyll $a$ (Chl $a$, mg m$^{-3}$). Bold type indicates those used in the statistical analyses.

Table 2: Whole cell abundances of coccolithophores and diatoms in surface samples of the Great Calcite Belt, number of species in each group (S), Pielou's evenness (J', **** denotes that J' was not calculated because only one species was present), the dominant species and its percentage contribution to the total numerical abundance of coccolithophores (%Co) or diatoms (%D). $^+$ denotes where one species had almost total numerical dominance (> 99.8%), with only one or two cells of a separate species enumerated, and was therefore rounded up to 100%. Holococcolithophores are abbreviated as
Holococco*. Position denotes the location relative to the Southern Ocean fronts and zones (Z; north of the defined front) as defined by Orsi et al., (1995), letters after the front abbreviation denote specific locations and proximity to landmasses: Patagonian Shelf (PS); north of South Georgia (n SG); South Sandwich Islands (SS); Crozet Island (Cr), Kerguelen Island (K); Heard Island (H).

Table 3: Principal component (PC) scores, percentage variation described (%V) and the Pearson's product moment correlation associated with each variable and its significance level: **p <0.0001***, p<0.001**, p<0.005*, p< 0.01**, p<0.05.

Table 4: Phytoplankton assemblage groups identified, using the SIMPROF routine at $p < 0.05$, in the GCB (see also Fig. 3), from the South Atlantic (GCB1) and the South Indian (GCB2) Oceans. Location is indicated as in Fig. 2. Group Average
Similarity (Group Av.Sim%) defines the percentage similarity of the community structure in all the stations within each group. The defining species contributing >50% to the species similarity for each group as identified through the SIMPER routine are presented alongside the average similarity for each species in each group (Average Similarity), where higher Similarity SD indicates more consistent contribution to similarity within the group. The percentage contribution per species to the group similarity (Contribution %) was also calculated.

**Table 1**

| Station | Date | Lat | Long | SST | Sal | Ē_MLD | NOx | PO$_4$ | Si(OH)$_4$ | NH$_4$ | $A_T$ | $C_T$ | pH | $p$CO$_2$ | Ω$_{calc}$ |
|---|---|---|---|---|---|---|---|---|---|---|---|---|---|---|---|
| | | $^o$ S | $^o$ E | $^o$ C | | mol PAR m$^{-2}$ d$^{-1}$ | | | μM | | | | | μatm | |
| GCB1-6 | 14.01.2011 | 51.79 | -56.11 | 8.6 | 34.0 | 17.8 | 14.2 | 1.05 | 1.7 | 0.64 | 2336 | 2138 | 8.09 | 367 | 3.3 |
| GCB1-16 | 17.01.2011 | 46.26 | -59.83 | 11.8 | 33.8 | 39.8 | 6.5 | 0.54 | 0.0 | 0.15 | 2333 | 2100 | 8.12 | 407 | 3.8 |
| GCB1-25 | 20.01.2011 | 45.67 | -48.95 | 16.1 | 35.1 | 25.5 | 0.0 | 0.23 | 0.2 | 0.16 | 2320 | 2047 | 8.12 | 390 | 4.6 |
| GCB1-32 | 22.01.2011 | 40.95 | -45.83 | 20.0 | 35.6 | 36.7 | 0.1 | 0.11 | 1.1 | 0.05 | 2307 | 2029 | 8.07 | 444 | 4.8 |
| GCB1-46 | 26.01.2011 | 42.21 | -41.21 | 18.3 | 34.9 | 16.0 | 0.2 | 0.19 | 0.3 | 0.00 | 2328 | 2050 | 8.09 | 356 | 4.7 |
| GCB1-59 | 29.01.2011 | 51.36 | -37.84 | 5.9 | 33.8 | 7.9 | 17.5 | 1.22 | 1.7 | 0.67 | 2368 | 2184 | 8.10 | 325 | 3.1 |
| GCB1-70 | 01.02.2011 | 59.25 | -33.15 | 1.1 | 34.0 | 9.7 | 22.3 | 1.74 | 78.5 | 1.54 | 2388 | 2235 | 8.10 | 407 | 2.6 |
| GCB1-77 | 03.02.2011 | 57.28 | -25.98 | 1.4 | 33.9 | 11.9 | 20.7 | 1.55 | 68.8 | 1.00 | 2386 | 2225 | 8.12 | 405 | 2.7 |
| GCB1-85 | 05.02.2011 | 53.65 | -17.75 | 4.1 | 33.9 | 8.9 | 19.1 | 1.33 | 0.7 | 0.30 | 2369 | 2191 | 8.12 | 363 | 3.0 |
| GCB1-92 | 07.02.2011 | 50.40 | -10.80 | 5.9 | 33.8 | 9.5 | 17.5 | 1.27 | 1.4 | 0.37 | 2362 | 2182 | 8.10 | 351 | 3.0 |
| GCB1-101 | 09.02.2011 | 46.31 | -3.21 | 11.0 | 34.0 | 17.1 | 12.5 | 0.95 | 0.6 | 0.16 | 2345 | 2134 | 8.08 | 400 | 3.5 |
| GCB1-109 | 11.02.2011 | 42.63 | 3.34 | 15.1 | 34.4 | 20.0 | 5.3 | 0.56 | 0.8 | 0.00 | 2332 | 2098 | 8.07 | 359 | 4.0 |
| GCB1-117 | 12.02.2011 | 39.00 | 9.49 | 18.8 | 35.0 | 19.4 | 0.0 | 0.20 | 0.7 | 0.06 | 2321 | 2047 | 8.08 | 299 | 4.7 |
| GCB2-5 | 21.02.2012 | 37.09 | 39.48 | 21.0 | 35.5 | 11.2 | 0.0 | 0.05 | 1.1 | 0.07 | 2310 | 2005 | 8.10 | 340 | 5.2 |
| GCB2-13 | 23.02.2012 | 40.36 | 43.50 | 18.4 | 35.3 | 13.7 | 0.1 | 0.17 | 0.2 | 0.02 | 2307 | 2032 | 8.09 | 351 | 4.7 |
| GCB2-27 | 26.02.2012 | 45.82 | 51.05 | 7.7 | 33.7 | 5.8 | 20.1 | 1.35 | 2.9 | 0.14 | 2344 | 2194 | 8.00 | 425 | 2.6 |
| GCB2-36 | 28.02.2012 | 46.74 | 57.48 | 8.1 | 33.7 | 8.7 | 18.9 | 1.40 | 1.7 | 0.49 | 2363 | 2175 | 8.08 | 355 | 3.1 |
| GCB2-43 | 01.03.2012 | 47.52 | 64.04 | 6.5 | 33.7 | 5.9 | 21.7 | 1.53 | 0.5 | 0.38 | 2358 | 2197 | 8.04 | 387 | 2.8 |
| GCB2-53 | 02.03.2012 | 49.30 | 71.32 | 5.1 | 33.7 | 8.5 | 23.8 | 1.66 | 7.1 | 0.17 | 2359 | 2210 | 8.03 | 396 | 2.6 |
| GCB2-63 | 04.03.2012 | 54.40 | 74.56 | 3.5 | 33.8 | 3.0 | 25.3 | 1.70 | 10.5 | 0.21 | 2363 | 2210 | 8.07 | 360 | 2.6 |
| GCB2-73 | 06.03.2012 | 59.71 | 77.75 | 1.1 | 33.9 | 4.3 | 28.0 | 1.91 | 40.4 | 0.34 | 2372 | 2233 | 8.07 | 360 | 2.4 |
| GCB2-87 | 10.03.2012 | 54.25 | 88.14 | 3.4 | 33.9 | 4.3 | 24.2 | 1.69 | 9.0 | 0.45 | 2367 | 2216 | 8.06 | 367 | 2.6 |
| GCB2-93 | 12.03.2012 | 49.81 | 94.13 | 7.8 | 34.0 | 5.9 | 17.5 | 1.27 | 1.5 | 0.26 | 2345 | 2149 | 8.10 | 333 | 3.3 |
| GCB2-100 | 13.03.2012 | 44.62 | 100.50 | 13.0 | 34.8 | 4.7 | 6.4 | 0.55 | 0.2 | 0.15 | 2328 | 2083 | 8.11 | 326 | 4.1 |
| GCB2-106 | 15.03.2012 | 40.13 | 105.38 | 17.0 | 35.4 | 12.8 | 0.1 | 0.14 | 0.3 | 0.03 | 2318 | 2029 | 8.13 | 313 | 4.9 |
| GCB2-112 | 17.03.2012 | 40.26 | 109.60 | 15.8 | 34.9 | 11.1 | 3.6 | 0.43 | 0.2 | 0.00 | 2323 | 2060 | 8.11 | 332 | 4.4 |
| GCB2-119 | 20.03.2012 | 42.08 | 113.40 | 13.8 | 34.8 | 11.2 | 5.3 | 0.55 | 0.2 | 0.01 | 2320 | 2080 | 8.10 | 342 | 4.1 |

**Table 2**

| Station | Position | Coccolithophores (Co) | | | | | Diatoms (D) | | | | |
|---|---|---|---|---|---|---|---|---|---|---|---|
| | | Cell mL$^{-1}$ | S | J' | Dominant species | % of Co | Cell mL$^{-1}$ | S | J' | Dominant species | % of D |
| GCB1-6 | SAF, PS | 243 | 2 | 0.02 | *E. huxleyi* | 100[+] | 127 | 15 | 0.79 | *C. deblis* | 26 |
| GCB1-16 | SAF, PS | 1636 | 2 | 0.00 | *E. huxleyi* | 100[+] | 4610 | 5 | 0.11 | *F. pseudonana* | 96 |
| GCB1-25 | SAFZ | 55 | 9 | 0.67 | *S. mollischi* | 38 | 28 | 10 | 0.84 | *Pseudo-nitzschia* sp. | 37 |
| GCB1-32 | STF | 23 | 8 | 0.83 | *U. tenuis* | 31 | 19 | 8 | 0.70 | *Nitzschia* sp. | 55 |
| GCB1-46 | STF | 3 | 1 | **** | Holococco* | 100 | 4 | 3 | 0.91 | *Chaetoceros* sp. | 56 |
| GCB1-59 | sPF, n SG | 565 | 1 | **** | *E. huxleyi* | 100 | 183 | 30 | 0.72 | *T. nitzschioides* | 29 |
| GCB1-70 | sPF | 103 | 1 | **** | *E. huxleyi* | 100 | 720 | 24 | 0.29 | *F. nana* | 81 |
| GCB1-77 | sPF, SS | 2 | 1 | **** | *E. huxleyi* | 100 | 6893 | 18 | 0.04 | *F. nana* | 98 |
| GCB1-85 | sPF | 28 | 1 | **** | *E. huxleyi* | 100 | 151 | 30 | 0.77 | *C. aequatorialis* sp. | 22 |
| GCB1-92 | PFZ | 77 | 2 | 0.13 | *E. huxleyi* | 98 | 111 | 28 | 0.73 | *Pseudo-nitzschia* sp. | 32 |
| GCB1-101 | SAFZ | 92 | 7 | 0.57 | *E. huxleyi* | 68 | 52 | 11 | 0.57 | *F. pseudonana* | 59 |
| GCB1-109 | SAFZ | 39 | 9 | 0.90 | *E. huxleyi* | 25 | 129 | 17 | 0.55 | *Pseudo-nitzschia* sp. | 61 |
| GCB1-117 | STF | 15 | 6 | 0.88 | *U. tenuis* | 35 | 209 | 9 | 0.13 | *C. closterium* | 95 |
| GCB2-5 | STFZ | 37 | 15 | 0.69 | *E. huxleyi* | 46 | 6 | 8 | 0.76 | *Nanoneis hasleae* | 47 |
| GCB2-13 | STFZ | 51 | 17 | 0.61 | *E. huxleyi* | 57 | 28 | 7 | 0.57 | *Nitzschia* sp.*<20μm* | 67 |
| GCB2-27 | SAF, Cr | 478 | 6 | 0.04 | *E. huxleyi* | 99 | 375 | 24 | 0.28 | *F. pseudonana* | 83 |
| GCB2-36 | SAF | 166 | 8 | 0.32 | *E. huxleyi* | 83 | 155 | 32 | 0.69 | *F. pseudonana* | 33 |
| GCB2-43 | PFZ | 12 | 4 | 0.18 | *E. huxleyi* | 95 | 90 | 25 | 0.57 | *F. pseudonana* | 54 |
| GCB2-53 | sPF, K | 51 | 3 | 0.90 | *E. huxleyi* | 56 | 512 | 28 | 0.39 | *F. pseudonana* | 47 |
| GCB2-63 | sPF, H | 132 | 1 | **** | *E. huxleyi* | 100 | 254 | 24 | 0.38 | *F. pseudonana* | 71 |
| GCB2-73 | sPF | 0 | 0 | **** | n/a | n/a | 538 | 24 | 0.55 | *F. pseudonana* | 56 |
| GCB2-87 | sPF | 106 | 1 | **** | *E. huxleyi* | 100 | 184 | 29 | 0.55 | *F. pseudonana* | 42 |
| GCB2-93 | PFZ | 100 | 11 | 0.33 | *E. huxleyi* | 80 | 75 | 29 | 0.67 | *Pseudo-nitzschia* sp. | 37 |
| GCB2-100 | SAFZ | 123 | 13 | 0.26 | *E. huxleyi* | 86 | 164 | 26 | 0.44 | *Pseudo-nitzschia* sp. | 67 |
| GCB2-106 | STF | 90 | 19 | 0.77 | *E. huxleyi* | 29 | 80 | 22 | 0.58 | *Pseudo-nitzschia* sp. | 54 |
| GCB2-112 | STF | 123 | 12 | 0.35 | *E. huxleyi* | 80 | 257 | 27 | 0.38 | *Pseudo-nitzschia* sp. | 74 |
| GCB2-119 | SAFZ | 121 | 17 | 0.32 | *E. huxleyi* | 82 | 68 | 21 | 0.55 | *Pseudo-nitzschia* sp. | 47 |

**Table 3**

| Variable | PC1 - EV 5 (58%) | | PC2 - EV 1.5 (17%) | |
|---|---|---|---|---|
| Temp | 0.42 | **(0.97\*\*\*)** | 0.08 | (-0.10) |
| Salinity | 0.36 | **(0.90\*\*\*)** | 0 | - |
| $\bar{E}_{MLD}$ | 0.24 | **(-0.55\*)** | 0.5 | **(0.62\*\*)** |
| NOx | -0.4 | **(-0.91\*\*\*)** | -0.05 | (-0.06) |
| $Si(OH)_4$ | -0.35 | **(-0.77\*\*\*)** | 0.02 | (-0.03) |
| $NH_4$ | -0.35 | **(-0.81\*\*\*)** | -0.07 | (-0.09) |
| pH | 0.18 | **(-0.39)** | -0.42 | **(-0.50\*)** |
| $p\mathrm{CO}_2$ | -0.15 | (-0.33) | 0.75 | **(0.89\*\*\*)** |
| $\Omega_{calcite}$ | 0.43 | **(-0.99\*\*\*)** | -0.02 | (-0.02) |

**Table 4**

| Group | Station | Location | Group Av.Sim% | Defining Species | Average Similarity | Similarity SD | Contribution % |
|---|---|---|---|---|---|---|---|
| **A** | GCB1-46 | STF | n/a | Holococco* | n/a | n/a | n/a |
| **B** | GCB1-117 | | | *Cylindrotheca* sp. | | | |
| **C** | GCB1-70 | SBDY | 54.5 | *F. nana* | 53.3 | n/a | 97.8 |
| | GCB1-77 | | | | | | |
| **D** | GCB1-25 | N of PF | 47.6 | *E. huxleyi* | 13.9 | 2.68 | 29.3 |
| | GCB1-109 | | | *Pseudo-nitzschia* sp. | 12.7 | 3.6 | 26.7 |
| | GCB2-36 | | | | | | |
| | GCB2-93 | | | | | | |
| | GCB2-100 | | | | | | |
| | GCB2-106 | | | | | | |
| | GCB2-112 | | | | | | |
| | GCB2-119 | | | | | | |
| **E** | GCB1-32 | N of SAF | 42.3 | *E. huxleyi* | 18.9 | 3.8 | 44.8 |
| | GCB1-101 | | | Holococco* | 8.45 | 4.01 | 20 |
| | GCB2-5 | | | | | | |
| | GCB2-13 | | | | | | |
| **F** | GCB1-6 | PS | 40.6 | *E. huxleyi* | 15.1 | 1.51 | 37.3 |
| | GCB1-16 | | | *F. pseudonana* | 14.2 | 1.25 | 35 |
| | GCB1-59 | S of SAF | | | | | |
| | GCB1-85 | | | | | | |
| | GCB1-92 | | | | | | |
| | GCB2-27 | | | | | | |
| | GCB2-43 | | | | | | |
| | GCB2-53 | | | | | | |
| | GCB2-63 | | | | | | |
| | GCB2-73 | | | | | | |
| | GCB2-87 | | | | | | |

# Figures

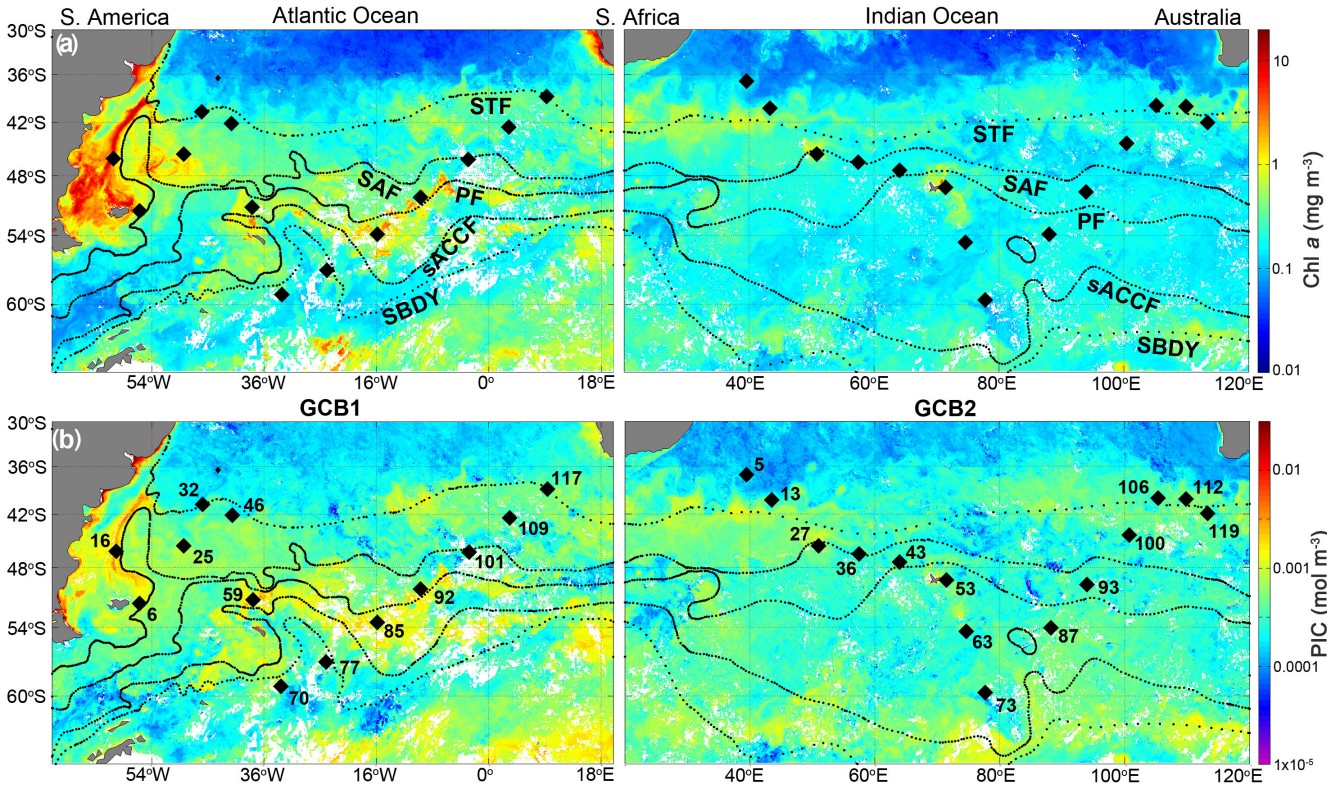

**Figure 1 Rolling 32 day composite from MODIS-Aqua for both (a) Chlorophyll *a* (mg m$^{-3}$) and (b) PIC (mol m$^{-3}$) for the South Atlantic sector (17th January to 17th February 2011) and the South Indian sector (18th February to 20th March 2012). Station number identifiers and averaged positions of fronts as defined by Orsi et al., (1995) are superimposed: Sub-Tropical front (STF), Sub Antarctic front (SAF), Polar Front (PF), Southern Antarctic Circumpolar Current Front (SACCF) and Southern Boundary (SBDY).**

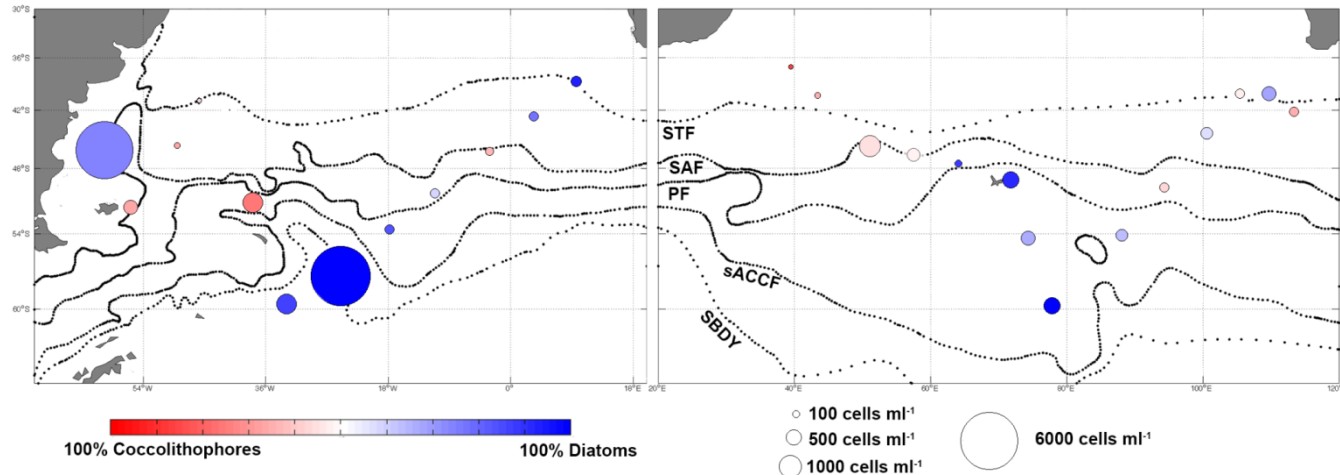

**Figure 2 Coccolithophore and diatom abundance and dominance information. The area of the circles denotes abundance while shading denotes percentage contribution of each phytoplankton group, where red denotes coccolithophore dominance and blue denotes diatom dominance. Fronts are defined as in Fig. 1**

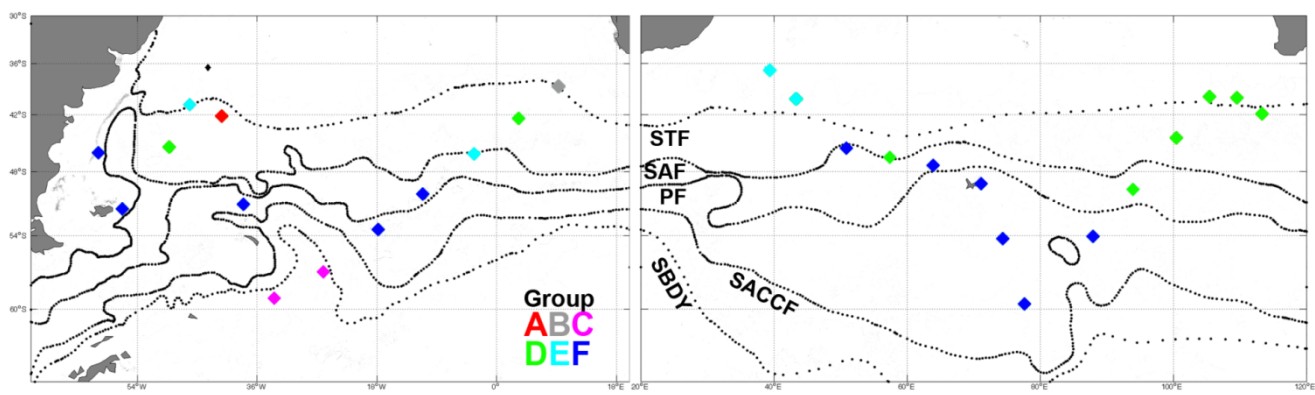

**Figure 3 Statistically significant groups of coccolithophore and diatom communities in the Great Calcite Belt as identified by the SIMPROF routine. The colors designate which statistical group defines the coccolithophore and diatom assemblage at each station as shown in the group key. Fronts are defined as in Fig. 1.  See Table 4 for full group species descriptions.**

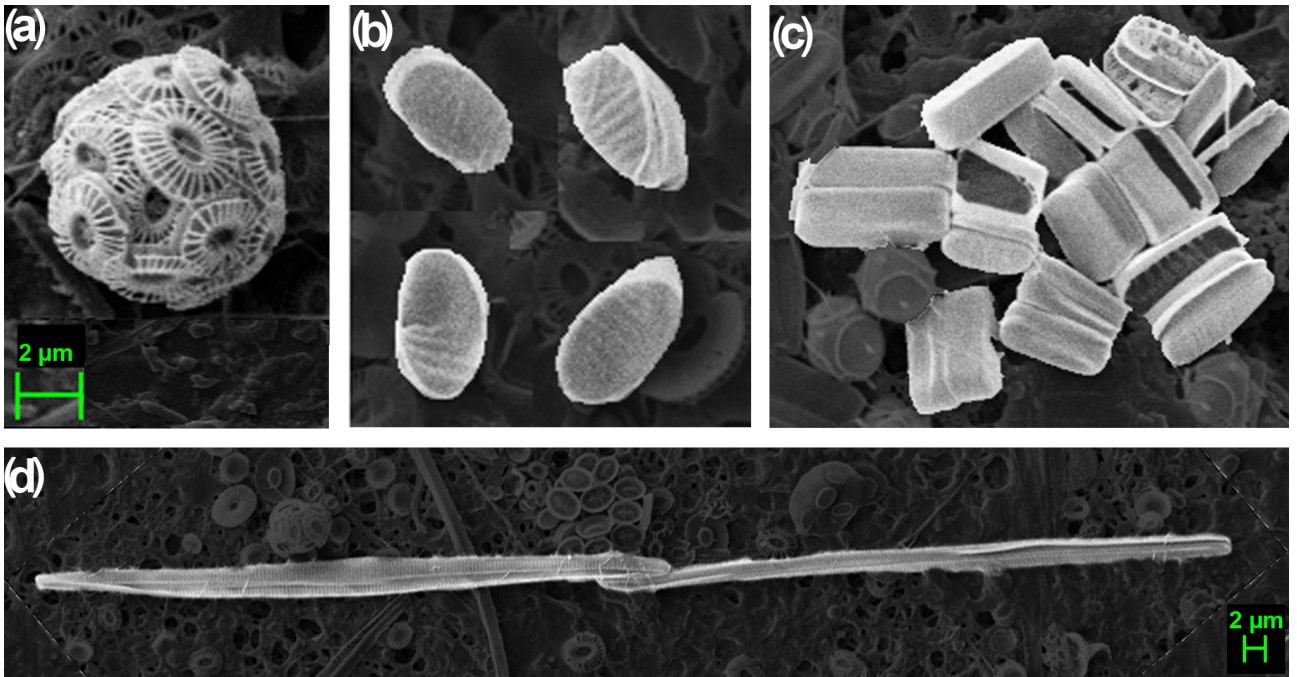

**Figure 4 SEM images of the four phytoplankton species identified by the SIMPER analysis as characterizing the significantly different community structures. (a)** *E. huxleyi*; **(b)** *F. pseudonana*; **(c)** *F. nana*; **and (d):** *Pseudo-nitzschia* sp..

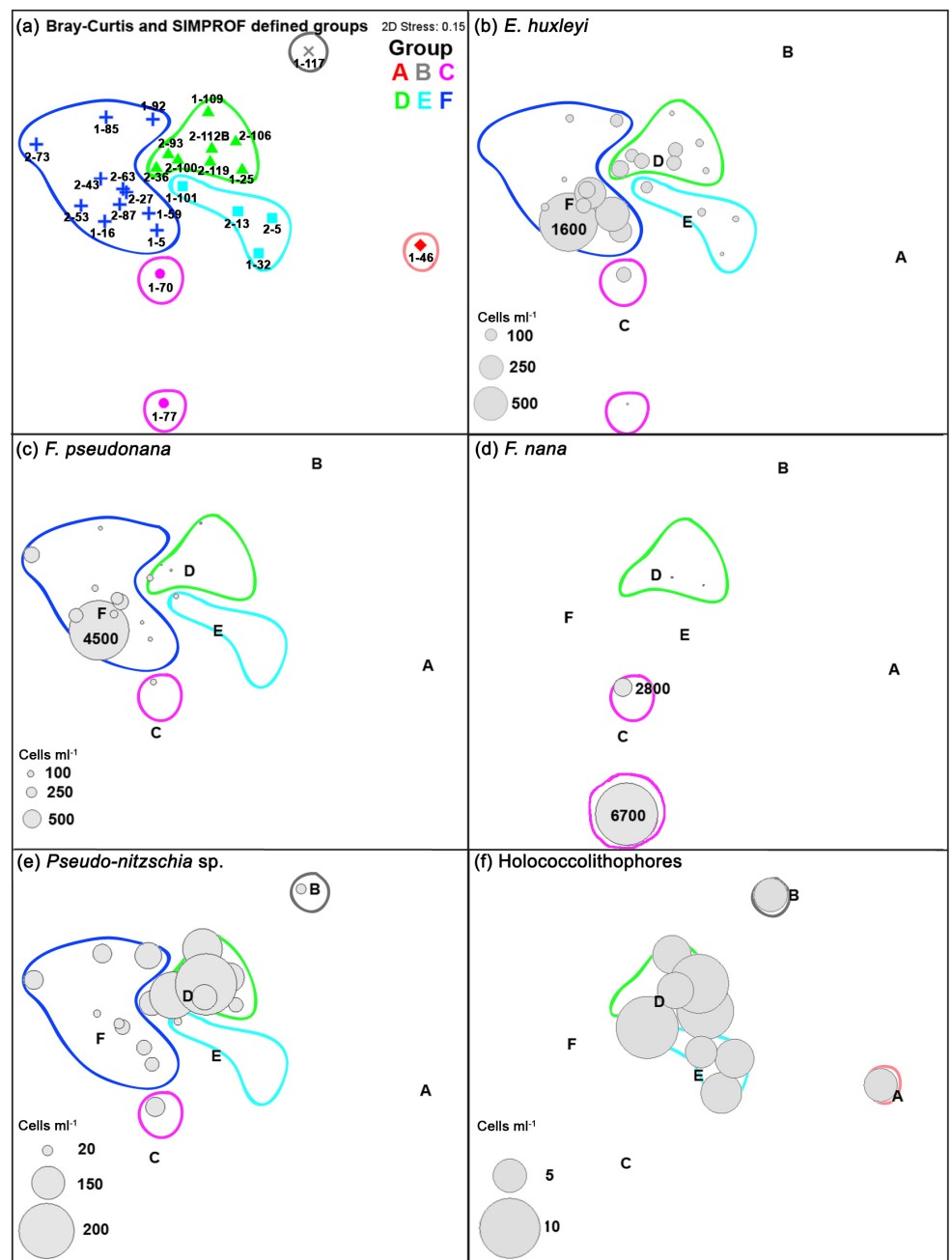

**Figure 5 Two dimensional non-metric multidimensional scaling (nMDS) ordination of station groupings A) as defined by the SIMPROF routine, with group color identifiers as in Fig. 3, where relative distances between samples represent the similarity of species composition between phytoplankton communities. Stations with statistically similar species composition are clustered together, whereas stations with low statistical similarity in terms of species composition are more widely spaced. Overlay of bubble plots of the defining species abundance (cells mL$^{-1}$) characterizing the statistically significant groups in the GCB (see also Table 4; (B)** *E. huxleyi* **abundance; (C)** *F. pseudonana* **abundance; (D)** *F. nana* **abundance; (E)** *Pseudo-nitzschia* **sp. abundance; and (E) Holococcolithophore abundance. The two-dimensional stress of 0.15 gives a 'reasonable' representation of the data in a 2-D space (Clarke and Warwick, 2001).**

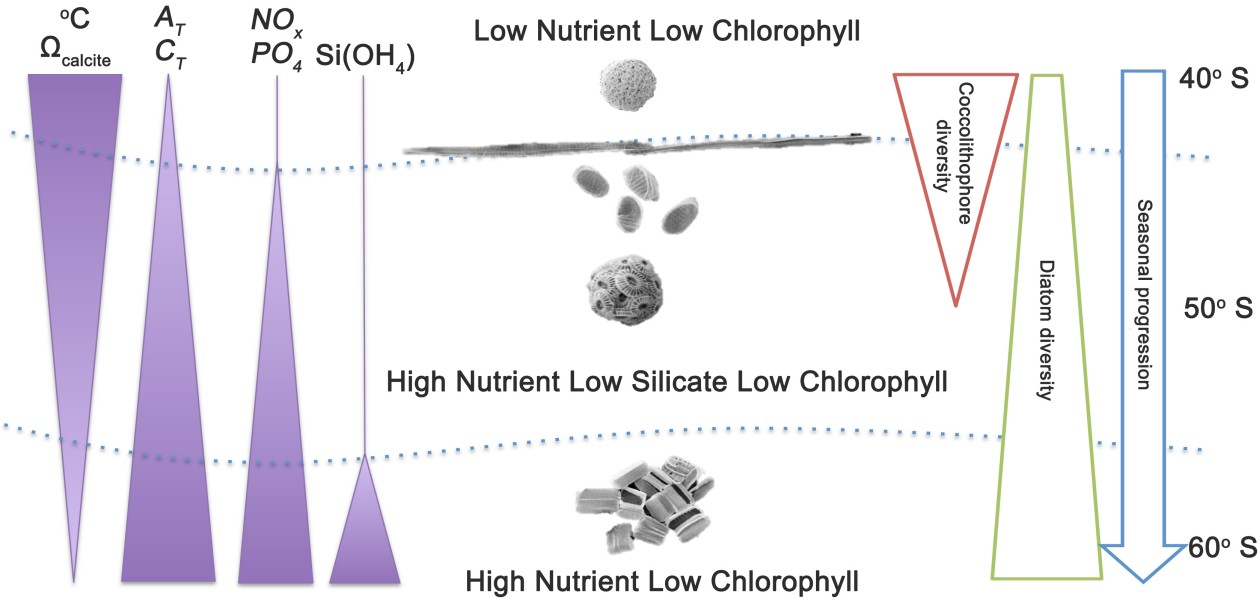

**Figure 6 Schematic of the potential seasonal progression occurring in the Great Calcite Belt, allowing coccolithophores to develop after the main diatom bloom. Note phytoplankton example images are not to scale.**