# Peer review of "The Influence of Environmental Variability on the Biogeography of Coccolithophores and Diatoms in the Great Calcite Belt"

_Biogeosciences, 2017_

## Referee Comment (RC1) · Anonymous Referee #1 · 8 May 2017

This article presents a comprehensive analysis of environmental forcing upon the distribution and abundance of dominant diatoms and coccolithophores in the Great Calcite Belt, a region of high importance for marine biogeochemical cycles. The study has been carefully conducted and the results are presented clearly and concisely. This work will contribute to improve our knowledge of the factors that control the biogeography of phytoplankton in the Southern Ocean. I support publication of this material in BG, provided the authors address some uncertainties in their analyses and conclusions. Reading the description of BGC at the beginning of the Introduction, one may be tempted to infer that the biogeochemical importance of the GCB (e.g. a region of net CO2 uptake) stems from the fact that it is a region of high PIC. However, its importance is probably more related to its being a region of generally increased plankton abundance and productivity. In fact Fig. 1 suggests that the region could be equally defined in terms of enhanced chla leveles. On a related note, is the PIC to POC ratio actually higher in this region than it is in tropical and subtropical waters? Some studies have shown that the coccolithophore to diatom biomass ratio actually increases in tropical, unproductive waters (Cermeno et al. PNAS 2008). This study uses abundance to assess dominance of different phytoplankton species. But, due to interspecific differences in cell size, an assesment based on carbon biomass could have been more reliable, as some of the authors have shown before (Daniels et al. MEPS 2016). For instance, section 3.2 starts by noting that nanoplanton tended to be more abundant than microphytoplankton, but this is always to be expected and cannot be directly translated to ecological dominance patterns. The authors should include a statement, and/or provide some sensitivity tests, on how results could change if dominance were assessed by biomass instead of abundance. page 12 line 20. A reference is needed here to support the value of chla content used for Ehux. However, the chla content of algal cells is highly dependent on temperature, light, nutrients, etc. which makes this calculation very uncertain. Carbon biomass is a more reliable metric to estimate relative importance of different species, because the C cellular content is less variable. The conclusion in the Abstract that temperature is the main driver of nanoplankton distribution should be qualified, as it may well be that temperature is co-varying with other factors that are the actual, ultimate drivers. On p. 10 line 10, what is the basis for statement that nanophytoplankton contribute 40% of total PP? The references provided do not have that kind of evidence (they are reviews on the ecology and biogeochemical role of diatoms). The authors should use instead remote sensing studies (e.g. Uitz et al. 2010 GBC) to support the statement that nanophytoplankton are the largest contributors to global marine PP. Minor point 'TOxN' is awkard and seems to suggest organic nitrogen. Better use 'NOx' or just nitrate (indicating in methods that nitrate acually refers to nitrate+nitrite). In any event nitrite concentrations are likely to be negligible, in comparison with nitrate, in these waters.

---

## Referee Comment (RC2) · Anonymous Referee #2 · 12 May 2017

Review of "The Influence of Environmental Variability on the Biogeography of Coccolithophores and Diatoms in the Great Calcite Belt " by Helen Smith

This manuscript presents phytoplankton cell counts results from the Southern Ocean from two cruises conducted in the GCB (Great Calcite Belt) together with a number of environemental physico-chemical data that are merged in a statistical analyses to provide causalistic hypotheses to plankton community structure.

The main results of this manuscript are that :

-coccolithophores and diatoms co-occur in the studied area and that coccolithophores in particular extend very far South,

-that community structure is mainly driven by four reprensative of the nanoplankton group (3 diatoms, 1 coccolithophore),

-that the key drivers of community structure are both T° and Si depletion which create different ecological niches.

Overall, I find the methods, results and main conclusions presented here are quite weak, with two main criticism :

**My first and main concern regards the phytoplankton cell counts.** I find that the method used for cellular abundance determination is not a very robust nor trustable method. Counting very small area of filtered samples in SEM is not usual for nano- or microphytoplankton determination. From what the authors indicate in their method section, I deducted that sample cell counts were determined on only 2 ml sample, which is insufficient in most cases to provide statistically robust results. If I agree with the authors general recommendation to use both SEM and light microscopy in paralell, it should be to count cell numbers in light microscopy on a sufficient volume (50-100 ml usually) and use SEM to improve species determination, and not the other way around. I don't understand why lugol/formol fixed samples were not collected or analyzed here. My second concern is on the large bias towards small species that this method implies, as correctly identified by the authors themselves. The main statement here about nanophytoplankton dominating the mineralizing algae is not trustworthy when large cells can not correctly be assessed by this method. The authors mispelled on several occasions diatom names, and include *Pseudo-nitzschia* sp. within the nanoplankton size-class which is quite surprising, as this species is most typically much larger than 20 μm, as can be seen very easily in figure 4. Also Figure 4 reveals very interestingly that a number of Parmales were present, they are part of the pico-nano- size fraction of siliceous plankton, so I find very surprising that no mention was made of that in the manuscript. SEM observations should also have allowed species determination for the dominant *Pseudo-nitzschia* species, which is not indicated. This suggests an overall lack of expertise for diatoms, and that calcifying algae were initially the focus of the study and that diatoms were only added lately to the analysis. I have a hard time believing the low species numbers (1-3) indicated for diatoms at certain stations.

Another point is the presentation of cellular abundance only. This is absolutely not the best metric to compare with physico-chemical parameters, and C biomass conversions are absolutely needed in this kind of data analysis. This would have allowed a relative estimation of the contribution of mineralizing algae to total POC (or Chla stretching it with POC:Chla ratios) and more robust conclusions regarding the real importance of both coccolithophores and nano-sized diatoms in total phytoplankton summer blooms.

**My second main concern is about the statistical analyses.** Although I will frankly admit that I am not qualified to expertise the tests presented here further than simple correlation matrixes, I really miss the added value of such extensive statistical tests. Quantifying so many environmental variables (such as carbonate chemistry which is very tricky) to collapse them in the end with T° and nutrients seem very odd to me. Finally, every bit of conclusion about the different phytoplankton communities and the overarching role of T° and silicic acid could have been stated by directly looking at the data and the statistics provided here are not at all convincing.

The discussion section leaves much to be desired and is a succession of short paragraphs that are very counter-intuitively organized and that should be entirely rewritten. A number of other papers regarding the succession patterns of coccolithophores and diatoms elsewhere are ignored.

I have several other comments/corrections/questions that are added as sticky notes in the manuscript pdf attached.

Based on these comments, I suggest either rejection or major revisions including entirely reworking both the dataset and its subsequent analysis.

[Figure]

**The Influence of Environmental Variability on the Biogeography of Coccolithophores and Diatoms in the Great Calcite Belt**

Helen E. K. Smith[1,2], Alex J. Poulton[1,3], Rebecca Garley[4], Jason Hopkins[5], Laura C. Lubelczyk[5], Dave T. Drapeau[5], Sara Rauschenberg[5], Ben S. Twining[5], Nicholas R. Bates[2,4], William M. Balch[5]

[1]National Oceanography Centre, European Way, Southampton, SO14 3ZH, U.K.
[2]School of Ocean and Earth Science, National Oceanography Centre Southampton, University of Southampton Waterfront Campus, European Way, Southampton, SO14 3ZH, U.K.
[3]Present address: The Lyell Centre, Heriot-Watt University, Edinburgh, EH14 7JG, U.K.
10   [4]Bermuda Institute of Ocean Sciences, 17 Biological Station, Ferry Reach, St. George's GE 01, Bermuda.
[5]Bigelow Laboratory for Ocean Sciences, 60 Bigelow Drive, P.O. Box 380, East Boothbay, Maine 04544, USA.

*Correspondence to*: Helen E.K. Smith (helen.eksmith@gmail.com)

**Abstract.** The Great Calcite Belt (GCB) of the Southern Ocean is a region of elevated summertime upper ocean calcite
15   concentration derived from coccolithophores, despite the region being known for its diatom predominance. The overlap of two major phytoplankton groups, coccolithophores and diatoms, in the dynamic frontal systems characteristic of this region, provides an ideal setting to study environmental influences on the distribution of different species within these taxonomic groups. Water samples for phytoplankton enumeration were collected from the upper 30 m during two cruises, the first to the South Atlantic sector (Jan-Feb 2011; 60$^o$ W-15$^o$ E and 36-60$^o$ S) and the second in the South Indian sector (Feb-Mar 2012;
20   40-120$^o$ E and 36-60$^o$ S). The species composition of coccolithophores and diatoms was examined using scanning electron microscopy at 27 stations across the Sub-Tropical, Polar, and Sub-Antarctic Fronts. The influence of environmental parameters, such as sea-surface temperature (SST), salinity, carbonate chemistry (i.e., pH, partial pressure of CO$_2$ ($p$CO$_2$), alkalinity, dissolved inorganic carbon), macro-nutrients (i.e., nitrate+nitrite, phosphate, silicic acid, ammonia), and mixed layer average irradiance, on species composition across the GCB, was assessed statistically. Nanophytoplankton (cells 2-20
25   μm) were the numerically abundant size group of biomineralizing phytoplankton across the GCB, the coccolithophore *Emiliania huxleyi* and the diatoms *Fragilariopsis nana*, *F. pseudonana* and *Pseudonitzschia* sp. were the most dominant and widely distributed species. A combination of SST, macro-nutrient concentrations and $p$CO$_2$ were the best statistical descriptors of biogeographic variability of biomineralizing species composition between stations. *Emiliania huxleyi* occurred in the silicic acid-depleted waters between the Sub-Antarctic Front and the Polar Front, indicating a favorable environment
30   for this coccolithophore in the GCB after spring diatom blooms remove silicic acid to limiting levels. After full consideration of variability in carbonate chemistry and temperature on the distribution of nanoplankton in the GCB, we find that temperature remains the dominant driver of biogeography in a large proportion of the modern Southern Ocean.

[Figure]

**1 Introduction**

The Great Calcite Belt (GCB), defined as an elevated particulate inorganic carbon (PIC) feature occurring in austral spring and summer in the Southern Ocean (Balch et al., 2005), plays an important role in climate fluctuations (Sarmiento et al. 1998, 2004), accounting for over 60% of the Southern Ocean area (30-60°S; Balch et al., 2011). The region between 30-50°S

5  is recognized as having the highest uptake of anthropogenic carbon dioxide ($CO_2$) alongside the North Atlantic Ocean (Sabine et al., 2004). The impact of future perturbations of ocean chemistry on Southern Ocean phytoplankton biogeography (e.g., Passow and Carlson, 2012) is poorly constrained. Understanding the current environmental influences on phytoplankton biogeography is therefore critical if model parameterizations are to improve (Boyd and Newton, 1999) and provide more accurate predictions of future biogeochemical change.

10  The Southern Ocean has often been considered as a micro-plankton (20-200 μm) dominated system with phytoplankton blooms dominated by large diatoms and *Phaeocystis* sp. (e.g., Bathmann et al., 1997; Poulton et al., 2007; Boyd, 2002). However, since the recent identification of the GCB as a consistent feature (Balch et al., 2005; 2016) and the recognition of the importance of pico- (< 2 μm) and nanoplankton (2-20 μm) in High Nutrient Low Chlorophyll (HNLC) waters (Barber and Hiscock, 2006) dynamics of small mineralizing plankton and their subsequent export need to be reconsidered. The

15  two dominant mineralizing phytoplankton groups in the GCB are coccolithophores and diatoms. Coccolithophores are generally found north of the PF (e.g., Mohan et al., 2008), though *Emiliania huxleyi* has been observed as far south as 58°S in the Scotia Sea (Holligan et al., 2010), at 61°S across Drake Passage (Charalampopoulou et al., 2016) and 65°S south of Australia (Cubillos et al., 2007).

[revised manuscript text omitted]

Increasing concentration of dissolved $CO_2$ in the oceans is resulting in 'ocean acidification' via a decrease in ocean pH (Caldeira and Wickett, 2003). In the high latitudes, where colder waters enhance the solubility of $CO_2$ and reduce the saturation state of calcite, there may be potential detrimental effects on calcifying phytoplankton (Doney et al., 2009). However, this may be species- (Langer et al., 2006) or even strain-specific (Langer et al., 2011), showing an optimum-response when the opposing influences of pH and bicarbonate are considered in a substrate-inhibitor concept (Bach et al., 2015). The response of non-calcifiers (e.g., diatoms) to ocean acidification is a greater unknown but no less important given their ~40 to 50% contribution to global primary production (e.g., Tréguer et al., 1995; Sarthou et al., 2005). Tortell et al.

[Figure]

(2008) observed a switch from small to large diatom species with increasing $CO_2$, indicating a potential change in future community structure. Large phytoplankton species (>50 µm) may also have the existing physiological traits to withstand changes in ocean chemistry over smaller (<50 µm) celled species (Flynn et al., 2012), as well as potentially being less susceptible to grazing pressure (Assmy et al., 2013). Alternatively, there may be a shift towards small phytoplankton groups

5    due to the expansion of low-nutrient subtropical regions (Bopp et al., 2001; Bopp, 2005). The response of Southern Ocean phytoplankton biogeography to future climate conditions, including ocean acidification, is complex (e.g., Charalampopolou et al., 2016; Petrou et al., 2016; Deppeler and Davidson, 2017) and therefore understanding existing relationships between *in situ* phytoplankton communities and ocean chemistry is an important stepping-stone for predicting future changes.

[revised manuscript text omitted]

5    *huxleyi* and *Pappamonas* sp*.)* were observed as intact coccospheres.

Diatoms dominated 15 stations in terms of mineralizing plankton numerical abundance across all environments sampled (Fig. 2, Table 2) and were found in every sample analyzed, contributing 62% on average to the total cell (coccolithophores + diatoms) abundance. Diatoms made up 100% of the total cell counts at the most southerly station in the South Indian Ocean (station GCB2-73) and 99.7% east of the South Sandwich Islands (station GCB1-77; Fig. 2). Seventy-six species of diatom

10   were identified as intact cells across the entire GCB. The most frequently occurring species in the GCB were small (< 5 µm in length) *Fragilariopsis* ssp.. The highest abundance of diatoms in the South Atlantic Ocean (6,787 cells $mL^{-1}$) was dominated by *F. nana* east of the South Sandwich Islands (station GCB1-77). The highest diatom abundance in the South Indian Ocean (514 cells $mL^{-1}$) was dominated by *F. pseudonana* at the most southerly station (station GCB2-73) sampled. Another frequently dominant diatom was *Pseudonitzschia* sp. that was most abundant north of the PF (Table 2).

15   Diatom species richness increased south towards $60^{o}$ S with the contribution of the different diatom species to total mineralizing plankton abundance fairly even (J' > 0.5, Table 2), except at stations (stations GCB1-70, GCB1-77, GCB2-27 and GCB2-63) where *Fragilariopsis* ssp. <5 µm were dominant (>70% of the diatom population, J' < 0.5). The highest diatom species richness (15 species) was found in the GCB south of the SAF (stations GCB1-85 and GCB2-36) at temperatures of $5^{o}$C to $8^{o}$C, in HNLSiLC conditions (TOxN >18 µmol $L^{-1}$, silicic acid <2 µmol $L^{-1}$, Chl a 0.21-1.11 mg $m^{-3}$).

20   **3.3 Statistical Analysis**

Three of the environmental variables were removed from the statistical analysis following a Spearman's rank ($r_s$) correlation analysis (Table S1). TOxN and $PO_4$ had a strong significant positive correlation ($r_s$ = 0.961, p < 0.0001) and so TOxN was deemed representative of both nutrients. Sea-surface temperature displayed significant negative correlations with both $C_T$ ($r_s$ = -0.981, p < 0.0001) and $A_T$ ($r_s$ = -0.953, p < 0.0001), and so sea surface temperature was taken as being representative of

25   these two variables of the carbonate chemistry system.

The variation in environmental variables across the GCB was examined using a Principal Component Analysis (PCA). The first principal component (PC1) accounted for 58% of the variation in environmental variables, with an additional 17% of environmental variation described by PC2 (Table 3). PC1 describes the main latitudinal gradients of environmental changes across the GCB (decreasing SST, increasing macronutrients). PC1 is a predominantly linear combination of SST, salinity,

30   TOxN, silicic acid, $NH_4$, and $\Omega_{calcite}$, where there is a significant positive correlation of PC1 with SST and salinity and a significant negative correlation with all other variables (Table 3). PC2 represented the environmental variation in the GCB

occurring independently of latitude, and was driven predominantly by variation in $p$CO$_2$, with weaker influences from $\bar{E}_{MLD}$ and pH (Table 3). PC2 had significant positive correlations with $p$CO$_2$ and $\bar{E}_{MLD}$ and a negative correlation with pH.

Variability in coccolithophore and diatom species composition across the GCB was assessed using a SIMPROF routine, comparing the abundance and diversity across all stations, to define groups with statistically similar community composition.

5 Six statistically significant groups ($p < 0.05$) were defined across the GCB (Fig. 3). Three groups of these groupings (A, B, C) were specific to the South Atlantic Ocean (Fig. 3). For example, groups A and B represented individual stations GCB1-46 and GCB1-117 respectively, in the sub-tropical region of the South Atlantic Ocean. The most southerly stations in the South Atlantic Ocean (stations GCB1-70 and GCB1-77) defined group C (Fig. 3). Groups D, E and F included stations across the GCB in both ocean regions. Here, group D was defined by eight stations sampled predominantly north of the SAF, while

10 group F was defined by 11 stations predominantly sampled south of the SAF (Fig. 3). These statistically defined similar community structures indicate that although the GCB covers a wide expanse of ocean, the community structure is consistently latitudinal defined across its longitudinal range.

The species driving the differences in mineralizing plankton community structure across the GCB were identified through a

[Figure]
 SIMPER routine (Table 4). Groups A and B were defined by the absence of *E. huxleyi* and the presence of either

15 holococcolithophores (group A) or the diatom *Cylindrotheca* sp. (group B). Group C was defined by the presence of *F. nana* (Table 4) and low contributions from *E. huxleyi* and *Pseudonitzschia* sp., with low diversity overall (total of 9 mineralizing species; Table 2) resulting in a significant difference from the other groups. Group D had higher total species diversity overall (i.e., 12-23 species; Table 2) and was defined by similar relative abundances of *E. huxleyi* and *Pseudonitzschia* sp., which were not found elsewhere (Table 4). Group E, including stations north of the SAF (Fig. 3), included *E. huxleyi*, *U.*

20 *tenuis* and holococcolithophores (Table 4). The low abundance of diatoms (3-125 cells mL$^{-1}$; Table 2) within group E separated it from the other groups (Table 4). The combination of *E. huxleyi*, *F. pseudonana* and *Pseudonitzschia* sp. that defined group F (Table 4) represented stations on the Patagonian Shelf and south of the SAF (Fig. 3). The almost mono-specific *E. huxleyi* coccolithophore community (Table 2) in group F highlights its strong dissimilarity from the other community structure groups identified (Table 4).

25 The abundance and distribution of four nanophytoplankton species, *E. huxleyi*, *Psuedonitzschia* sp., *F. nana* and *F. pseudonana* (Fig. 4), were identified as having the most significant contribution to differences in community structure across the GCB (Table 4, Fig. 5). *Emiliania huxleyi* and *F. pseudonana* were the most dominant coccolithophore and diatom species, respectively, across the GCB (Table 2). *Fragilariopsis pseudonana* was the numerically dominant diatom (> 30%) at seven stations in the South Indian Ocean (Table 2). The diatom with the highest abundance, *F. nana* (6797 cells mL$^{-1}$),

30 was almost exclusively found in the South Atlantic Ocean (Table 2; Fig. 5) and the more frequently occurring *Pseudonitzschia* sp. was present at all but two stations (Fig. 5).

[Figure]

The influence of environmental variables on the biogeography of coccolithophores and diatoms in the GCB was assessed using the BEST routine. The strongest Spearman's rank correlation ($r_s$ = 0.55, $p$ < 0.001) between all possible environmental variables and the biogeographical patterns observed came from a combination of five variables, including: (1) SST; (2-4) macronutrients (TOxN, silicic acid, $NH_4$); and (5) $p$$CO_2$. This was followed by a correlation of $r_s$ = 0.54 (p < 0.001) that included these parameters as well as $\Omega_{calcite}$. Salinity was included in the third highest correlation, whereas $\bar{E}_{MLD}$ and pH did not rank as significant factors in the BEST analysis.

**4 Discussion**

**4.1 Biogeography of coccolithophores and diatoms in the Great Calcite Belt**

Studies of Southern Ocean phytoplankton productivity have generally focused on the micro-phytoplankton (Barber and Hiscock, 2006) as these species contribute around 40% to total oceanic primary production (Tréguer et al., 1995; Sarthou et al., 2005). However, nanoplankton and picoplankton are becoming increasingly recognised as important contributors to total phytoplankton biomass, productivity and export in the Southern Ocean (e.g., Boyd, 2002; Hinz et al., 2012), both as the dominant size group in post-bloom (Le Moigne et al., 2013) and non-bloom conditions (Barber and Hiscock, 2006).

In this study, coccolithophores were generally numerically dominant at stations sampled north of the PF, particularly around the SubAntarctic Front, whereas diatoms were observed to be dominant at stations south of the PF (Fig. 2). There was also a significantly different species distribution (*a priori* ANOSIM; R = 0.227, $p$ < 0.01) north and south of the Sub-Antarctic Front, which has been previously identified as the divider between calcite and opal dominated export in the Southern Ocean (e.g., Honjo et al., 2000; Balch et al., 2016). Diatoms were more abundant (~570 cells $mL^{-1}$) than coccolithophores (~160 cells $mL^{-1}$) on average in the entire GCB. This contrasts to a study by Eynaud et al. (1999) in the South Atlantic Ocean at a similar time of year that reported a peak in coccolithophore cell abundance in the vicinity of the PF (a feature that was not observed in this study). These differences are likely to be due to the variability of Southern Ocean plankton on short temporal scales (Mohan et al., 2008), including variability in the seasonal progression of the spring bloom (Bathmann et al., 1997).

The coccolithophore *E. huxleyi* and diatoms *F. pseudonana*, *F. nana* and *Pseudonitzschia* sp. (Fig. 4) were all identified as being central to defining the statistical similarities within, and the differences between, the different mineralizing phytoplankton groups (Table 4, Fig. 5). These four species are all part of the nanoplankton and at the lower end of the size range of the microplankton (*Pseudonitzschia* sp. is ~20 µm in length), which can contribute significantly to biomass in the HNLC regions of the Southern Ocean (Boyd, 2002). *Emiliania huxleyi* and *Fragilariopsis* sp. less than 10 µm have been identified as two of the most abundant mineralizing phytoplankton further south in the Scotia Sea (Hinz et al. 2012). The results presented here further indicate that nanoplankton do have the potential to contribute a significant proportion to GCB

community composition alongside the larger phytoplankton (including large diatoms) typically associated with the HNLC region.
[Figure]

The abundance of HNLC diatoms such as *F. kerguelensis* (<10 cells mL$^{-1}$), *T. nitzschoides* (<20 cells mL$^{-1}$) and large *Chaetoceros* sp. (<10 cells mL$^{-1}$) were generally lower than those observed in other studies (e.g., Poulton et al., 2007;

5  Armand et al., 2008; Korb et al., 2010, 2012). Furthermore, the virtual absence of *Eucampia antarctica* (<1 cell mL$^{-1}$) in this study does not reflect the typical assemblage (sometimes > 600 cells mL$^{-1}$) found in previous studies (e.g., Kopczyaska et al., 1998; Eynaud et al., 1999; de Baar et al., 2005; Poulton et al., 2007; Salter et al., 2007; Korb et al., 2010). Low abundances of the large-celled diatoms in the silicic acid replete regions could be influenced by the small filter area analyzed using SEM; in this study the area imaged equates to a relatively small volume of water (i.e., 2-6 mL depending on

10  magnification) relative to the larger volumes (10-50 mL) often examined for light microscopy in other studies. Large, rare cells may not be enumerated from such small sample volumes, however the numerically abundant nanoplankton groups were well represented in SEM images. Conversely, samples preserved in acidic Lugol's solution for light microscopy analysis are biased towards larger species since small diatoms (<10 μm) are not clearly visible and coccolithophores are not well preserved (Hinz et al., 2012). Therefore, in future a combination of both imaging techniques should be used when examining

15  the phytoplankton community structure of the wider Southern Ocean.

**4.2 *Emiliania huxleyi* in the Great Calcite Belt**

The importance of coccolithophores in the GCB was examined via species community composition and abundance of intact cells, focusing on areas identified as having high PIC reflectance from underway sampling and satellite observations (Balch et al., 2014, 2016). Higher species diversity of coccolithophores occurred north of the STF (i.e., 4-13 species; Table 2).

20  Coccolithophores are diverse in the stratified and low-nutrient waters associated with lower latitudes (Winter et al., 1994). Only a few species are found in the colder waters south of the STF (Mohan et al., 2008), the most successful being *E. huxleyi*, which was observed at an abundance of 103 cells mL$^{-1}$ at 1$^{o}$C in this study in the South Atlantic (station GCB1-70). The 2$^{o}$C isotherm has been previously assumed to represent the southern boundary of *E. huxleyi* (e.g., Verbeek, 1989; Mohan et al., 2008) and inter-annual variability could be influenced by movement of the southern front of the Antarctic Circumpolar

25  Current (Holligan et al., 2010). The Southern Ocean *E. huxleyi* morphotype (Cook et al., 2011; Poulton et al., 2011) may therefore have a wider temperature tolerance than its northern hemisphere equivalent (Hinz et al., 2012) and has been observed poleward of 60$^{o}$ S further east in the Southern Ocean (Cubillos et al., 2007) and across Drake Passage (Charalampopoulou et al., 2016). There were three distinct *E. huxleyi* features (the Patagonian Shelf, north of South Georgia and north of the Crozet Islands) within the GCB where *E. huxleyi* contributed > 50% of the total cell counts of mineralizing

30  phytoplankton. *Emiliania huxleyi* was most abundant (1636 cells mL$^{-1}$) on the Patagonian Shelf and was the most frequently occurring coccolithophore across the entire GCB. The main *E. huxleyi* features are discussed further below to understand why this species is so widely distributed in the GCB.

[Figure]

**4.2.1 Patagonian Shelf**

The Patagonian Shelf is a well-known region for *E. huxleyi* blooms, as observed in satellite imagery between November and January (i.e., Signorini et al., 2006; Painter et al., 2010; Balch et al., 2011; Garcia et al., 2011; Balch et al., 2014). The *E. huxleyi* cell abundance observed in this study (~1600 cells mL$^{-1}$) was similar to that found by Poulton et al. (2013; >1000 cells mL$^{-1}$). Using a value of 0.2 pg Chl a per cell (Haxo, 1985) following Poulton et al. (2013), such *E. huxleyi* abundance levels are equivalent to contributions of only ~12% to the total Chl *a* signal (~2.8 mg m$^{-3}$), which is a similar contribution to that estimated in an identical way by Poulton et al. (2013). This data combined with the satellite observations supports the hypothesis of a similar phytoplankton structure repeating 
[revised manuscript text omitted]
 patchiness and deviation away from the theoretical species abundances relative to temperature or other environmental factors. In contrast, the negative correlation of *F. nana* (Pearson's product moment correlation = -0.976, $p < 0.05$, n = 4) versus the positive correlation of *Pseudonitzschia* sp. (Pearson's product moment correlation = 0.544, $p < 0.05$, n = 19) with SST indicates that these two species have distinctly different physiological tolerances. Southern Ocean diatoms are mostly observed to have negative relationship with temperature (e.g. Eynaud et al., 1999; Boyd, 2002). *Pseudonitzschia* sp. was predominantly found in waters north of the PF in this study, as seen by Kopczynska et al. (1986), and is likely to be out competed by other diatom species (e.g. *Chaetoceros* sp. and *Dactyliosolen* sp.) further south due to different nutrient affinities and requirements (Kopczynska et al., 1986), particularly for dissolved iron and silicic acid.

**4.3.2 Nutrients**

Macronutrient gradients, particularly silicic acid, are considered one of the key driving factors between the differences in community structure in the Southern Ocean (Nelson and Treguer, 1992). TOxN (and $PO_4$ by association) was identified in the BEST test as an important factor in the variability of phytoplankton distribution, but did not significantly correlate with the four dominant phytoplankton species (Fig. 4) contributing over 50% to changes in species composition in the GCB.

Nitrate drawdown by Southern Ocean diatoms is limited by dissolved iron (dFe) availability south of the STF (Sedwick et al., 2002), which may explain the dominance of the nanoplankton (with lower dFe and macronutrient requirements; Ho et al., 2003) in this study as they are not affected by low dFe concentrations as severely as the microplankton. The low silicic acid concentrations in the region between the SAF and the PF indicate that there was sufficient dFe to allow silicification and diatom growth, but either one or both of the macronutrients were then depleted to limiting concentrations (Assmy et al., 2013). As an essential nutrient for diatoms, silicic acid concentrations less than 2 µmol Si $L^{-1}$ were most common in the GCB, a level which is considered limiting for most diatom species (Paasche, 1973a & b; Egge and Asknes, 1992). However, even at stations with greater than 5 µmol Si $L^{-1}$, the small diatom species (<10 µm) were still dominant and represented over 40% of the total coccolithophore and diatom assemblage (numerically). There was a significant positive correlation between

silicic acid and the small (<5 µm) diatom *F. nana* (Pearson's product moment correlation = 0.986, *p* < 0.05, n = 4), although *F. nana* is likely to have a low cellular silicate requirement similar to *F. pseudonana* (Poulton et al., 2013) relative to larger diatom species, so the high abundance of *F. nana* in the high silicic acid waters could be indicative of a seasonal progression rather than silicic acid dependence. *Fragilariopsis* sp. have been observed at high abundances near the Ross Sea ice shelf

5    (Grigorov and Rigual-Hernandez, 2014) and high abundances of large diatoms in the silicic acid- (and dFe-) replete waters may have been found further south than the sampling strategy of this study allowed. In the South Atlantic and the South Pacific Ocean silicic acid depletion moves southwards as spring to summer progresses, with a maximum diatom biomass observed in late January at 65°S (Sigmon et al., 2002; Le Moigne et al., 2013).

A significant negative correlation between *E. huxleyi* and silicic acid (Pearson's product moment correlation = -0.410, *p* <

10    0.05, n = 24) was found in this study, as has also been identified in the Scotia Sea (Hinz et al., 2012) and Patagonian Shelf (Balch et al., 2014). Low silicic acid may be considered a positive selection pressure for coccolithophores (Holligan et al. 2010), especially when other macronutrients (and dFe) are replete. However, non-blooming coccolithophore species are now recognized as having silicic acid requirements, though this requirement is conspicuously absent from *E. huxleyi* (Durak et al., 2016). Therefore the positive selection pressure at low silicic acid concentrations in the GCB is likely to be *E. huxleyi*

15    specific rather than a coccolithophore-wide phenomena. To the south of the PF silicic acid increased (from < 1 to > 3 µmol Si L$^{-1}$) with five stations between the SAF and PF (and one south of the PF, station GCB1-59), all numerically dominated by *E. huxleyi*, while other stations to the south of the PF were dominated by diatoms (Fig. 2).

These results from the GCB indicate a progression of mineralizing phytoplankton southwards during spring as irradiance conditions become optimal and macronutrients are depleted. Low silicic acid is often associated with a high residual nitrate

20    concentrations (defined as [NO$_3^-$] - [Si(OH)$_4$]), as has been observed on the Patagonian Shelf (Balch et al., 2014). The highest coccolithophore abundances in this study (excluding the Patagonian Shelf) were indeed observed in regions with 'residual nitrate' concentrations greater than 10 µmol NO$_3$ L$^{-1}$ (Balch et al., 2016). As silicic acid is depleted in the more northerly surface waters in spring, diatoms progressively become more successful further south as irradiance conditions allow, thereby producing a large HNLSiLC area between the Sub-Antarctic Front and Polar Front; an ideal environment for

25    late summer *E. huxleyi* dominated communities to develop (Figure 6).

Dissolved iron is recognized as a strong control on phytoplankton growth, community composition and species biogeography (e.g., Boyd, 2002, Boyd et al., 2015). In this study, dFe measurements were only made at a small number of sampling stations (n = 6; Twining, unpublished data, Balch et al., 2016) limiting their use in the multivariate statistical analysis of community composition. For these stations dFe showed a statistically significant negative correlation (Pearson's

30    product moment = -0.957, *p* < 0.01) with PC2 from the environmental analysis (Fig. S2). PC2 described the environmental variables least related to latitude (pH, *p*CO$_2$ and Ē$_{MLD}$), indicating that dFe was also decoupled from the strong latitudinal gradient in the environmental parameters (i.e. SST, Ω$_{calcite}$, macronutrients) in the GCB in the austral spring/summer.

[Figure]

Interestingly, dFe concentrations did positively correlate with coccolithophore abundance (Pearson's product moment correlation = 0.858, $p$ <0.05) rather than diatom abundance ($p$ = 0.132, ns) (Fig. S2). Overall, these data support the hypothesis that coccolithophores occupy a niche unoccupied by large diatoms when dFe is replete and silicic acid is depleted (Balch et al., 2014; Hopkins et al., 2015). The numerical dominance of small diatoms less than 20 μm in the GCB during austral spring and summer, alongside the coccolithophore *E. huxleyi*, is thus potentially due to the reduced impact of nutrient limitation (dFe, silicic acid) on small cells with high ratios of surface area to volume (e.g., Hinz et al., 2012; Balch et al., 2014).

**4.4 Relating the Great Calcite Belt to carbonate chemistry**

Relating carbonate chemistry to phytoplankton distribution, growth and physiology is an important step when considering the potential effects of climate change and ocean acidification on marine biogeochemistry. In this study, no significant correlation (Spearman's r = 0.259, $p$ = 0.164, n = 27) occurred between pH and Chl *a*. The inclusion of $p$CO$_2$ and $\Omega_{calcite}$ as influential factors in describing the GCB biogeography highlights the importance of understanding phytoplankton responses to carbonate chemistry as a whole rather than as individual carbonate chemistry parameters (Bach et al., 2015). Of the four major species driving the differences in mineralizing plankton community composition and biogeography across the GCB, only *F. pseudonana* abundance was positively correlated with $p$CO$_2$ (Pearson's product moment coefficient = 0.577, $p$ < 0.05, n = 16).

The response of diatoms to increasing $p$CO$_2$ is not straight forward (e.g., Boyd et al., 2015), with some studies implying that large diatoms may be more successful in future climate scenarios (e.g., Tortell et al., 2008; Flynn et al., 2012), although changes in nutrient and light availability (via stronger stratification) may prevent a permanent switch in phytoplankton community structure (Bopp, 2005). The carbonate chemistry system is complex as biological activity also impacts on the concentration of each of the components. Organic matter production reduces dissolved inorganic carbon ($C_T$) and hence $p$CO$_2$ via photosynthesis, as well as increasing alkalinity ($A_T$) through nutrient uptake, while subsequent respiration and remineralisation of organic matter has the opposite impact. The simultaneous actions of biological and physical processes result in seasonal and localized changes in the carbonate system, which are often difficult to decouple.

In our study, there was no significant correlation between *E. huxleyi* and $\Omega_{calcite}$ (Pearson's product moment = 0.093), which may be viewed as somewhat surprising given the potential detrimental effects on calcifiers at low saturation states (e.g. Riebesell et al., 2000). 
[revised manuscript text omitted]

|---|---|---|---|---|---|---|---|---|---|---|---|---|---|---|---|
| | °S | °E | °C | | mol PAR m$^{-2}$ d$^{-1}$ | | | μmol L$^{-1}$ | | | | | μatm | | mg m$^{-3}$ |
| GCB1-6 | 51.79 | -56.11 | 8.6 | 34.0 | 17.8 | 14.2 | 1.05 | 1.7 | 0.64 | 2336 | 2138 | 8.09 | 367 | 3.3 | 0.84 |
| GCB1-16 | 46.26 | -59.83 | 11.8 | 33.8 | 39.8 | 6.5 | 0.54 | 0.0 | 0.15 | 2333 | 2100 | 8.12 | 407 | 3.8 | 2.78 |
| GCB1-25 | 45.67 | -48.95 | 16.1 | 35.1 | 25.5 | 0.0 | 0.23 | 0.2 | 0.16 | 2320 | 2047 | 8.12 | 390 | 4.6 | 0.73 |
| GCB1-32 | 40.95 | -45.83 | 20.0 | 35.6 | 36.7 | 0.1 | 0.11 | 1.1 | 0.05 | 2307 | 2029 | 8.07 | 444 | 4.8 | 0.05 |
| GCB1-46 | 42.21 | -41.21 | 18.3 | 34.9 | 16.0 | 0.2 | 0.19 | 0.3 | 0.00 | 2328 | 2050 | 8.09 | 356 | 4.7 | 0.09 |
| GCB1-59 | 51.36 | -37.84 | 5.9 | 33.8 | 7.9 | 17.5 | 1.22 | 1.7 | 0.67 | 2368 | 2184 | 8.10 | 325 | 3.1 | 0.71 |
| GCB1-70 | 59.25 | -33.15 | 1.1 | 34.0 | 9.7 | 22.3 | 1.74 | 78.5 | 1.54 | 2388 | 2235 | 8.10 | 407 | 2.6 | 0.13 |
| GCB1-77 | 57.28 | -25.98 | 1.4 | 33.9 | 11.9 | 20.7 | 1.55 | 68.8 | 1.00 | 2386 | 2225 | 8.12 | 405 | 2.7 | 0.90 |
| GCB1-85 | 53.65 | -17.75 | 4.1 | 33.9 | 8.9 | 19.1 | 1.33 | 0.7 | 0.30 | 2369 | 2191 | 8.12 | 363 | 3.0 | 1.11 |
| GCB1-92 | 50.4 | -10.8 | 5.9 | 33.8 | 9.5 | 17.5 | 1.27 | 1.4 | 0.37 | 2362 | 2182 | 8.10 | 351 | 3.0 | 0.57 |
| GCB1-101 | 46.31 | -3.21 | 11.0 | 34.0 | 17.1 | 12.5 | 0.95 | 0.6 | 0.16 | 2345 | 2134 | 8.08 | 400 | 3.5 | 0.46 |
| GCB1-109 | 42.63 | 3.34 | 15.1 | 34.4 | 20.0 | 5.3 | 0.56 | 0.8 | 0.00 | 2332 | 2098 | 8.07 | 359 | 4.0 | 0.39 |
| GCB1-117 | 39.00 | 9.49 | 18.8 | 35.0 | 19.4 | 0.0 | 0.20 | 0.7 | 0.06 | 2321 | 2047 | 8.08 | 299 | 4.7 | 0.32 |
| GCB2-5 | 37.09 | 39.48 | 21.0 | 35.5 | 11.2 | 0.0 | 0.05 | 1.1 | 0.07 | 2310 | 2005 | 8.10 | 340 | 5.2 | 0.12 |
| GCB2-13 | 40.36 | 43.5 | 18.4 | 35.3 | 13.7 | 0.1 | 0.17 | 0.2 | 0.02 | 2307 | 2032 | 8.09 | 351 | 4.7 | 0.19 |
| GCB2-27 | 45.82 | 51.05 | 7.7 | 33.7 | 5.8 | 20.1 | 1.35 | 2.9 | 0.14 | 2344 | 2194 | 8.00 | 425 | 2.6 | 0.47 |
| GCB2-35 | 46.74 | 57.48 | 8.1 | 33.7 | 8.7 | 18.9 | 1.40 | 1.7 | 0.49 | 2363 | 2175 | 8.08 | 355 | 3.1 | 0.21 |
| GCB2-43 | 47.52 | 64.04 | 6.5 | 33.7 | 5.9 | 21.7 | 1.53 | 0.5 | 0.38 | 2358 | 2197 | 8.04 | 387 | 2.8 | 0.34 |
| GCB2-53 | 49.3 | 71.32 | 5.1 | 33.7 | 8.5 | 23.8 | 1.66 | 7.1 | 0.17 | 2359 | 2210 | 8.03 | 396 | 2.6 | 0.41 |
| GCB2-63 | 54.4 | 74.56 | 3.5 | 33.8 | 3.0 | 25.3 | 1.70 | 10.5 | 0.21 | 2363 | 2210 | 8.07 | 360 | 2.6 | 0.26 |
| GCB2-73 | 59.71 | 77.75 | 1.1 | 33.9 | 4.3 | 28.0 | 1.91 | 40.4 | 0.34 | 2372 | 2233 | 8.07 | 360 | 2.4 | 0.29 |
| GCB2-87 | 54.25 | 88.14 | 3.4 | 33.9 | 4.3 | 24.2 | 1.69 | 9.0 | 0.45 | 2367 | 2216 | 8.06 | 367 | 2.6 | 0.28 |
| GCB2-93 | 49.81 | 94.13 | 7.8 | 34.0 | 5.9 | 17.5 | 1.27 | 1.5 | 0.26 | 2345 | 2149 | 8.10 | 333 | 3.3 | 0.18 |
| GCB2-100 | 44.62 | 100.5 | 13.0 | 34.8 | 4.7 | 6.4 | 0.55 | 0.2 | 0.15 | 2328 | 2083 | 8.11 | 326 | 4.1 | 0.33 |
| GCB2-106 | 40.13 | 105.38 | 17.0 | 35.4 | 12.8 | 0.1 | 0.14 | 0.3 | 0.03 | 2318 | 2029 | 8.13 | 313 | 4.9 | 0.24 |
| GCB2-112 | 40.26 | 109.6 | 15.8 | 34.9 | 11.1 | 3.6 | 0.43 | 0.2 | 0.00 | 2323 | 2060 | 8.11 | 332 | 4.4 | 0.36 |
| GCB2-119 | 42.08 | 113.4 | 13.8 | 34.8 | 11.2 | 5.3 | 0.55 | 0.2 | 0.01 | 2320 | 2080 | 8.10 | 342 | 4.1 | 0.27 |

[Figure]

[Figure]

**Table 2**

| Station | Position | Cell mL$^{-1}$ | S | J' | Dominant species | % of Co | Cell mL$^{-1}$ | S | J' | Dominant species | % of D |
|---------|----------|------|---|----|------|---------|------|---|----|------|--------|
| | | Coccolithophores (Co) | | | | | Diatoms (D) | | | | |
| GCB1-6 | SAF, PS | 242 | 1 | **** | *E. huxleyi* | 100 | 125 | 10 | 0.89 | *C. deblis* | 26 |
| GCB1-16 | SAF, PS | 1636 | 1 | **** | *E. huxleyi* | 100 | 4589 | 2 | 0.21 | *F. pseudonana* | 96 |
| GCB1-25 | SAFZ | 53 | 5 | 0.82 | *S. mollischi* | 38 | 25 | 7 | 0.87 | *Pseudonitzschia* sp. | 37 |
| GCB1-32 | STF | 22 | 6 | 0.89 | *U. tenuis* | 31 | 15 | 3 | 0.79 | *Nitzschia* sp. | 55 |
| GCB1-46 | STF | 3 | 1 | **** | Holococco* | 100 | 2 | 1 | **** | *Chaetoceros* sp. | 56 |
| GCB1-59 | sPF, n SG | 565 | 1 | **** | *E. huxleyi* | 100 | 164 | 14 | 0.78 | *T. nitzschoides* | 29 |
| GCB1-70 | sPF | 103 | 1 | **** | *E. huxleyi* | 100 | 700 | 8 | 0.36 | *F. nana* | 81 |
| GCB1-77 | sPF, SS | 2 | 1 | **** | *E. huxleyi* | 100 | 6787 | 1 | **** | *F. nana* | 98 |
| GCB1-85 | sPF | 28 | 1 | **** | *E. huxleyi* | 100 | 139 | 15 | 0.86 | *C. aequatorialis* sp. | 22 |
| GCB1-92 | PFZ | 77 | 2 | 0.13 | *E. huxleyi* | 98 | 102 | 14 | 0.8 | *Pseudonitzschia* sp. | 32 |
| GCB1-101 | SAFZ | 91 | 5 | 0.64 | *E. huxleyi* | 68 | 50 | 6 | 0.66 | *F. pseudonana* | 59 |
| GCB1-109 | SAFZ | 38 | 8 | 0.93 | *E. huxleyi* | 25 | 125 | 12 | 0.57 | *Pseudonitzschia* sp. | 61 |
| GCB1-117 | STF | 13 | 4 | 0.95 | *U. tenuis* | 35 | 204 | 2 | 0.17 | *C. closterium* | 95 |
| GCB2-5 | STFZ | 34 | 8 | 0.75 | *E. huxleyi* | 46 | 3 | 1 | **** | *Nanoneis hasleae* | 47 |
| GCB2-13 | STFZ | 46 | 8 | 0.65 | *E. huxleyi* | 57 | 25 | 3 | 0.7 | *Nitzschia* sp.*<20µm* | 67 |
| GCB2-27 | SAF, Cr | 472 | 1 | **** | *E. huxleyi* | 100 | 350 | 7 | 0.27 | *F. pseudonana* | 82 |
| GCB2-36 | SAF | 164 | 4 | 0.43 | *E. huxleyi* | 83 | 146 | 15 | 0.79 | *F. pseudonana* | 33 |
| GCB2-43 | PFZ | 11 | 1 | **** | *E. huxleyi* | 100 | 83 | 11 | 0.63 | *F. pseudonana* | 54 |
| GCB2-53 | sPF, K | 51 | 3 | 0.9 | *E. huxleyi* | 56 | 494 | 7 | 0.56 | *F. pseudonana* | 47 |
| GCB2-63 | sPF, H | 132 | 1 | **** | *E. huxleyi* | 100 | 245 | 9 | 0.46 | *F. pseudonana* | 71 |
| GCB2-73 | sPF | 0 | 0 | **** | n/a | n/a | 514 | 11 | 0.64 | *F. pseudonana* | 56 |
| GCB2-87 | sPF | 106 | 1 | **** | *E. huxleyi* | 100 | 172 | 8 | 0.73 | *F. pseudonana* | 42 |
| GCB2-93 | PFZ | 98 | 4 | 0.49 | *E. huxleyi* | 80 | 71 | 14 | 0.77 | *Pseudonitzschia* sp. | 37 |
| GCB2-100 | SAFZ | 121 | 6 | 0.31 | *E. huxleyi* | 87 | 155 | 8 | 0.55 | *Pseudonitzschia* sp. | 67 |
| GCB2-106 | STF | 88 | 13 | 0.84 | *E. huxleyi* | 29 | 76 | 10 | 0.66 | *Pseudonitzschia* sp. | 54 |
| GCB2-112 | STF | 120 | 6 | 0.41 | *E. huxleyi* | 80 | 242 | 9 | 0.43 | *Pseudonitzschia* sp. | 74 |
| GCB2-119 | SAFZ | 117 | 8 | 0.35 | *E. huxleyi* | 82 | 63 | 8 | 0.64 | *Pseudonitzschia* sp. | 47 |

[Figure]

**Table 3**

| Variable | PC1 - EV 5 (58%) | | PC2 - EV 1.5 (17%) | |
|---|---|---|---|---|
| Temp | 0.42 | **(0.97\*\*\*)** | 0.08 | (-0.10) |
| Salinity | 0.36 | **(0.90\*\*\*)** | 0 | - |
| EML | 0.24 | **(-0.55\*)** | 0.5 | **(0.62\*\*)** |
| TOXN | -0.4 | **(-0.91\*\*\*)** | -0.05 | (-0.06) |
| SIL | -0.35 | **(-0.77\*\*\*)** | 0.02 | (-0.03) |
| $NH_4$ | -0.35 | **(-0.81\*\*\*)** | -0.07 | (-0.09) |
| pH | 0.18 | **(-0.39)** | -0.42 | **(-0.50\*)** |
| $pCO_2$ | -0.15 | (-0.33) | 0.75 | **(0.89\*\*\*)** |
| $\Omega_{calcite}$ | 0.43 | **(-0.99\*\*\*)** | -0.02 | (-0.02) |

**Table 4**

| Group | Station | Location | Group Av.Sim% | Defining Species | Average Similarity | Similarity SD | Contribution % |
|---|---|---|---|---|---|---|---|
| **A** | GCB1-46 | STF | n/a | Holococco* | n/a | n/a | n/a |
| **B** | GCB1-117 | | | *Cylindrotheca* sp. | | | |
| **C** | GCB1-70 | SBDY | 54.5 | *F. nana* | 53.3 | n/a | 97.8 |
| | GCB1-77 | | | | | | |
| **D** | GCB1-25 | N of PF | 47.6 | *E. huxleyi* | 13.9 | 2.68 | 29.3 |
| | GCB1-109 | | | *Pseudonitzschia* sp. | 12.7 | 3.6 | 26.7 |
| | GCB2-36 | | | | | | |
| | GCB2-93 | | | | | | |
| | GCB2-100 | | | | | | |
| | GCB2-106 | | | | | | |
| | GCB2-112 | | | | | | |
| | GCB2-119 | | | | | | |
| **E** | GCB1-32 | N of SAF | 42.3 | *E. huxleyi* | 18.9 | 3.8 | 44.8 |
| | GCB1-101 | | | Holococco* | 8.45 | 4.01 | 20 |
| | GCB2-5 | | | | | | |
| | GCB2-13 | | | | | | |
| **F** | GCB1-6 | PS | 40.6 | *E. huxleyi* | 15.1 | 1.51 | 37.3 |
| | GCB1-16 | | | *F. pseudonana* | 14.2 | 1.25 | 35 |
| | GCB1-59 | S of SAF | | | | | |
| | GCB1-85 | | | | | | |
| | GCB1-92 | | | | | | |
| | GCB2-27 | | | | | | |
| | GCB2-43 | | | | | | |
| | GCB2-53 | | | | | | |
| | GCB2-63 | | | | | | |
| | GCB2-73 | | | | | | |
| | GCB2-87 | | | | | | |

[Figure]

[Figure]

**Figures**

[Figure]

**Figure 1 Rolling 32 day composite from MODIS-Aqua for both (a) Chlorophyll a (mg m-3) and (b) PIC (µmol L-1) for the South Atlantic sector (17th January to 17th February 2011) and the South Indian sector (18th February to 20th March 2012). Station number identifiers and averaged positions of fronts as defined by Orsi et al. (1995) are superimposed: Sub-tTropical front (STF), Sub Antarctic front (SAF), Polar Front (PF), Southern Antarctic Circumpolar Current Front (SACCF) and Southern Boundary (SBDY).**

[Figure]

[Figure]

**Figure 2 Coccolithophore and diatom abundance and dominance information. The area of the circles denotes abundance while shading denotes percentage contribution of each phytoplankton group, where red denotes coccolithophore dominance and blue denotes diatom dominance. Fronts are defined as in Figure 1**

[Figure]

**Figure 3 Statistically significant groups of coccolithophore and diatom communities in the Great Calcite Belt as identified by the SIMPROF routine. The colors designate which statistical group defines the coccolithophore and diatom assemblage at each station as shown in the group key. Fronts are defined as in Figure 1. See Table 4 for full group species descriptions.**

[Figure]

[Figure]

[Figure]

**Figure 4 SEM images of the four phytoplankton species identified by the SIMPER analysis as characterizing the significantly different community structures. (a)** *E. huxleyi*; **(b)** *F. pseudonana*; **(c)** *F. nana*; **and (d):** *Pseudonitzschia* sp.. **Scale bar 2 um for a-c and 5 um for d.**

[Figure]

[Figure]

**Figure 5 Two dimensional non-metric multidimensional scaling (nMDS) ordination of station groupings A) as defined by the SIMPROF routine, with group color identifiers as in Figure 3, where relative distances between samples represent the similarity of species composition between phytoplankton communities. Overlay of bubble plots of the defining species abundance (cells mL−1) characterizing the statistically significant groups in the GCB (see also Table 4; (B)** *E. huxleyi* **abundance; (C)** *F. pseudonana* **abundance; (D)** *F. nana* **abundance; (E)** *Pseudonitzschia* **sp. abundance; and (E) Holococcolithophore abundance. The two-dimensional stress of 0.15 gives a 'reasonable' representation of the data in a 2-D space (Clarke and Warwick, 2001).**

[Figure]

[Figure]

**Figure 6 Schematic of the potential seasonal progression occurring in the Great Calcite Belt, allowing coccolithophores to develop after the main diatom bloom. Note phytoplankton example images are not to scale.**

---

## Referee Comment (RC3) · Anonymous Referee #3 · 19 May 2017

This work adds incremental knowledge about the environmental forcing of coccolithophores and diatoms distribution in the southern ocean. In the future, the importance of this work may be that it serves as a base line study. The paper is well written and substantial with many references, although I believe it could be shortened by about 25% and still say the same. Based on the abstract and conclusions there is not much new insight except that the authors are looking at diatoms and coccolithophores at the same time. There is a host of environmental data and they are discussed at length but few significant patterns emerge which is often the case in beyond control "ships of opportunity studies" where the research is constrained by circumstances, timing, and sampling strategy. This brings up the next point:

The collections design was perhaps not ideal. In the paper (page 4 line 25) it says that water was collected from the upper 30 m. Apparently, only one liter of water was sampled that integrates the entire 30 m? This is really precious little water unless I am reading this incorrectly in which case it needs to be explained. It is well known that phytoplankton biomass can occur below this level (e.g. Hegseth and Sundfjord, 2008). Why was the collection limited to 30m?
Also the method of identification is not really suited to a detailed morphological analysis of *E.hux* which is important especially in the southern oceans where there exist various morpho/phenotypes of this species. What were the reasons for the magnifications differing at 5kx and 3kx? What about all the other material on the filter? There are many generalities in the paper that could use more explanation. Some of these are defined by G below;

G: Page 2 line 5: Takahashi wrote many papers on CO2 sequestration of CO2. How do we know whether the CO2 that is being taken up by areas of the ocean is anthropogenic or natural? Also the North Pacific is also such an area.

G: >Page 2 line 7-9: vague sentences. Poorly constrained, critical? Why
G: >Page 2 line 28-30: Why important

Page 3 line 1…."south of ~30ºS and extends to ~60º" (This has already been stated.

G: Page 3 line 25 "uncertainties" why?

Page 5 line 11: Why were individual coccoliths not counted? This can also say a lot about the age of the community.

Page 7 line 5-13. Can all these parameters be displayed graphically?

Page 7 line 29. Maybe I missed it but what were ALL of the 28 coccolithophore species?

Page 11 line 28 "occurrences" instead of "features"

Page 12 line 7. Where is the rest of the Chl coming from? This section is (lines 7-9) is not clear

Page 13 line 4 "coccolithophores IN this region"

Page 14 line 10 What is the "theoretical species abundance"?

Page 15 line13. "However A FEW non-blooming

Page 15 line 14 "conspicuously absent" why is this conspicuous?

Page 15 line 14-15 "to be a *E. huxleyi* specific rather than a coccolithophore-wide phenomena" not clear what the authors meant to say.   I don't agree.

Page 16 lines 25-27…There are many studies con and pro for this sentence

---

## Author Comment (AC3) · 7 Jul 2017

Reviewer #3 Reviewer Comment (RC) - This work adds incremental knowledge about the environmental forcing of coccolithophores and diatoms distribution in the southern ocean. In the future, the importance of this work may be that it serves as a base line study. The paper is well written and substantial with many references, although I believe it could be shortened by about 25% and still say the same. Based on the abstract and conclusions there is not much new insight except that the authors are looking at diatoms and coccolithophores at the same time. There is a host of environmental data and they are discussed at length but few significant patterns emerge which is often the

case in beyond control "ships of opportunity studies" where the research is constrained by circumstances, timing, and sampling strategy.

Author Response (AR) - We consider that this study presents a comprehensive analysis of environmental forcing upon the distribution and abundance of dominant diatoms and coccolithophores in the Great Calcite Belt, a region of high importance for marine biogeochemical cycles. This work will contribute to improve our knowledge of the factors that control the biogeography of phytoplankton in the Southern Ocean. It may well form a baseline for the standard of analysis required for future studies, in that they will require a comprehensive investigation over a wide suite of environmental data when considering phytoplankton biogeography in the Southern Ocean - there is a need to move beyond single factor analysis.

RC1 - This brings up the next point: The collections design was perhaps not ideal. In the paper (page 4 line 25) it says that water was collected from the upper 30 m. Apparently, only one liter of water was sampled that integrates the entire 30 m? This is really precious little water unless I am reading this incorrectly in which case it needs to be explained. It is well known that phytoplankton biomass can occur below this level (e.g. Hegseth and Sundfjord, 2008). Why was the collection limited to 30m?

AR1 - The focus of this study was on the upper mixed layer in the Southern Ocean, rather than deeper waters below the productive euphotic zone and noting that few subsurface chlorophyll maxima (SCM) were encountered (limited to sub-tropical waters). Sampling at 30 m is hence suitable for characterising variability in upper ocean phytoplankton communities. Sampling 1 litre of water is standard procedure for SEM identification of phytoplankton on a 25 mm filter area. Higher volumes lead to clogging of the filter and loss of useable filter area for enumeration when cells are covered in additional organic matter and/or other phytoplankton cells.

RC2 - Also the method of identification is not really suited to a detailed morphological analysis of E.hux which is important especially in the southern oceans where there

exist various morpho/phenotypes of this species.

AR2 - We acknowledge that there are various morphotypes of E. huxleyi in the Southern Ocean, however morphological examination of E. huxleyi was not performed as part of this study. Further, differentiating E. huxleyi morphotypes for the statistical analysis was not our specific focus, which was on differentiating different coccolithophore and small diatom species.

RC3 - What were the reasons for the magnifications differing at 5kx and 3kx?

AR3 - The difference in magnification for the two transects reflects the overall lower cell densities found in the Indian Ocean versus the Atlantic Ocean and our requirement to enable sufficient filter area for identification and enumeration.

RC4 - What about all the other material on the filter?

AR4 - There were occasionally other material present on the filter, but these were not straightforward to identify and therefore were not quantified. Additional material beyond coccolithophores and diatoms were not the focus of the study and so were not included in the manuscript.

RC - There are many generalities in the paper that could use more explanation. Some of these are defined by G below;

RC5 - G: Page 2 line 5: Takahashi wrote many papers on CO2 sequestration of CO2. How do we know whether the CO2 that is being taken up by areas of the ocean is anthropogenic or natural? Also the North Pacific is also such an area.

AR5 - We have included the North Pacific in this sentence as follows:

"The region between 30-50oS is recognized as having the highest uptake of anthropogenic carbon dioxide (CO2) alongside the North Atlantic Ocean and North Pacific Ocean (Sabine et al., 2004)."

Also, following back to the original work the anthropogenic uptake was estimated from

a carbon tracer technique (Gruber et al, 1996).

Gruber, N., Sarmiento, J. L., & Stocker, T. (1996). An improved method for detecting anthropogenic CO2 in the oceans. Global Biogeochemical Cycles, 10(4), 809–837.

RC6 - G: >Page 2 line 7-9: vague sentences. Poorly constrained, critical? Why

AR6 - We have rephrased this paragraph to read as follows:

"Our knowledge of the impact of interacting environmental influences on phytoplankton distribution in the Southern Ocean is limited, for example how light and iron availability or temperature and pH may interact to control phytoplankton biogeography (Boyd et al., 2010, 2012; Charalampopoulou et al., 2016). Hence, if model parameterizations are to improve (Boyd and Newton, 1999), and provide more accurate predictions of future biogeochemical change, a multivariate approach is required."

RC7 - G: >Page 2 line 28-30: Why important

AR7 - We have added context at the beginning of the paragraph that highlights the importance of studying mineralizing phytoplankton.

"In the context of climate change and future ecosystem function, the distribution of mineralizing phytoplankton is important to define when considering phytoplankton interactions with carbonate chemistry (e.g., Langer et al., 2006; Tortell et al., 2008) and ocean biogeochemistry (e.g., Baines et al., 2010; Assmy et al., 2013; Poulton et al., 2013)."

RC8 - Page 3 line 1...."south of ∼30oS and extends to ∼60o" (This has already been stated.

AR8 - This text has now been removed.

RC9 - G: Page 3 line 25 "uncertainties" why?

AR9 - To clarify we have altered the sentence to read as follows:

"... remains a significant issue when considering the impact of climate change on natural phytoplankton communities."

RC10 - Page 5 line 11: Why were individual coccoliths not counted? This can also say a lot about the age of the community.

AR10 - Our focus in the present study was on comparative biogeography of coccolithophores and small diatoms rather than coccolithophore growth dynamics. Hence coccolith counts were not included.

RC11 - Page 7 line 5-13. Can all these parameters be displayed graphically?

AR11 - These parameters could be displayed graphically, however, this would look confusing given the north-south and east-west cruise tracks and irregular distances covered between stations. It was decided that retaining the original data in table format also allowed better access to the parameter values.

RC12 - Page 7 line 29. Maybe I missed it but what were ALL of the 28 coccolithophore species?

AR12 - This information will be available as a Pangea dataset, combining the 28 coccolithophore species and 76 diatom species and their abundances.

RC13 - Page 11 line 28 "occurrences" instead of "features"

AR13 - This has now been altered.

RC14 - Page 12 line 7. Where is the rest of the Chl coming from? This section (lines 7-9) is not clear

AR14 - The remaining fraction of the Chl-a is most likely to represent phytoplankton not enumerated in this study such as small picoplankton, non-mineralising nanoplankton (e.g. naked flagellates), dinoflagellates and other diatoms.

RC15 - Page 13 line 4 "coccolithophores IN this region"

AR15 - We have noted this and corrected.

RC16 - Page 14 line 10 What is the "theoretical species abundance"?

AR16 - We have removed this comment and amended the sentence as follows.

"Nanoplankton are subject to high grazing pressure (Schmoker et al., 2013), with the growth and mortality of a species both directly influencing cell abundances (Poulton et al., 2010), which could result in nanoplankton abundance patchiness additional to the influence of temperature or other environmental gradients."

RC17 - Page 15 line 12. "However A FEW non-blooming

AR17 - We have noted this and corrected it.

RC18 - Page 15 line 13 "conspicuously absent" why is this conspicuous?

AR18 - We have removed conspicuously from this sentence.

RC19 - Page 15 line 14-15 "to be a E. huxleyi specific rather than a coccolithophore-wide phenomena" not clear what the authors meant to say. I don't agree.

AR19 - We have altered the sentence to read as follows

"Therefore, a low silicic acid concentration in the surface waters of the GCB may negatively impact coccolithophore species that do have a silicic acid requirement, such as Calcidiscus leptoporus, and favour bloom-forming species that do not require silicic acid such as E. huxleyi."

RC20 - Page 16 lines 25-27...There are many studies con and pro for this sentence

AR20 - We are not sure what the reviewer means here. We have removed part of the sentence for clarification.

"In our study, there was no significant correlation between E. huxleyi and $\Omega$calcite (Pearson's product moment = 0.093). However, the waters of the GCB remained over-saturated ($\Omega$calcite> 2) throughout, and furthermore the relationship between coccolithophores, calcification and carbonate chemistry is now recognized as being complex and non-linear. . ."

---

## Author Response (AR1)

Reviewer #1

Reviewer Comment (RC) - This article presents a comprehensive analysis of environmental forcing upon the distribution and abundance of dominant diatoms and coccolithophores in the Great Calcite Belt, a region of high importance for marine biogeochemical cycles. The study has been carefully conducted and the results are presented clearly and concisely. This work will contribute to improve our knowledge of the factors that control the biogeography of phytoplankton in the Southern Ocean. I support publication of this material in BG, provided the authors address some uncertainties in their analyses and conclusions.

Author Response (AR) - We thank the reviewer for their constructive comments and recommendation for publication pending our responses and further development of the manuscript. We address their comments below.

RC1 - Reading the description of BGC at the beginning of the Introduction, one may be tempted to infer that the biogeochemical importance of the GCB (e.g. a region of net CO2 uptake) stems from the fact that it is a region of high PIC. However, its importance is probably more related to its being a region of generally increased plankton abundance and productivity. In fact Fig. 1 suggests that the region could be equally defined in terms of enhanced chla levels.

AR1 - We are in agreement with the reviewer – the Great Calcite Belt is an area of both elevated chlorophyll-*a* and particulate inorganic carbon associated with increased seasonal production. The recent confirmation of the GCB as a significant coccolithophore phenomenon leads to this region being of interest in the context of upper ocean biogeochemistry and changing climate. Acknowledging the reviewers comment we have made the following changes to the introduction to better reflect the generalised increase in plankton abundance and productivity within the GCB, page 2 line 1-2.

*"The Great Calcite Belt (GCB), defined as an elevated particulate inorganic carbon (PIC) feature occurring alongside seasonally elevated chlorophyll-a in austral spring and summer in the Southern Ocean (Fig. 1; Balch et al., 2005), plays an important role in climate fluctuations…"*

RC2 - On a related note, is the PIC to POC ratio actually higher in this region than it is in tropical and subtropical waters?

AR2 - The GCB spatial extent is set by high satellite-detectable PIC concentrations rather than a change in the PIC to POC ratio – both PIC and POC may increase relative to subtropical waters to get the GCB signal, without necessarily changing the ratio between the two.

RC3 - Some studies have shown that the coccolithophore to diatom biomass ratio actually increases in tropical, unproductive waters (Cermeno et al. PNAS 2008).  This study uses

abundance to assess dominance of different phytoplankton species. But, due to interspecific differences in cell size, an assesment based on carbon biomass could have been more reliable, as some of the authors have shown before (Daniels et al. MEPS 2016). For instance, section 3.2 starts by noting that nanoplankton tended to be more abundant than microphytoplankton, but this is always to be expected and cannot be directly translated to ecological dominance patterns. The authors should include a statement, and/or provide some sensitivity tests, on how results could change if dominance were assessed by biomass instead of abundance.

AR3 - Indeed, considering biomass would most likely change the picture and decrease the dominance of coccolithophores (in many cases). Whilst, deriving coccolithophore biomass is relatively straightforward (as they are mostly spherical in shape, with no vacuoles or complex cell structures that may include biomass), diatoms are far more morphometrically complex (not spherical, often with setae which may or may not contain cell plasma, and many cells have large internal vacuoles), making direct comparison between the two potentially problematic (especially when the two may be equally abundant) – i.e. small errors in diatom estimates can cause species dominance to radically change. In contrast, comparisons across subtropical waters (no diatoms, some coccolithophores) and upwelling zones (many diatoms, few coccolithophores), as in Cermeno et al. (PNAS 2008), is relatively straightforward.

Furthermore, many potentially significant issues over carbon conversions are not straight forward. Although there are now extensive conversion tables for various phytoplankton carbon content, these come with important caveats, as described in detail (e.g.) in Leblanc et al. (2012), which include (but are not limited to) the effect of preservatives on cell size and content (shrinkage), simplistic bio-volume conversions from cell measurements, time of sampling and age of the community or population, and growth conditions (light, nutrients, temperature). The scope of our manuscript was not intended to cover full discussion of these issues.

We have now included at statement to show that we recognise the differences in species dominance if biomass was considered, page 7 line 22-24.

*"...not numerically dominant compared to the nanoplankton species at these locations. Consideration of community biomass would potentially reduce the dominance of the nanoplankton relative to microplankton in the GCB. However, converting cell size to biomass is not straightforward for diatoms, as highlighted in Leblanc et al. (2012), and to avoid these potential caveats we have considered species abundance only. Total cell abundances..."*

RC4 - page 12 line 20. A reference is needed here to support the value of chla content used for Ehux.

AR4 - We do state the appropriate references (i.e. Haxo, 1985 and Poulton et al., 2013 who applied these estimates previously) used to estimate the E. huxleyi chl-*a* contribution in section 4.2.1 page 12 line 22, and have now restated this in sections 4.2.2 (page 13 line 9) and 4.2.3 (page 13 line 25) to avoid confusion.

RC5 - However, the chla content of algal cells is highly dependent on temperature, light, nutrients, etc. which makes this calculation very uncertain. Carbon biomass is a more reliable metric to estimate relative importance of different species, because the C cellular content is less variable.

AR5 - Cell chl-*a* content is indeed variable with physiological growth conditions. Carbon biomass is possibly a more reliable metric, however this would rely on two requirements: (1) that the entire phytoplankton community (pico-plankton to micro-plankton) be assessed in terms of cell carbon (which few studies undertake), and (2) that there are few errors in estimates of cell carbon from cell size and biovolume (see earlier comment). Literature biovolume to carbon conversions are often generalist across multiple species and many (though not all) are based on culture values under optimum growth conditions rather than realistic in situ conditions (temperature, light, nutrients). Hence, there are also large potentials for cell carbon estimates to be as variable with physiological growth conditions than a rough conversion of cell numbers to chl-*a*. We have now added an appropriate caveat to the text to acknowledge potential issues over variable cell chl-*a* content and the estimates derived from them, page 12 line 26-28.

*"It should be noted that the cell Chl a content from Haxo (1985) falls at the lower end of the current range of measurements for E. huxleyi cell Chl a content (e.g., 0.24-0.38 pg Chl a per cell; Daniels et al., 2014) and leads to conservative estimates of Chl a contribution from this species."*

RC6 - The conclusion in the Abstract that temperature is the main driver of nanoplankton distribution should be qualified, as it may well be that temperature is co-varying with other factors that are the actual, ultimate drivers.

AR6 - We agree with the reviewer and have now rewritten the final line of the abstract to better reflect the results of the multivariate analysis.

*"Multivariate statistics identified a combination of carbonate chemistry and macro-nutrients, co-varying with temperature, as the dominant drivers of biomineralizing nanoplankton in the GCB sector of the Southern Ocean."*

RC7 - On p. 10 line 10, what is the basis for statement that nanophytoplankton contribute 40% of total PP? The references provided do not have that kind of evidence (they are reviews on the ecology and biogeochemical role of diatoms). The authors should use instead remote sensing studies (e.g. Uitz et al. 2010 GBC) to support the statement that nanophytoplankton are the largest contributors to global marine PP.

AR7 - We actually refer in the text to the micro-phytoplankton contribution, in order to highlight that the majority of studies in the Southern Ocean have focused on large phytoplankton species (i.e. most often diatoms). We have now inserted the Uitz et al. (2010)

reference in the relevant section to further highlight the contribution of micro-phytoplankton, but also the contribution of nano-phytoplankton, as discussed in the next sentence (starting p.10 line 25).

*"Studies of Southern Ocean phytoplankton productivity have generally focused on the micro-phytoplankton (Barber and Hiscock, 2006) as these species contribute around 40% to total oceanic primary production (Sarthou et al., 2005; Uitz et al., 2010). However, nanoplankton and picoplankton are becoming increasingly recognised as important contributors to total phytoplankton biomass, productivity and export in the Southern Ocean (e.g., Boyd, 2002; Uitz et al., 2010; Hinz et al., 2012)…"*

RC8 - Minor point 'TOxN' is awkward and seems to suggest organic nitrogen. Better use 'NOx' or just nitrate (indicating in methods that nitrate actually refers to nitrate+nitrite). In any event nitrite concentrations are likely to be negligible, in comparison with nitrate, in these waters.

AR8 - The notation for nitrate+nitrite has now been changed to NOx throughout the manuscript.

AR9 - Additional references used in responses

Cermeño, P., Dutkiewicz, S., Harris, R.P., Follows, M., Schofield, O. and Falkowski, P.G.. The role of nutricline depth in regulating the ocean carbon cycle. P. Natl. Acad. Sci. USA, *105*(51), 20344-20349, 2008.

Daniels, C.J., Sheward, R.M. and Poulton, A.J.. Biogeochemical implications of comparative growth rates of Emiliania huxleyi and Coccolithus species. Biogeosciences, *11*(23), 6915-6925, 2014.

Leblanc, K., Arístegui, J., Kopczynska, E., Marshall, H., Peloquin, J., Piontkovski, S., Poulton, A.J., Quéguiner, B., Schiebel, R., Shipe, R. and Stefels, J.. A global diatom database–abundance, biovolume and biomass in the world ocean. Earth Syst. Sci. Data, 4, 149-165, 2012.

Uitz, J., H. Claustre, B. Gentili, and Stramski D. Phytoplankton class-specific primary production in the world's oceans: Seasonal and interannual variability from satellite observations, Global Biogeochem. Cy., 24, GB3016, doi:*10.1029/2009GB003680*, 2010.

Reviewer #2

Reviewer Comment (RC) - This manuscript presents phytoplankton cell counts results from the Southern Ocean from two cruises conducted in the GCB (Great Calcite Belt) together with a number of environmental physico-chemical data that are merged in a statistical analyses to provide causalistic hypotheses to plankton community structure.
The main results of this manuscript are that: coccolithophores and diatoms co-occur in the studied area and that coccolithophores in particular extend very far South, that community structure is mainly driven by four reprensentative of the nanoplankton group (3 diatoms, 1 coccolithophore), that the key drivers of community structure are both T° and Si depletion which create different ecological niches.

Author Response (AR) - We thank the reviewer for their thorough review of the manuscript. We address their comments below

RC - Overall, I find the methods, results and main conclusions presented here are quite weak, with two main criticism:

**RC 1 - My first and main concern regards the phytoplankton cell counts.** I find that the method used for cellular abundance determination is not a very robust nor trustable method. Counting very small area of filtered samples in SEM is not usual for nano- or microphytoplankton determination. From what the authors indicate in their method section, I deducted that sample cell counts were determined on only 2 ml sample, which is insufficient in most cases to provide statistically robust results. If I agree with the authors general recommendation to use both SEM and light microscopy in parallel, it should be to count cell numbers in light microscopy on a sufficient volume (50-100 ml usually) and use SEM to improve species determination, and not the other way around. I don't understand why lugol/formol fixed samples were not collected or analyzed here. My second concern is on the large bias towards small species that this method implies, as correctly identified by the authors themselves. The main statement here about nanophytoplankton dominating the mineralizing algae is not trustworthy when large cells can not correctly be assessed by this method. The authors mispelled on several occasions diatom names, and include *Pseudo-nitzschia* sp. within the nanoplankton size-class which is quite surprising, as this species is most typically much larger than 20 μm, as can be seen very easily in figure 4. Also Figure 4 reveals very interestingly that a number of Parmales were present, they are part of the pico-nano- size fraction of siliceous plankton, so I find very surprising that no mention was made of that in the manuscript.

AR1 - Following these comments, we have identified and respond to the following points:
1. SEM counting of nano- and micro-plankton versus Light Microscopy: there have been several studies using SEM techniques to count coccolithophores and other nano-plankton, for example Mohan et al. (2008), Cubillos et al. (2007), Leblanc et al. (2009), Hinz et al. (2012) and Charalampopoulou et al. (2011). Though we do acknowledge that using SEM for enumeration is not typical for studying micro-phytoplankton communities

(exclusively), our focus is the small diatoms not typically identified by light microscopy. Furthermore, we also aim to put the mineralising nanoplankton in the wider context of the phytoplankton community. To better reflect this we have now amended the manuscript to make this clearer throughout.

2. Limited volume (2 mL) examined: We fully understand the reviewers concerns in terms of the statistically robustness of the count results (though our methods match those listed above). In our study, our pre-treatments of the data before multivariate analysis specifically aim to avoid any potential issues that may arise from low sampling resolution of the species composition of the community. Specifically, we have removed species with low cell densities (in our study < 1 cell mL$^{-1}$) to remove their potentially random influence on the multivariate statistics. We have also standardised our count data (converted to percentage relative abundances) and performed a square-root transformation of the relative abundances to reduce the influence of potential count bias (at both ends of the abundance spectra) on the end results.

3. The cell size of the diatom *Pseudo-nitzschia*: The initial definitions of size-fractions of phytoplankton were based on mesh sizes of plankton nets. In the case of the *Pseudo-nitzschia* in our study, its size affiliation depends on whether one considers its length (30-50 µm) or its width (2-5 µm). In recognition of the point of the reviewer we have now altered the revised manuscript to make it clear that in this case we have considered *Pseudo-nitzschia* to be at the small end of the micro-phytoplankton group.

4. No mention of the Parmales: The focus in the original manuscript was not on the rarer nanoplankton and hence we chose not to mention them. *Tetraparma sp.* were particularly abundant at only one station, where they were present at a cell density of 2000 cells mL$^{-1}$, and present in low numbers (< 5 cells mL$^{-1}$) at three more stations in the South Atlantic, whilst they were absent throughout the rest of our sampling of the GCB. We have now added this information to the revised manuscript (see page 7 line 31).

RC2 - SEM observations should also have allowed species determination for the dominant *Pseudo-nitzschia* species, which is not indicated. This suggests an overall lack of expertise for diatoms, and that calcifying algae were initially the focus of the study and that diatoms were only added lately to the analysis.

AR2 - The SEM images could have allowed for species-specific determination of the *Pseudo-nitzschia*, however the resolution and collapsed nature of the cells after filtration (i.e. they were weakly silicified species) was not adequate for high-resolution taxonomic identification. Reliable species-level taxonomic identification on all cells (or a representative majority) in all samples was also not feasible, and so we chose to retain identification at the genus level.

RC3 - I have a hard time believing the low species numbers (1-3) indicated for diatoms at certain stations.

AR3 - We apologise for the slight mistake or mis-understanding in Table 2. In the original version of the manuscript Table 2 presented the post-transformed species data (i.e. the counts minus the rare species prior to multivariate statistical analysis). We have now altered Table 2

to reflect the number of species identified prior to transformation of the data (i.e. removal of the rare species).

RC4 - Another point is the presentation of cellular abundance only. This is absolutely not the best metric to compare with physico-chemical parameters, and C biomass conversions are absolutely needed in this kind of data analysis. This would have allowed a relative estimation of the contribution of mineralizing algae to total POC (or Chla stretching it with POC:Chla ratios) and more robust conclusions regarding the real importance of both coccolithophores and nano-sized diatoms in total phytoplankton summer blooms.

AR4 - Indeed, a comparison of cell biomass from all species and phytoplankton groups would be the most comprehensive comparison. This would need to include all pico-plankton, nano-plankton and micro-plankton, which are not often all enumerated or reliably measured in terms of biomass. There are also issues (for each group), as described in detail (e.g.) in Leblanc et al. (2012), in terms of carbon conversions (from bio-volume or cell sizes), including preservation effects on cell size, variable cell sizes with growth conditions and nutritional strategies (autotrophic or mixotrophic).

Whilst deriving coccolithophore biomass is relatively straightforward (as they are mostly spherical in shape, with no vacuoles or complex cell structures that may include biomass), diatoms are far more morphometrically complex (not spherical, often with setae which may or may not contain cell plasma, and many cells have large internal vacuoles), making direct comparison between the two potentially problematic (especially when the two may be equally abundant) – i.e. small errors in diatom estimates can cause species dominance to radically change.

We have now included at statement to show that we recognise the differences in species dominance if biomass was considered, page 7 line 22-24.

*"...not numerically dominant compared to the nanoplankton species at these locations. Consideration of community biomass would potentially reduce the dominance of the nanoplankton relative to microplankton in the GCB. However, converting from cell size to biomass is not straightforward for diatoms, as highlighted by Leblanc et al. (2012), and to avoid such issues we consider species abundance only. Total cell abundances…"*

The suggestion to use comparison to POC, which includes a variable proportion of detrital material, bacteria and zooplankton, would seem to only compound issues over representativeness of the comparisons. Lastly, other previous studies have done the same type of comparisons as presented here; e.g. Kopczynska et al., 1986, Cefarelli et al., 2011, Chen et al., 2007, Hinz et al., 2012, Charalampopoulou et al., 2016.

**RC5 - My second main concern is about the statistical analyses.** Although I will frankly admit that I am not qualified to expertise the tests presented here further than simple correlation matrixes, I really miss the added value of such extensive statistical tests.

Quantifying so many environmental variables (such as carbonate chemistry which is very tricky) to collapse them in the end with T° and nutrients seem very odd to me. Finally, every bit of conclusion about the different phytoplankton communities and the overarching role of T° and silicic acid could have been stated by directly looking at the data and the statistics provided here are not at all convincing.

AR5 - The added value of such an extensive statistical test is that the ocean is not univariate, environmental factors vary at the same time, occasionally in the same direction or in a linear fashion (but not always) and a simple correlation matrix completely ignores the importance of a multivariate perspective on phytoplankton ecology. Our analysis also has no a priori assumptions in terms of driving factors and allows the data to identify the key correlating parameters. This is why the environmental variables collapse down to a limited number of factors. Making the conclusion reached in this study by solely looking at the data, with no attempt to statistically examine or balance the significance of the relationships found, goes against our approach to this type of research. In light of the reviewers comments we have now added text directing the reader as to why each statistical test is included (see Section 3.3 and specific response to Page 9 Line 14) to ensure that the importance of such extensive statistical techniques is made much clearer.

RC6 - The discussion section leaves much to be desired and is a succession of short paragraphs that are very counter-intuitively organized and that should be entirely rewritten. A number of other papers regarding the succession patterns of coccolithophores and diatoms elsewhere are ignored.

AR6 - We are not sure exactly what the reviewer means here by 'counter-intuitively organised'. We have ordered the discussion to reflect the order of the results and tailored the discussion from general trends towards more specific areas of interest that were highlighted by the statistical results. We are also not sure which papers the reviewer is referring to, but do recognise that our focus tends to be on Southern Ocean publications rather than ones from the northern hemisphere.

RC - I have several other comments/corrections/questions that are added as sticky notes in the manuscript pdf attached.

AC - Comments from sticky notes:
RC7 - Page 1 Line 26 – Spelling Pseudonitzschia to Pseudo-nitzschia (and thereafter within document)

AR7 - Thank you for highlighting this error in spelling of the diatom genus Pseudo-nitzschia – this has been amended throughout.

RC8 - Page 2 Line 15 - What about non mineralizing nanoplankton? Are they important?

AR8 - Non-mineralizing phytoplankton are important within the context of the overall function of the oceanic ecosystem and carbon export. However, the focus of this paper was to assess the distribution of the coccolithophores and diatoms in the Great Calcite Belt. As biomineral providers, the biogeographical distribution of mineralising phytoplankton species are of great interest when it comes to the resulting carbon export and surface ocean biogeochemistry.

RC9 - Page 2 Line 20 - Pseudo-nitzschia are very seldom <20µm. In your figure 4d, they are about 60 µm if scale bar is correct - or 150 µm if your legend is correct. I would not include them in the nanoplankton group.

AR9 – Now page 2 line 22 - We have removed the size classification from the sentence to avoid confusion about size classes of diatom species.

*"North of the PF, small diatom species (e.g. Pseudonitzschia sp. and Thalassiosira sp.) tend to dominate numerically, whereas large diatoms with higher silicic acid requirements (e.g. Fragilariopsis kerguelensis)…"*

RC10 - Page 4 Line 26 - bizarre annotation. NOx ? or DIN are more standard

AR10 - We have altered the annotation to NOx throughout the manuscript

RC11 - Page 5 Line 15 - I don't understand this sentence. Was a 200 µm mesh placed beneath the 0.8 µm filter on the filtration rig?

AR11 – Now page 5 line 12 - Apologies if this was not clear, the 200 µm mesh was placed beneath the 0.8 µm filter. The sentence has been rewritten as follows.

*"Seawater samples were gently filtered through a 25 mm, 0.8 µm Whatman® polycarbonate filter placed over a 200 µm backing mesh to ensure an even distribution of cells across the filter."*

RC12 - Page 5 Line 20 - this is only 1/500 of the surface of a 25 mm filter, this seems to be very little (equivalent to 2 ml of sample counted).

AR12 - Yes, this is a small surface area and equivalent volume, and does have its limitations (as with every sampling or analytical method). We have followed a standard method for enumerating phytoplankton from SEM images and statistically analysing species distributional patterns as applied in (e.g.) Charalampopoulou et al. (2011).

RC13 - Page 6 Line 1 - I am not qualified to review the robustness of the statistical analyses used in this paper

AR13 - We appreciate that unfamiliarity with multivariate statistics has not made this possible for the reviewer. We have endeavoured to make the statistics section as reader-friendly as possible to aid those unfamiliar with this type of statistical approach.

RC14 - Page 6 Line 30 - Date of sampling could have been included in this table (Table 1)

AR14 - We agree with the reviewer and have inserted the date of sampling into Table 1

RC15 - Page 7 Line 1 - use µM for nutrients

AR15 - We have amended to µM throughout the paper, tables and figures.

RC16 - Page 7 Line 16 - again I think this is potentially very biased if only fractions of SEM filters were analyzed and if no larger water volumes were counted.
Also, this kind of assertion needs to be substantiated by biomass estimates. Picoplankton abundance is most frequently always > nanoplankton > microplankton, but cell abundance conversion to C biomass often reverses these orders. Links with nutrient and light availability should preferably be considered with biomass rather than abundance.

AR16 - Please see response to main comments.

RC17 - Page 8 Line 11 - correct spp. (and occurrences thereafter)

AR17 - Thank you for bringing this to our attention, this was amended where necessary.

RC18 - Page 8 Line 25 - I understand the general assumption here, but it seems very strange to go through all this trouble measuring all parameters and C chemistry, which is tedious, just to collapse everything with NO3 and T° as explanatory variables in the end.

AR18 - Please see earlier comment regarding statistical analysis. The highly dynamic nature of the Southern Ocean requires a more robust approach to analysis. We felt it was best to start with the greatest range of parameters that may influence phytoplankton biogeography, and then let the statistical analysis determine significant patterns and correlations.

RC19 - Page 9 Line 14 - How is that different from the SIMPROF routine and Fig 3?

AR19 - Apologies if this is not clear in the text, we have altered the text to make this clearer. In short, the SIMPROF test statistically identifies groups of samples with more similar community structure, whilst the SIMPER test statistically identifies the specific-species that define these groups.

Page 9 Line 14 *"The SIMPROF routine identified the stations in the GCB that had statistically similar coccolithophore and diatom community composition through a comparison of Bray-Curtis similarities."*

Page 9 Line 23 *"A SIMPER routine statistically identified the species that define the difference between (and similarity within) the statistically different community structures defined by the SIMPROF routine (Table 4)."*

RC20 - Page 10 Line 13 - Agreed. This is why I find regrettable a better job was not done on accurate quantification of all size-classes, together with C conversions. Also, from Fig 4, the siliceous armored Parmales, which are spanning over the pico-nano size fractions are present, too bad they were not quantified. That would have strengthened this argument, and brought some new insights to SO communities.

AR20 - The *Tetraparma sp.* were only particularly abundant (2000 cells mL-1) at one station, whilst they were in limited numbers (<5 cells mL-1) at three more stations in the South Atlantic and absent across the rest of the GCB. Hence we do not think that addition of these counts would add to the statistical analysis. We have now added this information to the results section page 7 line 31.

RC21 - Page 10 Line 27 - I really disagree about Pseudo-nitzschia being part of the nanoplankton. They are very rarely <20 μm, and definitely much larger than that in your figure 4, no matter which scale is used (the figure's or the legend's which differ).

AR21 – Now page 11 line 11-13 - Apologies if this sentence is unclear, we have rephrased it to make clear that we do not include Pseudo-nitzschia in the nanoplankton class.

*"Three of these species (E. huxleyi, F. nana and F. pseudonana) are part of the nanoplankton, whilst Pseudo-nitzschia sp. is at the lower end of the size range of the microplankton (Pseudonitzschia sp. is >20 μm in length but <5 μm in width)…"*

RC22 - Page 11 Line 2 - Again this argument falls short, when 1 of the 4 species is not attributed to its correct size class, and when accurate cell abundance determinations of the microplankton size class were not made. I have a hard time believing the very low species numbers given for diatoms in Table 2 (between 1 and 3) at several sites.

AR22 - We have now amended Table 2 to reflect the number of species identified in the sample pre-statistical analysis.

RC23 - Page 11 Line 3 - correct nitzschioides (and occurrences thereafter)

AR23 - Thank you for highlighting this spelling mistake, the spelling has been altered

RC24 - Page 11 Line 9 - most certainly

RC25 - Page 11 Line 15 - If I totally agree with this recommendation, I feel that it should be reversed. Cell counts need to be made in fixed water samples, while correct species determination can be made using SEM, but not the other way around.

AR25 - This would provide a thorough analysis of the micro-plankton, however nanoplankton are rarely observed in light microscopy or accurately enumerated.

RC26 - Page 12 Line 30 - similar studies conducted in the North Atlantic could be cited here.

AR26 – Now page 13 line 19 - We have now included reference to Leblanc et al. (2009).

RC 27 - Page 15 Line 3 - this argument is unclear to me

AR27 – Now page 15 line 20 - We have altered the sentence for further clarification as follows.

*"...so the high abundance of F. nana in the high silicic acid waters could be indicative of a seasonal progression driven by light and/or temperature rather than silicic acid dependence."*

RC 28 - Page 15 Line 10 - this was also described in Leblanc et al. 2009

AR28 -  Now page 15 line 28 - This reference has now been incorporated into the sentence.

"… has also been identified in the Scotia Sea (Hinz et al., 2012) and the Patagonian Shelf (Balch et al., 2014) in the Southern Ocean, as well as in the North Atlantic (Leblanc et al., 2009).

RC 29 - Page 15 Line 15 – "Therefore the positive selection pressure at low silicic acid concentrations in the GCB is likely to be *E. huxleyi* 15 specific rather than a coccolithophore-wide phenomena."  Why not?

AR29 – Now page 15 line 31 - We have altered the sentence to read and explain better as follows:

*"Therefore, low silicic acid in surface waters of the GCB may negatively impact coccolithophore species that have a silicic acid requirement, such as Calcidiscus leptoporus, and favour bloom-forming species that have no silicic acid requirement (e.g., E. huxleyi)."*

RC30 - Page 15 Line 25 - Fig. 6

AR30 – Now page 16 line 12 - This has now been amended.

RC31 - Page 16 Line 15 - I would consider pCO2 being the result of phytoplankton bloom development, rather than its driver.

AR31 - In general we agree with the reviewer in the context of temporal changes, however our study has little temporal context (and was carried out in summer).

RC32 - Page 17 Line 6 – "…suggest that four nanoplankton..." Three

AR32 – Now page 17 line 23 - As suggested we have changed this to:

"...suggest that three nano- (<20 μm) and one micro- (>20 μm) phytoplankton species…"

RC33 - Page 17 Line 9 - estimated by cell/chla ratio conversions rather?

AR33 – Now page 17 line 26 As suggested we have changed this to:

"as estimated from cell counts and Chl a"

RC34 - Page 17 Line 13 - I don't find that this is properly demonstrated through similar estimations of cocco and diatom biomass or Chla contributions

AR34 – Now page 17 line 20 - We have re-written as follows to remove the direct comparison to diatoms.

"This indicates that in the post-spring bloom conditions of the GCB, E. huxleyi is an important contributor to phytoplankton biomass and primary production at localized spatial scales."

While there are no other estimations of coccolithophore and diatom biomass in this study, for reasons described in earlier comments, a conservative estimate indicating that E. huxleyi contributes up to 20% of the measured Chl a, does imply that this species is important within the overall phytoplankton community, even if at very local spatial scales and short time scales.

RC35 - Page 17 Line 14 - All right, this could have been hypothesized even before sample collection.

AR35 – Now page 17 line 23-25 - We have changed the emphasis of this sentence:

"Out of a wide suite of environmental variables, latitudinal gradients in temperature, macro-nutrients, $pCO_2$ and $\Omega_{calcite}$ 'best' described statistically the variation of phytoplankton community composition in this study, whereas $\bar{E}_{MLD}$ and pH did not rank as significant factors influencing species composition."

RC36 - Page 27 Line 11 - **** only one species present: this is highly unusual for diatoms, even though close to monospecific abundance can be noted. Probably an artefact linked to the small area of filter analyzed again.

AR36 - We have altered Table 2 to include the number of all species identified. Given that only a few cells of some species were identified in the area imaged, please note that we excluded these species from the statistical analysis given the uncertainty involved estimating abundance from a single cell.

RC37 - Page 27 Line 13 - so why is the dominant species at 100% when the **** code is given for coccos but not for diatoms ? this does not make sense with legend for instance at GCB1-46, S for diatoms =1, there is a ****, but then it is indicated that Chaetoceros represents 56% of diatoms ? this occurs again on the other two lines with ****

AR37 - Thank you for highlighting this discrepancy, Table 2 has been altered to include all species observed in the sample. Please see comment above.

RC38 - Page 33 Fig 4 c - I see quite a few Parmales in grey in this picture. They are beginning to be considered as abundant in the SO, did you not count them? They are part of the biomineralizing algae...

AR38 - See previous response to comment regarding Page 10, line 13. In short, yes they were counted and were abundant (2000 cells mL-1) at only one station and in limited abundance (<5 cells mL-1) at only three others, a comment has been inserted at page 7 line 31.

RC39 - Page 33 Fig 4 d - the scale bar says 2 μm, your legend says 5 μm, please correct. Also correct Pseudo-nitzschia

AR39 - Thank you for identifying this error.  The scale bar is correct, the text in the figure caption has now been removed.

RC40 - Page 34 Fig 5 -  I have a very hard time understanding the utility (and meaning) of this figure

AR40 - Figure 5 is included to visually represent how the specific phytoplankton species (and genus) play a role in defining the statistically different phytoplankton communities. We have now made this clearer in the main text and figure legend, see page 10 line 1-3.

RC50 - Based on these comments, I suggest either rejection or major revisions including entirely reworking both the dataset and its subsequent analysis.

AR51 - References referred to in the responses

Cefarelli, A.O., Vernet, M. and Ferrario, M.E.. Phytoplankton composition and abundance in relation to free-floating Antarctic icebergs. Deep-Sea Res. Pt II, 58(11), 1436-1450, 2011.

Charalampopoulou, A., Poulton, A. J., Tyrrell, T. and Lucas, M. I.: Irradiance and pH affect coccolithophore community composition on a transect between the North Sea and the Arctic Ocean, Mar. Ecol-Prog. Ser., 431, 25–43, doi:10.3354/meps09140, 2011.

Charalampopoulou, A., Poulton, A. J., Bakker, D. C. E., Stinchcombe, M., Lucas, M. I. and Tyrrell, T.: Environmental drivers of coccolithophore abundance and cellular calcification across Drake Passage (Southern Ocean), Biogeosciences 13, 5917-5935, doi:10.5194/bg-13-5917-5935, 2016.

Chen, Y.L.L., Chen, H.Y. and Chung, C.W.. Seasonal variability of coccolithophore abundance and assemblage in the northern South China Sea. Deep-Sea Res. Pt II, *54*(14), 1617-1633, 2007.

Cubillos, J., Wright, S., Nash, G., de Salas, M., Griffiths, B., Tilbrook, B., Poisson, A and Hallegraeff, G.: Calcification morphotypes of the coccolithophorid *Emiliania huxleyi* in the Southern Ocean: changes in 2001 to 2006 compared to historical data, Mar. Ecol-Prog. Ser., 348, 47–54, doi:10.3354/meps07058, 2007.

Hinz, D. J., Poulton, A. J., Nielsdóttir, M. C., Steigenberger, S., Korb, R. E., Achterberg, E. P. and Bibby, T. S.: Comparative seasonal biogeography of mineralising nannoplankton in the Scotia Sea: *Emiliania huxleyi*, Fragilariopsis spp. and Tetraparma pelagica, Deep-Sea Res. Pt II, 59–60, 57–66, doi: 10.1016/j.dsr2.2011.09.002, 2012.

Kopczynska, E. E., Weber, L. H. and El-Sayed, S. Z.: Phytoplankton species composition and abundance in the Indian Sector of the Antarctic Ocean, Polar Biol., 6, 161–169, 1986.

Leblanc, K., Hare, C. E., Feng, Y., Berg, G. M., DiTullio, G. R.,Neeley, A., Benner, I., Sprengel, C., Beck, A., Sanudo-Wilhelmy,S. A., Passow, U., Klinck, K., Rowe, J. M., Wilhelm, S. W.,Brown, C. W., and Hutchins, D. A.. Distribution of calcifyingand silicifying phytoplankton in relation to environmental andbiogeochemical parameters during the late stages of the 2005 North East Atlantic Spring Bloom, Biogeosciences, 6, 2155–2179, doi:10.5194/bg-6-2155-2009, 2009.

Leblanc, K., Arístegui, J., Kopczynska, E., Marshall, H., Peloquin, J., Piontkovski, S., Poulton, A.J., Quéguiner, B., Schiebel, R., Shipe, R. and Stefels, J., 2012. A global diatom database–abundance, biovolume and biomass in the world ocean. Earth Syst. Sci. Data, 4, 149-165, 2012

Mohan, R., Mergulhao, L. P., Guptha, M. V. S., Rajakumar, A., Thamban, M., AnilKumar, N., Sudhakar, M. and Ravindra, R.: Ecology of coccolithophores in the Indian sector of the Southern Ocean, Mar. Micropaleontol., 67(1–2), 30–45, doi:10.1016/j.marmicro.2007.08.005, 2008.

Reviewer #3

Reviewer Comment (RC) - This work adds incremental knowledge about the environmental forcing of coccolithophores and diatoms distribution in the southern ocean. In the future, the importance of this work may be that it serves as a base line study. The paper is well written and substantial with many references, although I believe it could be shortened by about 25% and still say the same.
Based on the abstract and conclusions there is not much new insight except that the authors are looking at diatoms and coccolithophores at the same time. There is a host of environmental data and they are discussed at length but few significant patterns emerge which is often the case in beyond control "ships of opportunity studies" where the research is constrained by circumstances, timing, and sampling strategy.

Author Response (AR) - We consider that this study presents a comprehensive analysis of environmental forcing upon the distribution and abundance of dominant diatoms and coccolithophores in the Great Calcite Belt, a region of high importance for marine biogeochemical cycles. This work will contribute to improve our knowledge of the factors that control the biogeography of phytoplankton in the Southern Ocean. It may well form a baseline for the standard of analysis required for future studies, in that they will require a comprehensive investigation over a wide suite of environmental data when considering phytoplankton biogeography in the Southern Ocean - there is a need to move beyond single factor analysis.

RC1 - This brings up the next point:
The collections design was perhaps not ideal. In the paper (page 4 line 25) it says that water was collected from the upper 30 m. Apparently, only one liter of water was sampled that integrates the entire 30 m? This is really precious little water unless I am reading this incorrectly in which case it needs to be explained. It is well known that phytoplankton biomass can occur below this level (e.g. Hegseth and Sundfjord, 2008). Why was the collection limited to 30m?

AR1 - The focus of this study was on the upper mixed layer in the Southern Ocean, rather than deeper waters below the productive euphotic zone and noting that few subsurface chlorophyll maxima (SCM) were encountered (limited to sub-tropical waters). Sampling at 30 m is hence suitable for characterising variability in upper ocean phytoplankton communities. Sampling 1 litre of water is standard procedure for SEM identification of phytoplankton on a 25 mm filter area. Higher volumes lead to clogging of the filter and loss of useable filter area for enumeration when cells are covered in additional organic matter and/or other phytoplankton cells.

RC2 - Also the method of identification is not really suited to a detailed morphological analysis of *E.hux* which is important especially in the southern oceans where there exist various morpho/phenotypes of this species.

AR2 - We acknowledge that there are various morphotypes of *E. huxleyi* in the Southern Ocean,

however morphological examination of *E. huxleyi* was not performed as part of this study. Further, differentiating *E. huxleyi* morphotypes for the statistical analysis was not our specific focus, which was on differentiating different coccolithophore and small diatom species.

RC3 - What were the reasons for the magnifications differing at 5kx and 3kx?

AR3 - The difference in magnification for the two transects reflects the overall lower cell densities found in the Indian Ocean versus the Atlantic Ocean and our requirement to enable sufficient filter area for identification and enumeration.

RC4 - What about all the other material on the filter?

AR4 - There were occasionally other material present on the filter, but these were not straightforward to identify and therefore were not quantified. Additional material beyond coccolithophores and diatoms were not the focus of the study and so were not included in the manuscript.

RC - There are many generalities in the paper that could use more explanation. Some of these are defined by G below;
RC5 - G: Page 2 line 5: Takahashi wrote many papers on CO2 sequestration of CO2. How do we know whether the CO2 that is being taken up by areas of the ocean is anthropogenic or natural? Also the North Pacific is also such an area.

AR5 - We have included the North Pacific in this sentence as follows:

*"The region between 30-50oS has the highest uptake of anthropogenic carbon dioxide ($CO_2$) alongside the North Atlantic and North Pacific Oceans (Sabine et al., 2004)."*

Also, following back to the original work the anthropogenic uptake was estimated from a carbon tracer technique (Gruber et al, 1996).

Gruber, N., Sarmiento, J. L., & Stocker, T. (1996). An improved method for detecting anthropogenic CO2 in the oceans. *Global Biogeochemical Cycles*, *10*(4), 809–837.

RC6 - G: >Page 2 line 7-9: vague sentences. Poorly constrained, critical? Why

AR6 – Page 2 line 6-11 - We have rephrased this paragraph to read as follows:

*"Our knowledge of the impact of interacting environmental influences on phytoplankton distribution in the Southern Ocean is limited. For example, we do not yet fully understand how*

*light and iron availability, or temperature and pH, may interact to control phytoplankton biogeography (Boyd et al., 2010, 2012; Charalampopoulou et al., 2016). Hence, if model parameterizations are to improve (Boyd and Newton, 1999) to provide more accurate predictions of future biogeochemical change, a multivariate understanding of the full suite of environmental drivers is required."*

RC7 - G: >Page 2 line 28-30: Why important

AR7 – Page 2 line 30-32 - We have added context at the beginning of the sentence that highlights the importance of studying mineralizing phytoplankton.

*"In the context of climate change and future ecosystem function, the distribution of biomineralizing phytoplankton is important to define when considering phytoplankton interactions with carbonate chemistry (e.g., Langer et al., 2006; Tortell et al., 2008) and ocean biogeochemistry (e.g., Baines et al., 2010; Assmy et al., 2013; Poulton et al., 2013)."*

RC8 - Page 3 line 1...."south of ~30oS and extends to ~60o" (This has already been stated.

AR8 - This text has now been removed.

RC9 - G: Page 3 line 25 "uncertainties" why?

AR9 – Page 3 line 24 To clarify we have altered the sentence to read as follows:

*"... remains a significant issue when considering the impact of future climate change."*

RC10 - Page 5 line 11: Why were individual coccoliths not counted? This can also say a lot about the age of the community.

AR10 - Our focus in the present study was on comparative biogeography of coccolithophores and small diatoms rather than coccolithophore growth dynamics. Hence coccolith counts were not included.

RC11 - Page 7 line 5-13. Can all these parameters be displayed graphically?

AR11 - These parameters could be displayed graphically, however, this would look confusing given the north-south and east-west cruise tracks and irregular distances covered between stations. It was decided that retaining the original data in table format also allowed better access to the parameter values.

RC12 - Page 7 line 29. Maybe I missed it but what were ALL of the 28 coccolithophore species?

AR12 - This information will be available as a Pangea dataset, combining the coccolithophore diatom species and their abundances.
https://doi.org/10.1594/PANGAEA.879790

RC13 - Page 11 line 28 "occurrences" instead of "features"

AR13 - Now on Page 12 line 13 This has now been altered.

RC14 - Page 12 line 7. Where is the rest of the Chl coming from? This section (lines 7-9) is not clear

AR14 – Now page 12 line 24 - The remaining fraction of the Chl-a is most likely to represent phytoplankton not enumerated in this study such as small picoplankton, non-mineralising nanoplankton (e.g. naked flagellates), dinoflagellates and other diatoms. We have included a statement to clarify this point.

*"This estimate is similar to that estimated in an identical way by Poulton et al. (2013) and highlights the significant contribution of phytoplankton other than coccolithophores (flagellates, diatoms) to phytoplankton biomass and production during coccolithophore blooms."*

RC15 - Page 13 line 4 "coccolithophores IN this region"

AR15 – Now page 13 line 22 We have noted this and corrected.

RC16 - Page 14 line 10 What is the "theoretical species abundance"?

AR16 – Now page 14 line 25 We have removed this comment and amended the sentence as follows.

*"Nanoplankton are subject to high grazing pressure (Schmoker et al., 2013), with the growth and mortality of a species both directly influencing cell abundances (Poulton et al., 2010), which could result in nanoplankton abundance patchiness additional to the influence of temperature and/or other environmental gradients."*

RC17 - Page 15 line 12. "However A FEW non-blooming

AR17 – Now page 15 line 30 We have noted this and corrected it.

RC18 - Page 15 line 13 "conspicuously absent" why is this conspicuous?

AR18 – Now page 15 line 32 - We have removed conspicuously from this sentence.

RC19 - Page 15 line 14-15 "to be a *E. huxleyi* specific rather than a coccolithophore-wide phenomena" not clear what the authors meant to say. I don't agree.

AR19 – Now page 15 line 30 - We have altered the sentence to read as follows

*"Therefore, a low silicic acid concentration in the surface waters of the GCB may negatively impact coccolithophore species that do have a silicic acid requirement, such as Calcidiscus leptoporus, and favour bloom-forming species that have no silicic acid requirement (e.g  E. huxleyi)."*

RC20 - Page 16 lines 25-27...There are many studies con and pro for this sentence

AR20 – Now page 17 line 12-14 We are not sure what the reviewer means here. We have removed part of the sentence for clarification.

*"In our study, there was no significant correlation between E. huxleyi and Ωcalcite (Pearson's product moment = 0.093). However, the waters of the GCB remained oversaturated (Ωcalcite> 2) throughout, and furthermore the relationship between coccolithophores, calcification and carbonate chemistry is now recognized as being complex and non-linear…"*

**The Influence of Environmental Variability on the Biogeography of Coccolithophores and Diatoms in the Great Calcite Belt**

Helen E. K. Smith[1,2], Alex J. Poulton[1,3], Rebecca Garley[4], Jason Hopkins[5], Laura C. Lubelczyk[5], Dave T. Drapeau[5], Sara Rauschenberg[5], Ben S. Twining[5], Nicholas R. Bates[2,4], William M. Balch[5]

[1]National Oceanography Centre, European Way, Southampton, SO14 3ZH, U.K.
[2]School of Ocean and Earth Science, National Oceanography Centre Southampton, University of Southampton Waterfront Campus, European Way, Southampton, SO14 3ZH, U.K.
[3]Present address: The Lyell Centre, Heriot-Watt University, Edinburgh, EH14 4AS7JG, U.K.
[4]Bermuda Institute of Ocean Sciences, 17 Biological Station, Ferry Reach, St. George's GE 01, Bermuda.
[5]Bigelow Laboratory for Ocean Sciences, 60 Bigelow Drive, P.O. Box 380, East Boothbay, Maine 04544, USA.

*Correspondence to*: Helen E.K. Smith (helen.eksmith@gmail.com)

**Abstract.** The Great Calcite Belt (GCB) of the Southern Ocean is a region of elevated summertime upper ocean calcite concentration derived from coccolithophores, despite the region being known for its diatom predominance. The oOverlap of two major phytoplankton groups, coccolithophores and diatoms, in the dynamic frontal systems characteristic of this region, provides an ideal setting to study environmental influences on the distribution of different species within these taxonomic groups. Water sSamples for phytoplankton enumeration were collected from the upper mixed layer (~30 m) during two cruises, the first to the South Atlantic sector (Jan-Feb 2011; 60$^o$ W-15$^o$ E and 36-60$^o$ S) and the second in the South Indian sector (Feb-Mar 2012; 40-120$^o$ E and 36-60$^o$ S). The species composition of coccolithophores and diatoms was examined using scanning electron microscopy at 27 stations across the Sub-Tropical, Polar, and Sub-Antarctic Fronts. The influence of environmental parameters, such as sea-surface temperature (SST), salinity, carbonate chemistry (i.e., pH, partial pressure of $CO_2$ ($p$CO$_2$), alkalinity, dissolved inorganic carbon), macro-nutrients (i.e., nitrate+nitrite, phosphate, silicic acid, ammonia), and mixed layer average irradiance, on species composition across the GCB, was assessed statistically. Nanophytoplankton (cells 2-20 μm) were the numerically abundant size group of biomineralizing phytoplankton across the GCB, with the coccolithophore *Emiliania huxleyi* and the diatoms *Fragilariopsis nana*, *F. pseudonana* and *PseudonitzschiaPseudo-nitzschia* sp. were the most numerically dominant and widely distributed species. A combination of SST, macro-nutrient concentrations and $p$CO$_2$ were the best statistical descriptors of biogeographic variability of biomineralizing species composition between stations. *Emiliania huxleyi* occurred in the silicic acid-depleted waters between the Sub-Antarctic Front and the Polar Front;, indicating a favorable environment for this coccolithophore species in the GCB after spring diatom blooms remove silicic acid to limiting levels. Multivariate statistics After statistical consideration of the influence of spatial variability in a diverse suite of environmental factors on the distribution of nanoplankton in the GCB, we identifiedy a combination of carbonate chemistry and macro-nutrients, co-varying with temperature, as the dominant drivers of

biomineralizing nanoplankton After*in* the GCB sector of the Southern Ocean. .

**1 Introduction**

The Great Calcite Belt (GCB), defined as an elevated particulate inorganic carbon (PIC) feature occurring alongside seasonally elevated chlorophyll *a* in austral spring and summer in the Southern Ocean (Fig. 1; Balch et al., 2005), plays an important role in climate fluctuations (Sarmiento et al. 1998, 2004), accounting for over 60% of the Southern Ocean area

10 (30-60$^o$S; Balch et al., 2011). The region between 30-50$^o$S has the highest uptake of anthropogenic carbon dioxide ($CO_2$) alongside the North Atlantic and North Pacific Ocean (Sabine et al., 2004). Our knowledge of the impact of interacting environmental influences on phytoplankton distribution in the Southern Ocean is limited. For example, we do not yet fully understand  how light and iron availability, or temperature and pH,  interact to control phytoplankton biogeography (Boyd et al., 2010, 2012; Charalampopoulou et al., 2016). Hence, if model parameterizations

15 are to improve (Boyd and Newton, 1999) to  provide  accurate predictions of  biogeochemical change, a multivariate understanding of the full suite of environmental drivers  is required.

20

The Southern Ocean has often been considered as a micro-plankton (20-200 μm) dominated system with phytoplankton blooms dominated by large diatoms and *Phaeocystis* sp. (e.g., Bathmann et al., 1997; Poulton et al., 2007; Boyd, 2002). However, since the  identification of the GCB as a consistent feature (Balch et al., 2005; 2016) and  recognition of  pico- (< 2 μm) and nanoplankton (2-20 μm) importance in High Nutrient Low Chlorophyll (HNLC) waters

25 (Barber and Hiscock, 2006), the dynamics of small (bio-)mineralizing plankton and their  export need to be acknowledged. The two dominant biomineralizing phytoplankton groups in the GCB are coccolithophores and diatoms. Coccolithophores are generally found north of the PF (e.g., Mohan et al., 2008), though *Emiliania huxleyi* has been observed as far south as 58$^o$S in the Scotia Sea (Holligan et al., 2010), at 61$^o$S across Drake Passage (Charalampopoulou et al., 2016) and 65$^o$S south of Australia (Cubillos et al., 2007).

30 Diatoms are present throughout the GCB, with the Polar Front marking a strong divide between different size fractions (Froneman et al., 1995). North of the PF, small diatom species  such as *Pseudo-nitzschia* sp. and

*Thalassiosira* sp. tend to dominate numerically, whereas large diatoms (> 20 µm) with higher silicic acid requirements (e.g. *Fragilariopsis kerguelensis*) are generally more abundant south of the PF (Froneman et al., 1995). High abundances of nanoplankton (coccolithophores, small diatoms, chrysophytes) have also been observed on the Patagonian shelf (Poulton et al., 2013) and in the Scotia Sea (Hinz et al., 2012). Currently, few studies incorporate small biomineralizing phytoplankton to species level (e.g., Froneman et al., 1995; Bathmann et al., 1997; Poulton et al., 2007; Hinz et al., 2012). Rather, the focus has often been on the larger and non-calcifying species of phytoplankton in the Southern Ocean due to sample preservation issues (i.e., acidified Lugol's solution dissolves calcite and light microscopy restricts accurate identification to cells > 10 µm; Hinz et al., 2012). In the context of climate change and future ecosystem function, the distribution of biomineralizing phytoplankton is important to define when considering phytoplankton interactions with carbonate chemistry (e.g., Langer et al., 2006; Tortell et al., 2008) and ocean biogeochemistry (e.g., Baines et al., 2010; Assmy et al., 2013; Poulton et al., 2013). The distribution of mineralizing phytoplankton is important to define when considering phytoplankton interactions with carbonate chemistry (e.g., Langer et al., 2006; Tortell et al., 2008) and ocean biogeochemistry (e.g., Baines et al., 2010; Assmy et al., 2013; Poulton et al., 2013).

The GCB begins south of ~30°S and extends to ~60°S covering an area of ~88 × 10$^6$ km$^2$ (Balch et al., 2011), spanning 
[revised manuscript text omitted]

---

## Author Response (AR2)

Comments from the editor

We thank the editor for taking the time to make technical comments on the manuscript and have responded as below

P2, L9: if model parameterizations are improved for providing accurate predictions …

5   This suggestion changes the emphasis of the sentence slightly, therefore we shall keep the original text.

P2, L29: cells < 10 µm

The text is correct, in this instance traditional examination of Lugol's samples is restricted to identification of the larger cells > 10 µm

P5, L4 and P17, L8: total dissolved inorganic carbon

10   Amended

P5, L4: total alkalinity

Amended

P7, L10: Phosphate followed by …

Not sure what the editor means here, have left the sentence as is.

15   P7, L31: "sp." should not be italicized.

Amended

P15, L29: Insert a comma immediately after "al.".

Amended throughout manuscript

P33, caption of Figure 1: The "a" of "Chlorophyll a" should be italic. Also the "-3" of "mg -3" should be superscript

20   Amended

[revised manuscript text omitted]